# Falling Trees: A Model Class for Interpretable Risk Prioritization

**Varun Babbar** [* 1]  **Zachery Boner** [* 1]  **Margo Seltzer** [2]  **Cynthia Rudin** [1]

## Abstract

Many real-world decisions require prioritizing high-risk cases, such as clinicians prioritizing high-risk patients before lower-risk ones. Falling rule lists (FRLs), which are ordered if–then rules with monotonically decreasing risks, provide an interpretable framework for such tasks; however, their single-path structure yields a highly restricted model class. We introduce falling trees, a new family of interpretable models that enforces the same monotonic risk constraint while permitting tree-structured branching. We present **GRAVITree**, a novel dynamic-programming-with-bounds algorithm for learning the Rashomon set of falling trees under depth and branching constraints. Our formulation can interpolate between rule lists and full decision trees, enabling user-desired model expressivity. In a new clinical dataset and in many public classification benchmarks, falling trees match or outperform FRLs and other interpretable baselines, often producing more sparse decisions for high-risk instances. Our results show that falling trees strike a practical balance between interpretability, expressiveness, and risk prioritization for high-stakes settings.

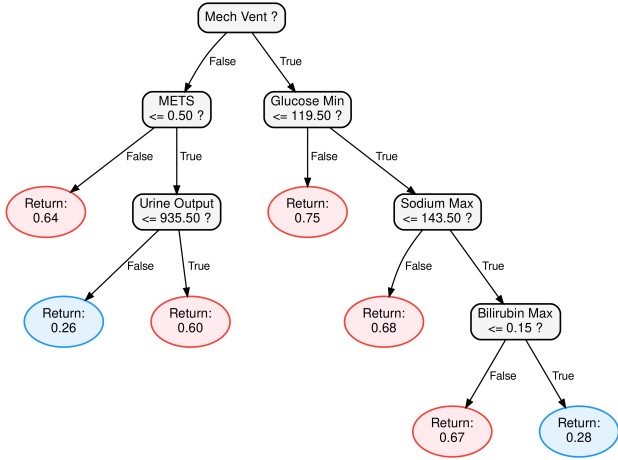

*Figure 1.* An example of a falling tree found by our algorithm on the MIMIC-III dataset. This model predicts mortality of ICU patients. All features represent the worst value in the first 24h of the patient's stay. Further details on the data and models are in Appendix B.

## 1. Introduction

*Falling Rule Lists* (FRLs) are an important logical model class for high-stakes decisions (Wang & Rudin, 2015; Chen & Rudin, 2018). FRLs are composed of a sequence of if-then rules such that the risks $P(y = 1|\mathbf{x})$ for the outcome $y$ given features $\mathbf{x}$ decrease ("fall") along the sequence. This falling constraint allows high-risk cases to be identified more quickly, because they are decided within the first few rules. In that sense, falling rule lists are most sparse for the highest risk examples.

However, the current algorithms for optimizing FRLs suffer from three major drawbacks. First, they are unable to guarantee full optimality; the work of Wang & Rudin (2015) uses a Bayesian sampling approach, and the work of Chen & Rudin (2018) uses a constrained Monte Carlo sampling approach to generate candidate rule lists. Second, they use rules generated beforehand with algorithms such as FPGrowth (Han et al., 2000). These rules are too numerous to be fully optimized with methods that compute provably optimal models (e.g., Angelino et al., 2017; Xin et al., 2022; Aglin et al., 2020). Finally, existing algorithms for optimizing FRLs suffer in practice from the *interaction bottleneck*, i.e., the difficulty of working with a machine learning algorithm to design predictive models (Rudin et al., 2024). The algorithms proposed by Wang & Rudin (2015) and Chen & Rudin (2018) output a single model in each iteration. This results in a slow and difficult feedback cycle between domain experts critiquing models and practitioners responding to these critiques.

We present a new model class called *falling trees*, which are constrained decision trees designed to address these three issues. We provide a corresponding algorithm called

---

[*]Equal contribution  [1]Department of Computer Science, Duke University, Durham, NC, USA [2]Department of Computer Science, University of British Columbia, Vancouver, BC, CA. Correspondence to: Varun Babbar <varun.babbar@duke.edu>, Zachery Boner <zachery.boner@duke.edu>.

*Proceedings of the $43^{rd}$ International Conference on Machine Learning*, Seoul, South Korea. PMLR 306, 2026. Copyright 2026 by the author(s).

**GRAVITree** to discover the set of all near-optimal falling trees (i.e., the *Rashomon set*, Rudin et al., 2024; Fisher et al., 2019; Semenova et al., 2022). Our new model class generalizes the notion of the falling constraint to trees rather than decision lists. Figure 1 shows an example of a falling tree on a dataset from the MIMIC-III dataset (Johnson et al., 2016).

Our optimization procedure incorporates a *branching penalty* that softly penalizes branches that deviate from the 'if/then' rule list structure. This results in models that branch out from the rule list structure either to avoid violating the falling constraint or when branching results in a sufficient performance increase on the training data. This effectively allows the model to create higher-order rules when necessary for good performance and avoids the precomputation of rules required for previous falling rule list optimization algorithms.

We present experiments showing that our approach significantly decreases the average number of decisions needed to make predictions on the positive class while maintaining similar or better test performance than other methods. We also present a case study in Appendix B that uses the MIMIC-III data to demonstrate **GRAVITree**'s ability to control the structure of trees and provide interpretable models.

## 2. Related Work

### 2.1. Decision Tree and Decision List Optimization

Classical approaches to decision tree optimization have focused on greedy induction, i.e., expanding on a split by choosing features that are locally optimal according to a heuristic such as the Gini index or classification accuracy (Breiman, 1984; Quinlan, 2014), but forgo closeness to global optimality. More recent work has substantially improved on these methods by leveraging dynamic-programming-with-bounds techniques to prune parts of the search space and recover provably optimal trees (McTavish et al., 2022; Aglin et al., 2020; van der Linden et al., 2023; Chaouki et al., 2025). Other recent work uses principled heuristics for fast recovery of near-optimal trees (Babbar et al., 2025; Heile et al., 2026). There is also work on learning the related model class of rule lists; this dates back at least to Rivest (1987); there are many papers on creating sets of rules and assembling them into lists. The CORELs algorithm of Angelino et al. (2017) is unique in fully optimizing rule lists using a dynamic-programming-with-bounds approach.

Our work falls into this general category of optimization for trees and lists with bounds, but the model class and problem formulation are unique, yielding new bounds and necessitating new optimization techniques. Specifically, we show that monotonicity (falling) constraints imposed on trees can

be used to further prune the search space, enhancing the efficiency of **GRAVITree**.

To our knowledge, only two works have proposed algorithms for learning falling rule lists. Wang & Rudin (2015), who introduced FRLs, proposed a Bayesian (generative) modeling approach and performed maximum a posteriori optimization. Chen & Rudin (2018) proposed an optimization-based approach that is more efficient for finding high-quality FRLs. At each iteration, they incrementally build a rule list by randomly selecting among rules (mined beforehand) that satisfy a set of bounds. These bounds are used to prune the search space and are the key to finding good models quickly. The algorithm reports the best FRL found after a fixed number of iterations. We modified this algorithm to output a Rashomon set of falling rule lists as a baseline for comparison with our method. In contrast to these methods, our approach does not rely on rules being constructed beforehand; it outputs trees with structures deviating from rule lists only when it is necessary to achieve good performance.

### 2.2. Why Rashomon Sets are Useful

There is rich theoretical, algorithmic, and sociotechnical literature demonstrating that Rashomon sets are useful in myriad ways. Kobylińska et al. (2023) show that analyzing predictions from diverse models in the Rashomon set can lead to more trust in model outputs and more robust decision making in high stakes settings. Hsu et al. (2026) show that Rashomon sets contain models that can perform well on various trustworthiness criteria such as fairness, robustness, and privacy. Nguyen et al. (2024) demonstrate accelerated convergence and improved predictive accuracy when using Rashomon set ensembles to perform active learning. Rudin et al. (2024) reviews many reasons why the Rashomon Effect is useful, from reliable variable importance analysis (Donnelly et al., 2023; 2026) to proofs of the existence of competitive simpler models on noisy tabular data (Semenova et al., 2022; 2023; Boner et al., 2024).

### 2.3. Rule-Based Models and Rashomon Sets

Previous work on finding the Rashomon set of sparse decision trees (Xin et al., 2022; Babbar et al., 2025; Arslan et al., 2025) uses dynamic-programming-with-bounds on search graphs to explore the space of decision trees efficiently. The bounds in these algorithms depend on a sparsity regularization term, which penalizes the number of leaves in a tree. Mata et al. (2022) find Rashomon sets of rule lists, using an approach combining ideas from Angelino et al. (2017) and Xin et al. (2022). Ciaperoni et al. (2024) find Rashomon sets of the related hypothesis space of rule sets.

These are all algorithms for Rashomon sets of logical models, but none of them consider class probabilities or constraints on the class probabilities, such as falling constraints.

# 3. Notation and Definitions

We now present background concepts and notation for formally defining the model class of falling trees.

## 3.1. Notation

Let $D = \{\mathbf{x}_i, y_i\}_{i=1}^n$ be a dataset of size $n$, where $\mathbf{x}_i \in \{0,1\}^K$ is a binary feature vector of size $K$ and $y_i \in \{0,1\}$ is a binary label. Our formulation is general enough to extend to continuous and categorical features that have been binarized; binarized features are also more intuitive for users (Warren et al., 2023). Given a search space $\mathcal{F}_d$ of decision trees of depth $\leq d$, we define a tree $T \in \mathcal{F}_d$ as a function $T : \{0,1\}^K \rightarrow [0,1]$ that takes a feature vector $\mathbf{x}$ as input and returns a probability (which can be thresholded to produce a binary outcome) after making a series of queries on $\mathbf{x}$, where each query involves one feature (e.g., `age` $> 70$). A *rule list* also involves a series of queries, but rule lists use multiple features per query (e.g., a rule may be "`age` $> 70$ and `chest_pain` $= 1$") and every node has at least one child that is a leaf. The probabilities in the leaves of FRLs monotonically decrease down the list. We denote $N(T)$ as the set of non-leaf nodes in $T$, $L(T)$ as the leaf set of $T$, $p^+(l)$ as the positive class probability of any leaf $l \in L(T)$, depth$(l)$ as the depth of a leaf (with the root at depth 0), and $1 \leq |l| \leq n$ as the support size of $l$ (the number of data points captured by the leaf).

## 3.2. Generalizing the Falling Constraint to Trees

Falling rule lists have simple constraints: the leaf probabilities decrease as we move down the list. We now generalize this definition to trees, which have a branching structure. We want the highest predicted risk samples to be positioned higher in the tree than the lower risk samples.

**Definition 3.1** (Adjacent Leaves). Given a root-to-leaf path $P \subseteq N(T) \cup L(T)$ in tree $T$, a leaf $l$ is *adjacent* to $P$ if it is either in $P$ or is the child of a node in $P$, and it is *not* a sibling of the terminal leaf. Define $L(P, T) \subseteq L(T)$ to be the set of leaves adjacent to path $P$, ordered by depth.

**Definition 3.2** (Falling Constraint for Trees). Let $\mathcal{P}(T)$ be the set of all root-to-leaf paths in tree $T$. $T$ is *falling* if, $\forall P \in \mathcal{P}(T), \forall l \in L(P, T)$, $p^+(l)$ is monotonically decreasing in depth. We denote $F(T, D) \in \{0, 1\}$ as the falling constraint, where 1 is satisfaction and 0 otherwise.

This constraint implies that along any given path, leaves with higher positive class probabilities must be closer to the root. It is easy to verify that a falling rule list satisfies this definition; each leaf in an FRL must necessarily be the highest probability leaf in the subtree rooted in the internal node above it, and this leaf is at a higher depth than any other leaf in the subtree. Figure 2 shows an example of a falling tree, providing a visual aid for its definitions.

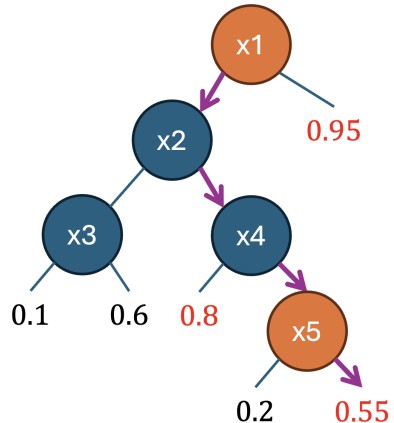

*Figure 2.* A toy example decision tree that satisfies the falling constraint. One path from root to leaf is denoted by purple arrows. The '0.2' leaf is not on the path, because it is a sibling of the terminal '0.55' leaf. The red highlighted leaves are adjacent to the path and their empirical positive proportions decrease monotonically. The falling constraint is satisfied for all paths in this tree. This model incurs branching cost $H(T) = 1$.

## 3.3. Branching Cost Formulation

Sparse decision tree algorithms optimize the number of leaves but not the shape of the tree. A falling rule list (with one condition per query) is a highly *imbalanced* tree, as it is completely one-sided. We may find trees that are more useful, intuitive, or have lower decision sparsity (fewer queries per decision) if we can control tree shape. We now introduce a novel formulation that enables an optimization algorithm to control tree shape, specifically imbalance, allowing the user to gradually transition between complete decision trees and rule lists. Given a decision tree $T$ with internal nodes $N(T)$, we define the branching cost of $T$, $H(T)$, as

$$H(T) := \sum_{n \in N(T)} \mathbb{1}[\text{both of } n\text{'s children are internal nodes}].$$

In other words, the branching cost is the number of internal nodes that branch into two different non-leaf subtrees. Any tree with a branching cost of 0 is a rule list, and a tree on $d$ variables with a branching cost of $2^{d-1}$ is a complete tree. This formulation, however, still does not ensure sparsity with respect to the number of leaves. One could hypothetically construct a rule list of arbitrary length that incurs no branching cost penalty. To resolve this, we impose a maximum depth and a minimum support in each leaf to eliminate arbitrarily deep falling trees from our search space. These are standard constraints used in many decision tree approaches (Breiman, 1984; 2001; Quinlan, 2014; van der Linden et al., 2023). With the branching cost, depth constraint, and support constraint, we are able to achieve various attributes of trees we care about, i.e., *sparsity*, *falling-ness*, and *rule-listy-ness*.

# 4. GRAVITree: Finding the Rashomon Set of Falling Trees

**GRAVITree** uses a novel objective function that penalizes the number of misclassifications as well as the number of *branches* in the tree:

$$\mathcal{L}(T, D) = \frac{1}{n} \sum_{i=1}^{n} \mathbb{1}[T(\mathbf{x}_i) \neq y_i] + \mu H(T) \qquad (1)$$

where $\mu$ is the branching penalty. We now define the following optimization problem:

$$\mathcal{L}^*(D) = \min_{T \in \mathcal{F}_d} \mathcal{L}(T, D) \text{ such that} \qquad (2)$$

$$\min_{l \in L(T)} \frac{|l|}{n} \geq \alpha, F(T, D) = 1 \qquad (3)$$

where $\alpha$ is the minimum leaf support and $d$ is the depth budget. Given tolerance $\varepsilon \geq 0$, the Rashomon set of falling trees is

$$\mathcal{R}(D, d, \mu, \varepsilon) := \begin{cases} T \in \mathcal{F}_d : \mathcal{L}(T; D) \leq (1+\varepsilon)\mathcal{L}^*(D), \\ \min_{l \in L(T)} \frac{|l|}{n} \geq \alpha, \ F(T, D) = 1 \end{cases}.$$
$$(4)$$

Our algorithm for finding $\mathcal{R}(D, d, \mu, \varepsilon)$ recursively constructs a search graph. It assigns a budget to each node in the search graph, pruning candidates whose optimal completions do not keep them in the Rashomon set. It also applies pruning and caching strategies that exploit the falling constraint, making the search process efficient. Once the Rashomon sets for a node's left and right children are found, it merges and filters them according to the falling constraint.

As part of finding Rashomon sets defined in Equation 4, we find Rashomon sets of subproblems, where a subproblem is defined by a subset of data. For subset $D_{\text{sub}} \subseteq D$, we need to find optimal trees according to $\mathcal{L}^*(D_{\text{sub}})$.

**Finding the optimal falling tree for a subproblem.** A key subroutine in **GRAVITree** is Algorithm 1, which uses dynamic programming with bounds to compute the optimal falling tree for a given subproblem. It returns two solutions: the best overall solution $S^* = \{\mathcal{L}^*(D), T^*, p_{\max}^+(T^*)\}$ (loss, best tree, and maximum leaf probability observed in the tree), and, if $T^*$ is a leaf, the best solution that is not a leaf $S^\diamond = \{\mathcal{L}^\diamond(D), T^\diamond, p_{\max}^+(T^\diamond)\}$ (or $\emptyset$ if none is feasible). The alternative solution $S^\diamond$ handles interactions between the falling constraint and branching penalty $\mu$.

After reaching the base case (terminal depth), we construct a set of candidate features and prune those whose child subproblems (a) violate min-support $\alpha$ or (b) have empirical positive proportion exceeding $p_{\text{recent}}^+$ and thus inevitably violate falling (Theorem 4.2). Each remaining feature is evaluated by recursively solving its child subproblems.

Unlike standard tree search, child subproblems are coupled: the branching penalty depends on whether both children are internal nodes, and the falling constraint ties the bound on each child's leaves to the sibling's leaf probabilities. To resolve this, we recurse first on the child $h$ with the larger positive leaf probability. Let $T_h$ be the corresponding optimal tree found on child $h$ and $p_{\max}^+(T_h)$ the largest leaf probability observed in any leaf in $T_h$ (resp. $T_l, p_{\max}^+(T_l)$). We now examine 2 cases:

**Case 1 (line 14): The best solution for child $h$ is a leaf.** Two sub-options are compared: *(A)* keep the leaf $T_h$ and solve $\ell$ under the tightened bound $p_{\max}^+(T_h)$ (lines 16–19), since any leaf in $\ell$ adjacent to a leaf at $h$ must satisfy falling; or *(B)* don't use the best solution, rather use $S_h^\diamond$ (the best non-leaf at $h$, which is the second best solution) and solve $\ell$ under the original $p_{\text{recent}}^+$ (lines 21–23), which is feasible because $h$ being internal removes the adjacent-leaf constraint. Note that if $S_h^\diamond$ does not exist (i.e. is an empty set) then case B will not be encountered.

**Case 2 (line 24): The best solution for child $h$ is a tree.** We solve $\ell$ under $p_{\text{recent}}^+$. If $T_\ell$ is a leaf, that induces a new constraint on the other child, which creates two sub-options: *(A)* use the optimal solution for the $\ell$ side (which is a leaf, and which induces a constraint on the $h$ side) and re-solve the $h$ side, or *(B)* use the optimal solution for the $h$ side and use the second best solution for the $\ell$ side (which is a tree). That is, we either: *(A)* re-solve $h$ under the tightened bound $p_{\max}^+(T_\ell)$ (lines 29–32), or *(B)* keep $T_h$ and use $S_\ell^\diamond$ (lines 34–36). Sub-option B incurs no extra computational cost, since $S_\ell^\diamond$ was already returned alongside $S_\ell$. As mentioned above, if $S_h^\diamond$ does not exist (i.e. is an empty set) then we need not consider case B.

Lastly, if $T_\ell$ is a tree, neither child constrains the other, and the solutions combine directly. The loss-minimizing solution is returned as $S^*$, and the loss-minimizing non-leaf solution (i.e., the second best solution) as $S^\diamond$, if $S^*$ is a leaf.

**Falling constraint-based pruning strategies.** We can exploit the falling constraint to prune large parts of the search space for both the optimal solutions $\mathcal{L}^*(D_{\text{sub}})$ for each subproblem $D_{\text{sub}}$ using Algorithm 1 and for finding the Rashomon set, which is presented in Algorithm 3. The pruning is done distinctly at local and global levels. Local pruning occurs before making any recursive calls on the child subproblems. If the fraction of positive examples in a child node exceeds the positive probability of the most recent preceding leaf $p_{\text{recent}}^+$ on the path containing the parent, the feature is immediately discarded as a violation of monotonicity. We prove that this is a valid pruning method in Theorem 4.2.

**Algorithm 1** OPTFALLINGTREE

**Require:** Feature matrix $X$, labels $y$, $I$ row indices of $X$, depth budget $d$, branching cost $\mu$, feature set $F$, min support $\alpha$, global dataset size $n$, current path $P$, max positive-class proportion (seen on the most recent leaf adjacent to $P$) $p_{\text{recent}}^+$,

1: $p_I^+ \leftarrow \mathbb{E}[y[I]]; \quad \mathcal{L}_{\text{leaf}}(I) \leftarrow \frac{\#\text{minority in } I}{n}$
2: $S^* \leftarrow (\mathcal{L}_{\text{leaf}}(I), \text{LEAF}(p_I^+), p_I^+); \quad S^\diamond \leftarrow \emptyset$
3: **if** $d = 0$ **then**
4:     **return** $(S^*, S^\diamond)$ {Base case}
5: **end if**
6: Build candidate feature set $C$ (prune features $j \in F$ based on min-support, $\max(p_L^+, p_R^+) \le p_{\text{recent}}^+$)
7: **for** each $(j, \dots) \in C$ **do**
8:     $(I_h, P_h) \leftarrow$ row indices and path of higher-$p^+$ child
9:     $(I_\ell, P_\ell) \leftarrow$ row indices and path of lower $p^+$ child.
10:     $(S_h, S_h^\diamond) \leftarrow$ OPTFALLINGTREE$(\dots, I_h, P_h, p_{\text{recent}}^+)$
11:     $(L_h, T_h, p_{\max}^+(T_h)) \leftarrow S_h$ {$S_h$ contains these 3 things}
12:     $(L_h^\diamond, T_h^\diamond, p_{\max}^+(T_h^\diamond)) \leftarrow S_h^\diamond$
13:     $S_j \leftarrow \emptyset$ {Initializing solution for split $j$}
14:     **if** $T_h$ is a leaf **then**
15:         ▷ A: solve low problem under tighter $p_{\max}^+(T_h)$ bound
16:         **if** $p_{\max}^+(T_h) \le p_{\text{recent}}^+$ **then**
17:             $(S_\ell^A, \cdot) \leftarrow$ OPTFALLINGTREE$(\dots, I_\ell, P_\ell, p_{\max}^+(T_h))$
18:             $S_{j,T_h,A} \leftarrow$ COMBINE$(j, S_h, S_\ell^A)$
19:         **end if**
20:         ▷ B: solve low problem under original $p_{\text{recent}}^+$ bound
21:         $(S_\ell^B, \cdot) \leftarrow$ OPTFALLINGTREE$(\dots, I_\ell, P_\ell, p_{\text{recent}}^+)$
22:         $S_{j,T_h,B} =$ COMBINE$(j, S_h^\diamond, S_\ell^B)$
23:         $S_j \leftarrow \min(S_{j,T_h,A}, S_{j,T_h,B})$
24:     **else**
25:         $T_h$ is a tree
26:         $(S_\ell, S_\ell^\diamond) \leftarrow$ OPTFALLINGTREE$(\dots, I_\ell, P_\ell, p_{\text{recent}}^+)$
27:         **if** $T_\ell$ is a leaf **then**
28:             ▷ A: re-solve high prob under tighter $p_{\max}^+(T_\ell)$ bound
29:             **if** $p_{\max}^+(T_\ell) \le p_{\text{recent}}^+$ **then**
30:                 $(S_h', \cdot) \leftarrow$ OPTFALLINGTREE$(\dots, I_h, P_h, p_{\max}^+(T_\ell))$
31:                 $S_{j,T_\ell,A} \leftarrow$ COMBINE$(j, S_h', S_\ell)$
32:             **end if**
33:             ▷ B: keep $T_h$, use second best solution $T_\ell^\diamond$
34:             $S_{j,T_\ell,B} =$ COMBINE$(j, S_h, S_\ell^\diamond)$
35:             $S_j \leftarrow \min(S_{j,T_\ell,A}, S_{j,T_\ell,B}))$
36:         **else**
37:             Optimal solutions for $h$ and $l$ problems are trees
38:             $S_j \leftarrow$ COMBINE$(j, S_h, S_\ell)$
39:         **end if**
40:     **end if**
41:     **if** $S_j \ne \emptyset$ **then**
42:         $S^* \leftarrow \min(S^*, S_j); \quad S^\diamond \leftarrow \min(S^\diamond, S_j)$
43:     **end if**
44: **end for**
45: **return** Best solution $S^*$ and best non-leaf solution $S^\diamond$

---

**Algorithm 2** COMBINE

**Require:** Split feature $j$, left solution $S_a = (\mathcal{L}_a, T_a, p_a^+)$, right solution $S_b = (\mathcal{L}_b, T_b, p_b^+)$, branching cost $\mu$
1: $\mathcal{L} \leftarrow \mathcal{L}_a + \mathcal{L}_b + \mu \cdot \mathbf{1}[T_a, T_b \text{ both internal}]$
2: $T \leftarrow$ NODE$(j, T_a, T_b)$ {Split on $j$}
3: $p^+ \leftarrow \max(p_a^+, p_b^+)$ {Store the highest leaf probability associated with this new tree}
4: **return** Combined sol $S = (\mathcal{L}, T, p^+)$

---

**Algorithm 3** GRAVITree

**Require:** Feature matrix $X$, labels $y$, $I$ row indices of $X$, depth budget $d$, branching cost $\mu$, feature set $F$, min support $\alpha$, global dataset size $n$, current path $P$, max positive-class proportion (seen on the most recent leaf adjacent to $P$) $p_{\text{recent}}^+$, Rashomon bound $B = (1 + \varepsilon)\mathcal{L}^*$

1: $\mathcal{L}_{\text{leaf}}(I) \leftarrow \frac{\#\text{minority class points in } I}{n}$
2: **if** $(\mathcal{L}_{\text{leaf}}(I) \le B)$ **then** $\mathcal{R} \leftarrow$ LEAF$(\mathcal{L}_{\text{leaf}}(I))$ **else** $\mathcal{R} \leftarrow \emptyset$
3: **if** $d = 0$ **then**
4:     **return** $\mathcal{R}$ {Base case}
5: **end if**
6: $C \leftarrow \emptyset$
7: Build candidate feature set $C$ (prune feaures $j \in F$ based on min-support, both left and right positive class probabilities are $\le p_{\text{recent}}^+$) {Same as Algorithm 1}
8: **for** each $(j, \dots) \in C$ **do**
9:     $I_{\text{left}}^j, I_{\text{right}}^j =$ split $I$ into left and right children
10:     $(S_{\text{left}}, .) \leftarrow$ OPTFALLINGTREE$(.., I_{\text{left}}^j, p_{\text{recent}}^+)$
11:     $(S_{\text{right}}, .) \leftarrow$ OPTFALLINGTREE$(.., I_{\text{right}}^j, p_{\text{recent}}^+)$
12:     $(L_{\text{left}}, T_{\text{left}}, .) \leftarrow S_{\text{left}}$
13:     $(L_{\text{right}}, T_{\text{right}}, .) \leftarrow S_{\text{right}}$
14:     **if**
    $(L_{\text{left}} + L_{\text{right}} + \mu \cdot \mathbf{1}[T_{\text{left}}, T_{\text{right}} \text{ are internal nodes}]) > B$ **then**
15:         **continue** {Prune if lower bounds exceed budget}
16:     **end if**
17:     Let $h, \ell$ be the children of split $j$ with higher and lower $p^+$ respectively
18:     $(I_h, P_h, L_h) \leftarrow$ indices, path, loss of child $h$
19:     $(I_\ell, P_\ell, L_\ell) \leftarrow$ indices, path, loss of child $\ell$
20:     $\mathcal{R}_h \leftarrow$ **GRAVITree**$(\dots, I_h, B - L_\ell, P_h, \dots, p_{\text{recent}}^+)$ {Initial recursion on the higher probability leaf}
21:     $p_{\text{temp}}^j = p_{\text{recent}}$
22:     **if** $\mathcal{R}_h$ only contains a single leaf **then**
23:         Leaf$_{\mathcal{R}_h} \leftarrow$ leaf in $\mathcal{R}_h$
24:         **if** $p^+($Leaf$_{\mathcal{R}_h}) \ge p_{\text{recent}}^+$ **then continue**
25:         **else** $p_{\text{temp}}^j = p^+($Leaf$_{\mathcal{R}_h})$ {Tighten falling constraint}
26:     **end if**
27:     $\mathcal{R}_\ell \leftarrow$ **GRAVITree**$(\dots, I_\ell, B - L_h, P_\ell, \dots, p_{\text{temp}}^j)$ {Recurse lower probability leaf on tighter falling constraint}
28:     $\mathcal{R} \leftarrow \mathcal{R} \cup$ MERGEANDFILTERRSET$(\mathcal{R}_l, \mathcal{R}_h, j, B, \mu)$
29: **end for**
30: **return** SORT$(\mathcal{R})$

---

Global pruning is achieved by propagating constraints across sibling nodes. We first explore the subproblem with the higher positive class proportion because it is more likely to be solved quickly and result in a leaf. This value becomes

a strict upper bound ($p_{\text{tight}}^+$) for the low-probability sibling ($I_{\text{low}}$). Any tree in the $I_{\text{low}}$ subspace that contains a leaf with probability greater than $p_{\text{tight}}^+$ would violate the global falling order and can be pruned without full evaluation. We prove this constraint in Theorem 4.1 below:

**Theorem 4.1** (Necessary and sufficient condition for falling trees). *Let $T$ be a decision tree. Let $v \in N(T)$ be an internal node of $T$ with children $v_1$ and $v_2$ such that $v_1$ is a leaf and $v_2$ is an internal node. Note $v_1$ and $v_2$ can be either the left or right child of $v$. Denote $p_1^+$ to be the empirical positive proportion of $v_1$, and let $p_{\max}^+(v_2)$ be the maximum positive proportion of any leaf in the subtree rooted at $v_2$.*

*$T$ satisfies the falling constraint if and only if, for all nodes $v \in N(T)$ where one child is a leaf and the other is an internal node, $p_1^+ \geq p_{\max}^+(v_2)$.*

**Theorem 4.2** (Falling trees allow for enhanced pruning). *Suppose that we run Algorithm 1, and we are solving a subproblem corresponding to path $P$ consisting of a sequence of feature splits $P = \{(j_1, \gamma_1), (j_2, \gamma_2), \ldots, (j_P, \gamma_P)\}$, where $1 \leq j_k \leq K$ and $\gamma_k \in \{0, 1\}$ indicates that $P$ splits on the rule $j_k = \gamma_k$. Let $p_{recent}^+$ denote the empirical positive proportion of the deepest leaf adjacent to $P$ (this leaf is the optimal solution to its corresponding subproblem). Let $j \in \{1, \ldots, K\}$ be a candidate split feature, and let $p_L^+, p_R^+$ denote the empirical positive proportions of the left ($j = 0$) and right ($j = 1$) children created by splitting on $j$. Then, if $\max(p_L^+, p_R^+) > p_{recent}^+$, any tree $T_P$ grown from $P$ via a split on $j$ will violate the falling constraint, and $j$ may be eliminated as a candidate.*

**Recursive filtering and merging process.** An important aspect of our algorithm is the recursive filtration and merging process. The Rashomon set output during a recursive call is sorted by objective, enabling early pruning when cross products of the left and right children are taken (Algorithm 4). More precisely, a Rashomon set $\mathcal{R}_s$ corresponding to subproblem $s$ is stored as a min-heap (list) of tuples $[(T_1, \mathcal{L}(T_1, D), \text{metadata}_{T_1}), \ldots.]$ containing trees and their metadata, with the heap created with respect to the objective function. For any tree $T \in \mathcal{R}_s$, the metadata we store includes the tree objective and the maximum positive leaf probability over all of its existing leaves $\max_{l \in L(T)} p^+(l)$. During the merge step, for every tree in $s$'s left child's Rashomon set, $T_L \in \mathcal{R}_L$, we iterate through trees within $s$'s right child's Rashomon set, $T_R \in \mathcal{R}_R$. Since $\mathcal{R}_R$ is sorted by loss, as soon as the combined loss $\mathcal{L}(T_L) + \mathcal{L}(T_R)$ exceeds the budget $B$, we can terminate the inner loop immediately. This transforms the potentially quadratic complexity of the merge operation into a log-linear time operation.

**Caching subproblems.** Since the same subset of data indices $D_{\text{sub}}$ may be reached via different splits in the search graph (permutation of features), we memoize to avoid redundant computations. Regular tree algorithms such as those of McTavish et al. (2022); Babbar et al. (2025) maintain a hash map where the key is defined by the bitmask of sample indices $D_{\text{sub}}$ and the current depth $d$. However, for the falling tree problem, caching becomes more challenging as the state must strictly include the incoming constraint $p_{\text{recent}}^+$, as the valid set of trees for a subproblem depends on the permissible probability ceiling. To still leverage the benefits of caching in this setting, we cache the optimal solution associated with the tuple $\left(\text{subproblem } D_{\text{sub}}, d, \lceil \frac{p_{\text{recent}}^+}{q} \rceil\right)$, where $q \in [0, 1]$ is a quantization parameter that allows for some slack in $p_{\text{recent}}^+$ and enables caching of at least a subset of the $d!$ paths that reach a subproblem. This approach makes the algorithm faster with minimal changes to the optimal objective and the Rashomon set size. Note that if we retrieve cached solutions during the search process, we still check them for falling violations given the local value of $p_{\text{recent}}^+$ in the current call, so all output trees are guaranteed to obey the falling constraint.

## 5. Experiments

We now show three sets of experiments that evaluate **GRAVITree**'s ability to achieve the desired tradeoffs between the global loss, performance on positive examples, and the expected decision sparsity on high-risk examples. The first set of experiments shows that we can effectively interpolate between wide decision trees, '*rule-listy*' decision trees, and rule lists. In these experiments, we examine the runtime, the '*rule-listy-ness*' of models, and the size of the Rashomon set for different branching penalties $\mu$. As an illustration, Figure 3 shows two decision trees, one with a normalized Colless index of 1 and another with an NCI of 0.

The second set of experiments compares **GRAVITree** to TreeFARMS (Xin et al., 2022). TreeFARMS is an algorithm for finding the Rashomon set of sparsity-regularized decision trees (where sparsity is defined as the number of leaves). This model class has been established to be (a) competitive in performance with more complicated models (McTavish et al., 2022; Arslan et al., 2025; Chaouki et al., 2025), and (b) a conveniently interpretable model class (Wang et al., 2022). We show that, despite a small tradeoff to overall loss, we typically achieve better performance and decision sparsity on the positive class.

The third set of experiments compares **GRAVITree** with a mild branching cost to **GRAVITree** with a branching cost of 1.0, which forces the algorithm to find the Rashomon set of falling rule lists. We show that falling trees achieve better overall loss, comparable performance on positive examples, and slightly worse decision sparsity compared to falling rule lists.

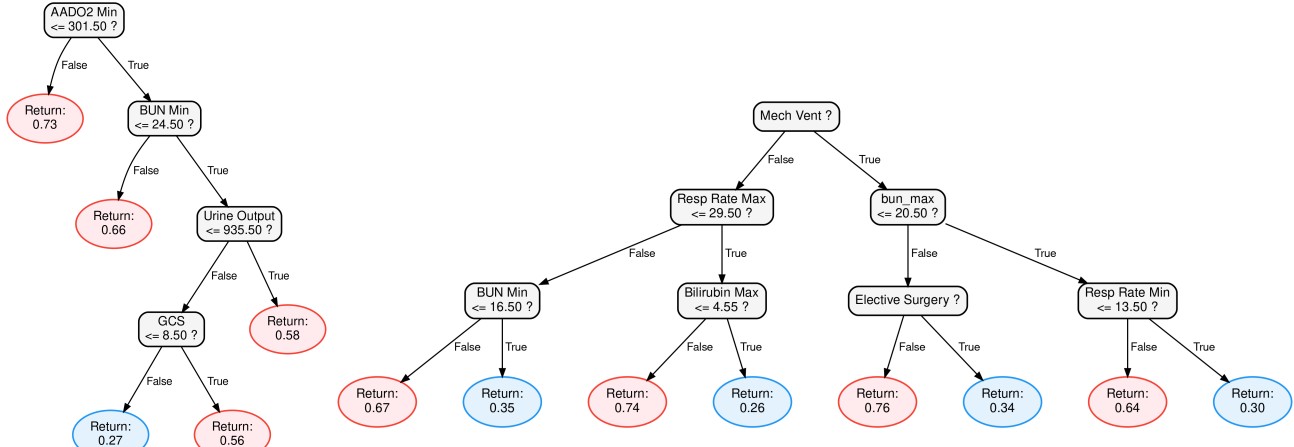

*Figure 3.* Two example trees found on the MIMIC dataset. The rule list is in the Rashomon set with a depth of 4, a Rashomon parameter of 0.05, branching cost 0.01, and minimum support 0.04. The fully grown tree was found by setting the depth to 3, with no branching cost or min support to allow the tree to be fully grown.

### 5.1. Experimental Setup

We ran all methods on 21 datasets sourced from the UCI Machine Learning Repository and former Kaggle competitions. Details on the datasets can be found in Appendix F. For some experiments, we vary the branching penalty $\mu$ to test its effect on tree structure and performance. For each dataset, we take five 80-20 train-test splits, compute the Rashomon set on the training split, compute the metrics described below on the test set (averaged across all trees in the Rashomon set), and take the mean and standard error over the splits.

**Baselines.** Our main external baseline is the TreeFARMS algorithm (Xin et al., 2022) for finding the unconstrained Rashomon set of decision trees (albeit on a different objective). We also show results with other Rashomon set methods, such as Arslan et al. (2025) and a modified version of Chen & Rudin (2018) in the appendix.

**Metrics.** We evaluate the properties of the Rashomon set of falling trees using several metrics. Our first metric, *decision sparsity*, measures the expected number of decisions made when predicting on a random example. For a tree $T$ and dataset $D$,

$$\beta(T, D) = \mathbb{E}_{\mathbf{x} \sim D}[\text{depth}_T(\mathbf{x})]$$
$$= 1 + \mathbb{P}(\mathbf{x}_f = 1)\beta(T_1, D_1) + \mathbb{P}(\mathbf{x}_f = 0)\beta(T_0, D_0),$$

where $f = \text{feature}(T)$, $T_b$ is the $b$-child of $T$, and $D_b = \{\mathbf{x} \in D : \mathbf{x}_f = b\}$ for $b \in \{0, 1\}$. We define $\beta(T, D^+)$ as the sparsity of all examples whose label is positive.

To quantify how '*rule-listy*' a tree is (i.e., its imbalance), we use the *normalized Colless index*. For each internal node

$v$ of $T$, let $n_L(v)$ and $n_R(v)$ denote the number of leaves in its left and right subtrees. The (unnormalized) Colless index is $C(T) = \sum_{v \in \mathcal{I}(T)} |n_L(v) - n_R(v)|$, where $\mathcal{I}(T)$ is the set of internal nodes. For a binary tree with $n$ leaves, the maximum value is $C_{\max}(n) = \frac{(n-1)(n-2)}{2}$, yielding the normalized Colless index: $\text{NCI}(T) = \frac{C(T)}{C_{\max}(n)} \in [0, 1]$, where $\text{NCI}(T) = 0$ corresponds to a perfectly balanced tree and $\text{NCI}(T) = 1$ corresponds to a maximally imbalanced (rule-list-like) tree. We also report the overall misclassification error and the false negative rate.

**GRAVITree can structurally interpolate between trees and rule lists.** Our first experiment varies the branching cost $\mu \in [0, 0.005, 0.01, 0.02, 0.03, 0.04, 0.05, 0.075, 0.1]$ and analyses the effect of the branching cost on $\text{NCI}(T)$, the Rashomon set size, and the runtime. To ensure a fair starting point, we set the same Rashomon bound ($1 + \varepsilon$ times the loss of the optimal tree with $\mu = 0$) across all branching costs.

Figure 4 shows that as we increase $\mu$, the average NCI of the Rashomon set approaches 1, suggesting that all trees become rule lists. Because it operates in a more restricted model class, we also see that the corresponding Rashomon set size and runtime tend to reduce with increasing $\mu$. The $\mu$ parameter is different from the sparsity parameter $\lambda$ often seen in the decision tree literature (McTavish et al., 2022; van der Linden et al., 2023; Babbar et al., 2025), and offers the ability to directly control the branching factor.

**GRAVITree vs Optimal Sparse Decision Trees.** In this experiment, we compare **GRAVITree** with TreeFARMS (Xin et al., 2022), an algorithm for finding the exact Rashomon set of sparse decision trees. TreeFARMS solves

| Dataset | GRAVITree $\mu = 0.01$ (trees) | | | | GRAVITree $\mu = 1$ (rule lists) | | | |
| --- | --- | --- | --- | --- | --- | --- | --- | --- |
| | RSet Size | Test Loss ↓ | $\beta(T, D+)$ ↓ | FNR ↓ | RSet Size | Test Loss ↓ | $\beta(T, D+)$ ↓ | FNR ↓ |
| NIJ Recidivism | 227.4 ± 30.0 | 0.305 ± 0.003 | 2.79 ± 0.04 | **0.126 ± 0.010** | 16.6 ± 5.5 | 0.311 ± 0.007 | **2.47 ± 0.16** | 0.183 ± 0.018 |
| Adult | 1.0 ± 0.0 | **0.204 ± 0.003** | 3.39 ± 0.05 | 0.206 ± 0.008 | 6.6 ± 2.1 | 0.290 ± 0.002 | **2.39 ± 0.28** | **0.089 ± 0.011** |
| Aging | 57.0 ± 90.4 | 0.385 ± 0.020 | 3.83 ± 0.11 | 0.379 ± 0.043 | 13.0 ± 12.1 | 0.402 ± 0.056 | **3.18 ± 0.51** | 0.368 ± 0.060 |
| Bar | 8.8 ± 2.9 | 0.299 ± 0.016 | 2.82 ± 0.07 | **0.414 ± 0.036** | 3.4 ± 3.1 | 0.317 ± 0.019 | **1.99 ± 0.11** | 0.567 ± 0.034 |
| Bar7 | 4.6 ± 2.5 | 0.309 ± 0.016 | 2.31 ± 0.41 | 0.523 ± 0.067 | 2.0 ± 0.0 | 0.311 ± 0.014 | 2.02 ± 0.12 | 0.556 ± 0.037 |
| BCW | 1.0 ± 0.0 | 0.070 ± 0.014 | 2.96 ± 0.31 | **0.075 ± 0.019** | 2.0 ± 0.0 | 0.081 ± 0.019 | 2.92 ± 0.13 | 0.221 ± 0.056 |
| Car Evaluation | 1.0 ± 0.0 | **0.141 ± 0.017** | 2.09 ± 0.20 | 0.000 ± 0.000 | 3.0 ± 1.0 | 0.376 ± 0.018 | **1.07 ± 0.06** | 0.000 ± 0.000 |
| Carryout Takeaway | 12.4 ± 9.4 | 0.371 ± 0.008 | 3.12 ± 0.23 | 0.333 ± 0.004 | 5.0 ± 4.5 | 0.342 ± 0.034 | 2.13 ± 0.29 | **0.258 ± 0.037** |
| Coffee House | 77.2 ± 99.7 | **0.300 ± 0.010** | 3.39 ± 0.09 | 0.253 ± 0.011 | 2.6 ± 3.0 | 0.341 ± 0.020 | 2.54 ± 0.15 | 0.354 ± 0.022 |
| COMPAS | 1248.4 ± 962.5 | 0.331 ± 0.013 | 2.98 ± 0.05 | 0.379 ± 0.017 | 154.6 ± 55.7 | 0.349 ± 0.013 | 3.08 ± 0.08 | 0.360 ± 0.014 |
| Heart | 1.0 ± 0.0 | **0.190 ± 0.028** | 3.49 ± 0.20 | 0.186 ± 0.030 | 23.4 ± 26.4 | 0.272 ± 0.030 | **1.85 ± 0.19** | 0.263 ± 0.073 |
| FICO | 538 ± 383 | **0.292 ± 0.008** | 2.73 ± 0.22 | 0.161 ± 0.013 | 12.0 ± 20.0 | 0.312 ± 0.007 | 2.15 ± 0.52 | **0.103 ± 0.015** |
| MGH | 2.6 ± 0.5 | **0.061 ± 0.006** | 2.01 ± 0.01 | 0.025 ± 0.010 | 2.4 ± 0.9 | 0.232 ± 0.013 | **1.22 ± 0.31** | 0.028 ± 0.012 |
| Monks1 | 12.0 ± 6.9 | 0.178 ± 0.123 | 2.81 ± 0.57 | **0.151 ± 0.150** | 2.2 ± 0.4 | 0.288 ± 0.100 | **1.27 ± 0.22** | 0.423 ± 0.129 |
| Monks2 | 1.0 ± 0.0 | 0.441 ± 0.095 | 3.55 ± 0.31 | **0.615 ± 0.054** | 2.2 ± 0.4 | 0.418 ± 0.053 | **0.47 ± 0.51** | 0.938 ± 0.138 |
| Monks3 | 1.0 ± 0.0 | **0.096 ± 0.100** | 3.52 ± 0.32 | **0.150 ± 0.137** | 27.0 ± 7.2 | 0.351 ± 0.026 | **1.65 ± 0.06** | 0.346 ± 0.070 |
| Mushroom | 1.0 ± 0.0 | **0.030 ± 0.002** | 2.51 ± 0.02 | 0.034 ± 0.006 | 3.0 ± 0.0 | 0.053 ± 0.003 | **1.43 ± 0.02** | 0.030 ± 0.003 |
| Restaurant 20 | 3.4 ± 1.5 | 0.298 ± 0.012 | 1.65 ± 0.23 | 0.250 ± 0.014 | 2.8 ± 0.4 | 0.298 ± 0.012 | 1.55 ± 0.10 | 0.249 ± 0.012 |
| Spambase | 1.0 ± 0.0 | **0.105 ± 0.013** | 3.17 ± 0.32 | 0.189 ± 0.031 | 12.8 ± 6.7 | 0.126 ± 0.007 | **2.43 ± 0.06** | 0.178 ± 0.027 |
| Student | 1.0 ± 0.0 | 0.192 ± 0.051 | 3.75 ± 0.29 | 0.182 ± 0.080 | 2.8 ± 1.3 | 0.224 ± 0.060 | **1.38 ± 0.55** | 0.120 ± 0.088 |
| Wine | 349 ± 292 | **0.294 ± 0.022** | 3.06 ± 0.15 | 0.184 ± 0.017 | 19.4 ± 20.2 | 0.321 ± 0.016 | **2.73 ± 0.21** | **0.160 ± 0.025** |

*Table 1.* Comparison between **GRAVITree** with $\mu = 0.01$ and $\mu = 1$. We set the min support $\alpha = 0.05$ for both configurations. The latter penalty forces all models in the Rashomon set to be falling rule lists. **Bold** indicates a statistically significant difference (Welch's $t$-test, $p < 0.05$); underlined values indicate the method is numerically better but not statistically significantly so. While **GRAVITree** with $\mu = 0.01$ consistently has better test loss than its rule list counterpart, the latter generally makes more sparse decisions on the positive class. Performance on false negative rate (FNR) is mixed, with neither configuration uniformly dominating the other.

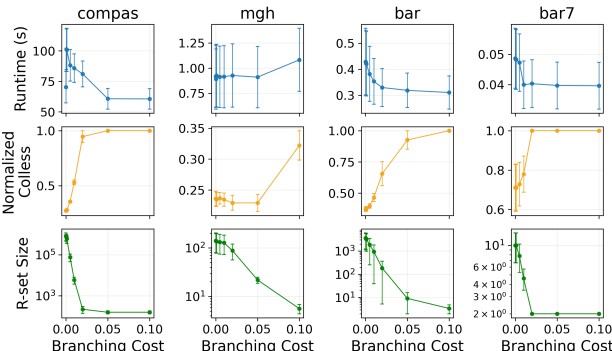

*Figure 4.* Average runtime, Normalized Colless Index (NCI), and Rashomon set size obtained by **GRAVITree** on four different datasets as the branching cost increases. Of particular interest is the NCI plot, which increases as the branching cost is increased. This is direct evidence of interpolation from rule lists to fully grown trees. Increasing the branching cost for a fixed budget also tends to reduce runtime.

a different optimization problem from **GRAVITree**; namely, finding trees that perform within $\varepsilon$ of the minimizer of $\frac{1}{n} \sum_{i=1}^{n} \mathbf{1}[y_i \neq T(x_i)] + \lambda \times |\text{num leaves}|$. This makes a direct comparison difficult. To make things fair, we choose modest sparsity and branching penalty values for **GRAVITree**, specifically $\mu = 0.01, \alpha = 0.05, d = 5, \varepsilon = 0.01$. For TreeFARMS, we set $\lambda = 0.005$. We provide a more detailed rationale behind these hyperparameter choices in Appendix E. From Table 2, we see that while **GRAVITree** has a higher overall loss compared to TreeFARMS, this is generally compensated for by lower decision sparsity and lower loss on the positive examples compared to TreeFARMS.

**GRAVITree vs Falling Rule Lists.** We also want to understand how the flexibility of falling trees compares with the interpretability of falling rule lists. We do this by comparing **GRAVITree** with $\mu = 0.01$ and $\mu = 1$. Because the latter penalty is high enough to strongly discourage branching, it effectively forces all models to be falling rule lists. From Table 1, we observe that the $\mu = 0.01$ configuration generally discovers models with lower test loss and lower false negative rates, reflecting the additional expressive power gained from allowing controlled branching. In contrast, the $\mu = 1$ configuration yields models with smaller Rashomon sets and substantially lower decision sparsity on the positive examples since rule lists consist of a single path with no internal branching. This yields interpretable sequential explanations but at the expense of predictive performance.

**Conclusion.** We introduced the model class of *falling trees*; these are a type of decision tree that is constrained to have descending leaf probabilities on leaves adjacent to any path from root to leaf. We also introduced a new optimization penalty for decision trees, the *branching cost*, which penalizes the number of branching nodes, rather than the traditional sparsity penalty in tree optimization (van der Linden et al., 2023; Demirović et al., 2022; Lin et al., 2020), necessitating the development of **GRAVITree**, a dynamic-programming-with-bounds algorithm. We presented experimental evidence that falling trees have significantly reduced average decision sparsity relative to sparsity-regularized decision trees, and better overall performance than falling rule lists, while maintaining similar sparsity. This model class is well suited to settings where practitioners need to evaluate sparse models by hand in a high risk environment.

| Dataset | Test Loss (↓) | | FNR (↓) | | Pos Sparsity $\beta(T, D^+)$ (↓) | |
|---|---|---|---|---|---|---|
| | GRAVITree | TreeFARMS | GRAVITree | TreeFARMS | GRAVITree | TreeFARMS |
| NIJ Recidivism | $0.305 \pm 0.003$ | $\underline{0.304 \pm 0.002}$ | $\mathbf{0.126 \pm 0.010}$ | $0.154 \pm 0.006$ | $2.79 \pm 0.04$ | $\mathbf{2.27 \pm 0.16}$ |
| Adult | $\mathbf{0.204 \pm 0.003}$ | $0.227 \pm 0.014$ | $0.206 \pm 0.008$ | $\mathbf{0.177 \pm 0.023}$ | $3.39 \pm 0.05$ | $\mathbf{2.94 \pm 0.09}$ |
| Aging | $0.385 \pm 0.020$ | $\mathbf{0.349 \pm 0.026}$ | $0.379 \pm 0.043$ | $\underline{0.349 \pm 0.037}$ | $\mathbf{3.83 \pm 0.11}$ | $4.25 \pm 0.19$ |
| Bar7 | $0.309 \pm 0.016$ | $\underline{0.291 \pm 0.008}$ | $0.523 \pm 0.067$ | $\underline{0.492 \pm 0.047}$ | $\mathbf{2.31 \pm 0.41}$ | $3.18 \pm 0.26$ |
| Bar | $0.299 \pm 0.016$ | $\underline{0.281 \pm 0.008}$ | $0.414 \pm 0.036$ | $\underline{0.390 \pm 0.019}$ | $\mathbf{2.82 \pm 0.07}$ | $3.66 \pm 0.03$ |
| BCW Bin | $0.070 \pm 0.014$ | $\underline{0.069 \pm 0.008}$ | $\underline{0.075 \pm 0.019}$ | $0.076 \pm 0.009$ | $\mathbf{2.96 \pm 0.31}$ | $3.07 \pm 0.24$ |
| Bike | $0.175 \pm 0.006$ | $\mathbf{0.148 \pm 0.006}$ | $\mathbf{0.084 \pm 0.015}$ | $0.095 \pm 0.009$ | $\mathbf{3.01 \pm 0.14}$ | $3.38 \pm 0.09$ |
| Car Evaluation | $0.141 \pm 0.017$ | $0.141 \pm 0.017$ | $0.000 \pm 0.000$ | $0.000 \pm 0.000$ | $2.09 \pm 0.20$ | $\underline{2.00 \pm 0.00}$ |
| Carryout Takeaway | $\mathbf{0.371 \pm 0.008}$ | $0.395 \pm 0.018$ | $\mathbf{0.333 \pm 0.004}$ | $0.390 \pm 0.028$ | $\underline{3.12 \pm 0.23}$ | $3.41 \pm 0.18$ |
| Coffee House | $0.300 \pm 0.010$ | $\underline{0.295 \pm 0.009}$ | $0.253 \pm 0.011$ | $\underline{0.248 \pm 0.013}$ | $\underline{3.39 \pm 0.09}$ | $3.47 \pm 0.06$ |
| Compas | $0.331 \pm 0.013$ | $\underline{0.330 \pm 0.014}$ | $0.379 \pm 0.017$ | $0.388 \pm 0.017$ | $2.98 \pm 0.05$ | $\mathbf{2.70 \pm 0.17}$ |
| FICO | $0.296 \pm 0.011$ | $0.296 \pm 0.010$ | $\underline{0.351 \pm 0.032}$ | $0.353 \pm 0.019$ | $2.92 \pm 0.25$ | $\mathbf{2.12 \pm 0.09}$ |
| Heart | $0.223 \pm 0.033$ | $\mathbf{0.203 \pm 0.032}$ | $\mathbf{0.198 \pm 0.031}$ | $0.237 \pm 0.039$ | $\mathbf{2.70 \pm 0.35}$ | $3.26 \pm 0.31$ |
| MGH | $0.061 \pm 0.006$ | $\mathbf{0.031 \pm 0.007}$ | $\underline{0.025 \pm 0.010}$ | $0.024 \pm 0.010$ | $\mathbf{2.01 \pm 0.01}$ | $3.05 \pm 0.04$ |
| Monks1 | $0.178 \pm 0.123$ | $0.170 \pm 0.113$ | $\underline{0.151 \pm 0.150}$ | $0.160 \pm 0.136$ | $2.81 \pm 0.57$ | $\underline{2.55 \pm 0.41}$ |
| Monks2 | $\underline{0.441 \pm 0.095}$ | $0.442 \pm 0.084$ | $0.615 \pm 0.054$ | $\underline{0.577 \pm 0.147}$ | $\mathbf{3.55 \pm 0.31}$ | $4.28 \pm 0.17$ |
| Monks3 | $0.096 \pm 0.100$ | $\underline{0.091 \pm 0.066}$ | $0.150 \pm 0.137$ | $\underline{0.140 \pm 0.113}$ | $3.52 \pm 0.32$ | $3.60 \pm 0.27$ |
| Mushroom | $0.030 \pm 0.002$ | $\mathbf{0.007 \pm 0.002}$ | $0.034 \pm 0.006$ | $\mathbf{0.014 \pm 0.004}$ | $\mathbf{2.51 \pm 0.02}$ | $3.63 \pm 0.01$ |
| Restaurant 20 | $\mathbf{0.298 \pm 0.012}$ | $0.316 \pm 0.008$ | $\mathbf{0.250 \pm 0.014}$ | $0.285 \pm 0.014$ | $\mathbf{1.65 \pm 0.23}$ | $2.31 \pm 0.49$ |
| Spambase | $0.126 \pm 0.006$ | $\mathbf{0.102 \pm 0.004}$ | $0.181 \pm 0.022$ | $\mathbf{0.154 \pm 0.032}$ | $\mathbf{2.43 \pm 0.05}$ | $3.33 \pm 0.28$ |
| Student | $\underline{0.192 \pm 0.051}$ | $0.201 \pm 0.043$ | $\underline{0.182 \pm 0.080}$ | $0.194 \pm 0.055$ | $\mathbf{3.75 \pm 0.29}$ | $4.40 \pm 0.39$ |
| Wine | $0.291 \pm 0.017$ | $\mathbf{0.277 \pm 0.008}$ | $\mathbf{0.169 \pm 0.017}$ | $0.203 \pm 0.021$ | $\mathbf{2.58 \pm 0.03}$ | $2.59 \pm 0.16$ |

*Table 2.* A comparison between **GRAVITree** and TreeFARMS across many datasets. We set $\mu = 0.01$, $\alpha = 0.05$ for **GRAVITree**, $\lambda = 0.005$ for TreeFARMS, and $\varepsilon = 0.01$ for both methods. **Bold** indicates a statistically significant improvement (Welch's $t$-test, $p < 0.05$); underlined values indicate the method is numerically better but not statistically significantly so. While TreeFARMS generally achieves lower overall loss than **GRAVITree** due to its unconstrained model class, **GRAVITree** is competitive on false negative rates and generally sparser on positive examples.

## Impact Statement

This paper presents work on interpretable machine learning, which is essential for many ethical AI applications.

## Acknowledgments

We acknowledge support from the National Institutes of Health/NIDA grant number R01DA054994. This material is based upon work supported by the National Science Foundation Graduate Research Fellowship Program under Grant No. DGE-2139754. Any opinions, findings, and conclusions or recommendations expressed in this material are those of the author(s) and do not necessarily reflect the views of the National Science Foundation. We acknowledge the support of the Natural Sciences and Engineering Research Council of Canada (NSERC). Nous remercions le Conseil de recherches en sciences naturelles et en génie du Canada (CRSNG) de son soutien.

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

## A. Proofs

**Theorem 4.1.** *Let $T$ be a decision tree. Let $v \in N(T)$ be an internal node of $T$ with children $v_1$ and $v_2$ such that $v_1$ is a leaf and $v_2$ is an internal node. Note $v_1$ and $v_2$ can be either the left or right child of $v$. Denote $p_1^+$ to be the empirical positive proportion of $v_1$, and let $p_{\max}^+(v_2)$ be the maximum positive proportion of any leaf in the subtree rooted at $v_2$.*

*$T$ satisfies the falling constraint if and only if, for all nodes $v \in N(T)$ where one child is a leaf and the other is an internal node, $p_1^+ \geq p_{\max}^+(v_2)$.*

*Proof.* Assume for contradiction that the condition holds, but $T$ does not satisfy the falling constraint. Then, by definition, there must exist a path $P$ in $T$ from root to leaf with leaves $\ell_1, \ell_2$ adjacent to $P$, where $\ell_1$ is higher on the path than $\ell_2$ and $p_{\ell_1}^+ < p_{\ell_2}^+$.

Denote $h_1$ to be the parent node of $\ell_1$. Since $\ell_1$ is a leaf and it is higher on the path than $\ell_2$, $h_1$ must be an internal node and $\ell_1$'s sibling must be an internal node (denoted $s_1$). $\ell_2$ must be a leaf in the subtree rooted at $s_1$. Let $p_{max}^+$ be the maximum empirical positive proportion of any leaf in the subtree rooted at $s_1$. Then $p_{max}^+ \geq p_{\ell_2}^+ > p_{\ell_1}^+$, which contradicts the assumption that $T$ satisfies the condition in the theorem. We conclude that $T$ must satisfy the falling constraint.

For the other direction, assume that $T$ is falling and that there exists an internal node $v \in N(T)$ with children $v_1$ and $v_2$ such that $v_1$ is a leaf and $v_2$ is an internal node. Assume for contradiction that $p_1^+ < p_{max}^+(v_2)$. Denote the leaf at which the maximum probability below $v_2$ is achieved as $v_{max}$. Then, on the path from the root to $v_{max}$, the leaf probabilities are not monotonically decreasing and $T$ cannot be falling. This is a contradiction, so we must have $p_1^+ \geq p_{max}^+(v_2)$. $\square$

**Theorem 4.2.** *Suppose that we run Algorithm 1, and we are solving a subproblem corresponding to path $P$ consisting of a sequence of feature splits $P = \{(j_1, \gamma_1), (j_2, \gamma_2), \ldots, (j_P, \gamma_P)\}$, where $1 \leq j_k \leq K$ and $\gamma_k \in \{0,1\}$ indicates that $P$ splits on the rule $j_k = \gamma_k$. Let $p_{recent}^+$ denote the empirical positive proportion of the deepest leaf adjacent to $P$ (this leaf is the optimal solution to its corresponding subproblem). Let $j \in \{1, \ldots, K\}$ be a candidate split feature, and let $p_L^+, p_R^+$ denote the empirical positive proportions of the left $(j = 0)$ and right $(j = 1)$ children created by splitting on $j$. Then, if $\max(p_L^+, p_R^+) > p_{recent}^+$, any tree $T_P$ grown from $P$ via a split on $j$ will violate the falling constraint, and $j$ may be eliminated as a candidate.*

*Proof.* Assume WLOG $p_R^+ \geq p_L^+$ and $p_R^+ > p_{recent}^+$. Let $I_R$ denote the indices of points in $X$ reaching the right child of the split on $j$ (i.e., satisfying all splits in $P$ together with $j = 1$), and let $T_R$ be an arbitrary tree grown from this child. Assume for contradiction that $T_R$ does not violate the falling constraint. Consider the leaves of $T_R$, $\ell_1, \ldots, \ell_L$, each containing $n_{\ell_z}$ samples and with positive proportion $p_{\ell_z}^+$. Since the leaf with proportion $p_{recent}^+$ is adjacent to $P$, it is also adjacent to every root-to-leaf path in $T_R$. Thus, we must have $p_{\ell_z}^+ \leq p_{recent}^+, \forall 1 \leq z \leq L$ (where $L$ is the number of leaves in $T_R$), since $T_R$ does not violate the falling constraint. Then

$$
\begin{aligned}
p_R^+ &= \frac{1}{|I_R|} \sum_{i \in I_R} y_i \\
&= \frac{1}{|I_R|} \sum_{z=1}^{L} n_{\ell_z} \cdot p_{\ell_z}^+ \\
&= \sum_{z=1}^{L} \frac{n_{\ell_z}}{|I_R|} p_{\ell_z}^+ \\
&\leq \max_z p_{\ell_z}^+ \qquad\qquad\qquad \text{convex combination bounded by maximum} \\
&\leq p_{recent}^+ \qquad\qquad\qquad\qquad \text{since } T_R \text{ is falling.}
\end{aligned}
$$

This contradicts the assumption that $p_R^+ > p_{recent}^+$, and we conclude that if $\max(p_L^+, p_R^+) > p_{recent}^+$, any tree resulting from a split on feature $j$ will violate the falling constraint. $\square$

## B. Case Study on ICU Data

The falling trees in our paper were found in the Rashomon set of falling trees on the MIMIC-III dataset (Johnson et al., 2016). The features and outcomes were processed using the preprocessing pipeline developed by Zhu et al. (2025) in their work developing an interpretable risk score for critical care patients. The data is heavily imbalanced in its initial state – there are 3148 positive examples and 27090 negative examples. To find nontrivial models that do not always predict negative, we balanced the data to a 1:1 ratio by downsampling the negative class and keeping the positive examples. After rebalancing the data, we used the GOSDT+Guesses method presented by McTavish et al. (2022) to binarize the continuous features in the data. We randomly split the balanced and binarized data into a training, validation, and test set along a 70/10/20 train/val/test split.

In Figure 5, we present several models found in the Rashomon set of falling trees, computed by **GRAVITree**, with different numbers of branches. Each of these trees has the highest validation accuracy (using a threshold for classification of $0.5$) among trees with the same number of branches in the Rashomon set. The tree with three branches has a test accuracy of $0.69$, the tree with two branches has a test accuracy of $0.71$, the tree with one branch has a test accuracy $0.69$, and the tree with zero branches has a test accuracy of $0.66$. The test accuracy of an XGBoost (Chen, 2016) model trained on the same data as GraviTree was $0.73$. The test accuracy of an XGBoost model trained on the balanced continuous data, before binarization with GOSDT+Guesses, was $0.78$; determining how to binarize while maintaining accuracy comparable to the continuous data is a subject of future work. These performances are not directly comparable to those presented by Zhu et al. (2025) because of the balancing operation we performed on the data.

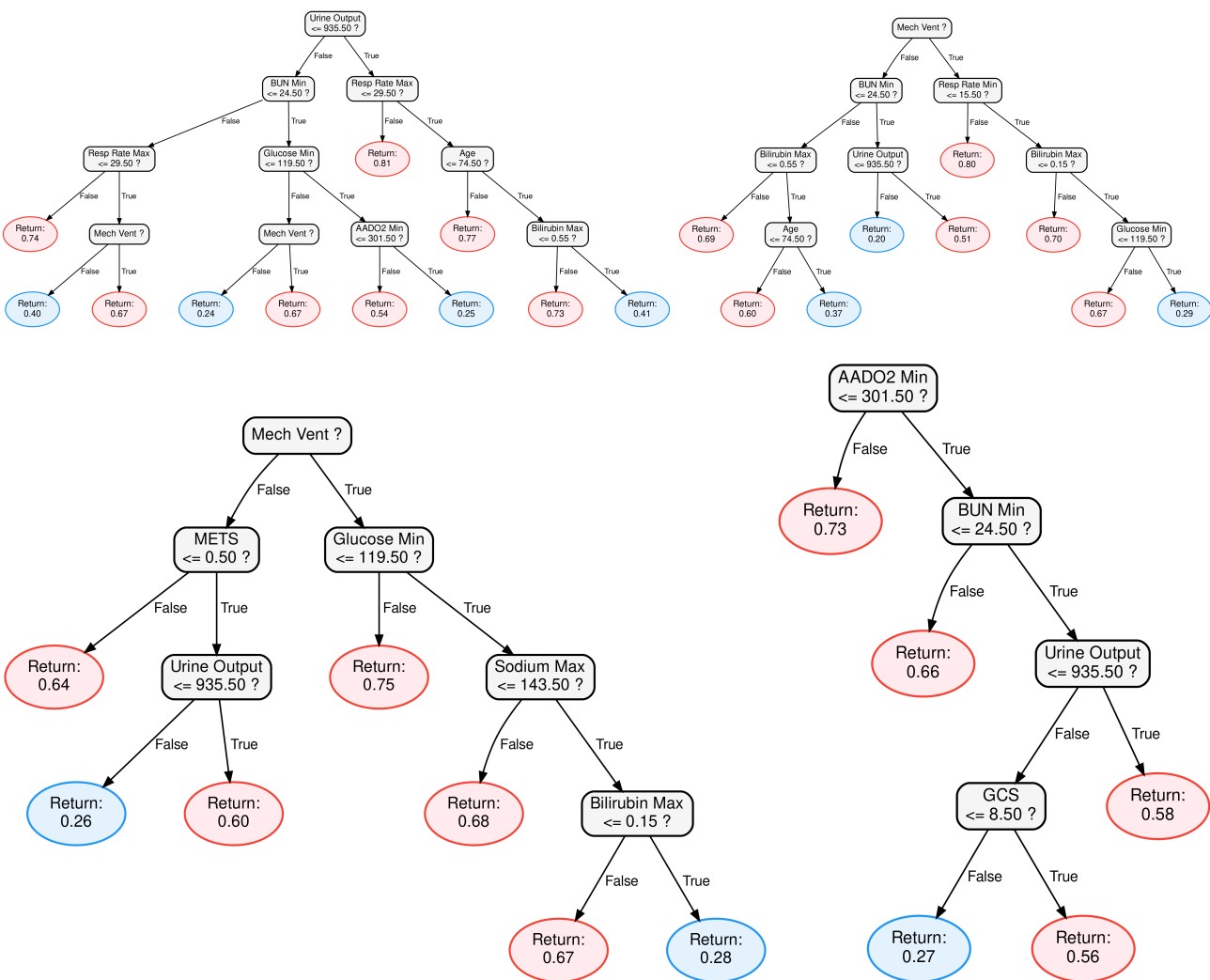

*Figure 5.* Four falling trees in the Rashomon set on the MIMIC dataset. The top left tree has three branches, top right has two branches, bottom left has one branch, and bottom right zero branches. Leaf nodes with blue outlines represent positive classifications greater than the 0.5 threshold for the classification decision boundary. The trees are all in the Rashomon set with a depth of 4, Rashomon parameter 0.05, branching cost of 0.01, and minimum support of 0.04.

# C. Algorithms

## C.1. The merge and filter subroutine of GRAVITree

In this section, we describe the subroutine responsible for combining the Rashomon sets of the left and right subproblems associated with a candidate split on feature $j$. This operation is nontrivial; although each subtree individually satisfies both the Rashomon loss bound and the falling constraint, not all pairs of left and right models can be combined without violating these global conditions.

The subroutine constructs candidate internal-node models by taking the Cartesian product of the Rashomon sets from the left and right subtrees. Each candidate pair is evaluated by summing the corresponding subtree losses and adding a branching penalty when both subtrees are internal. Candidates whose total loss (+ the branching cost $\mu$, if neither are leaves) exceeds the current budget $B$ are discarded (line 10). To improve efficiency, both Rashomon sets are sorted by loss, allowing early termination when even the minimum-loss pairing violates the budget (lines 2-5).

In addition to enforcing the objective bound, the procedure filters out combinations that violate the falling constraint. Specifically, if one child subtree is a leaf and the other is internal, the leaf must not have a smaller positive-class proportion than the purest leaf in the internal subtree (lines 13-19). This ensures that the maximum positive-class probability along any root-to-leaf path remains non-increasing, preserving the falling structure of the resulting model.

The surviving combinations are assembled into new internal-node models splitting on feature $j$ and added to the merged Rashomon set (lines 20-21). This subroutine is a key component of **GRAVITree**, enabling efficient exploration of valid model combinations while aggressively pruning infeasible or suboptimal candidates.

---

**Algorithm 4** MERGEANDFILTERRSET

---

**Require:** $\mathcal{R}_L, \mathcal{R}_R$, Input feature $j$, Budget $B$, branching cost $\mu$
1: $\mathcal{R} \leftarrow \emptyset$
2: Sort $\mathcal{R}_L, \mathcal{R}_R$ according to increasing loss
3: $\mathcal{L}_R^{\min} \leftarrow \min\{\mathcal{L}_R : (T_R, \mathcal{L}_R, p_R^+) \in \mathcal{R}_R\}$ {Minimum loss of the right Rashomon set}
4: **for** each $(T_L, \mathcal{L}_L, p_L^+) \in \mathcal{R}_L$ **do**
5:     **if** $\mathcal{L}_L + \mathcal{L}_R^{\min} > B$ **then**
6:         **break**
7:     **end if**{Even using the minimum loss of the right Rset breaks the budget}
8:     **for** each $(T_R, \mathcal{L}_R, p_R^+) \in \mathcal{R}_R$ **do**
9:         $\mathcal{L} \leftarrow \mathcal{L}_L + \mathcal{L}_R + \mu \cdot \mathbf{1}[T_L, T_R \text{ internal}]$
10:        **if** $\mathcal{L} > B$ **then**
11:           **break**
12:        **end if**
13:        $p_{\max}^+ \leftarrow \max(p_L^+, p_R^+)$ {Overall purest leaf (in terms of positive class proportion) seen so far}
14:        **if** $T_L$ leaf, $T_R$ internal, $p_L^+ < p_R^+$ **then**
15:           **continue**
16:        **end if**
17:        **if** $T_L$ internal, $T_R$ leaf, $p_R^+ < p_L^+$ **then**
18:           **continue**
19:        **end if**
20:        $T \leftarrow \text{NODE}(j, T_L, T_R)$ {Split on feature $j$, with $T_L$ and $T_R$ as the new child subtrees in the Rashomon set}
21:        add $(T, \mathcal{L}, p_{\max}^+)$ to $\mathcal{R}$
22:     **end for**
23: **end for**
24: **return** $\mathcal{R}$

---

## C.2. Falling Rule Lists Baseline: An extension to the algorithm of Chen & Rudin (2018)

In this section, we describe an approximate algorithm for finding the Rashomon set of falling rule lists. This is intended to be a simple extension to the algorithm of Chen & Rudin (2018), which performs Monte Carlo Tree Search to find the optimal falling rule list. We provide this as a baseline for comparison with our method, since there are no other existing methods to compute the Rashomon set of falling rule lists.

Given training data $D$, positive class weight $w$, regularization parameter $\lambda$, and tolerance $\varepsilon$, our baseline algorithm proceeds in two phases. First, we use the algorithm of Chen & Rudin (2018) to construct a near-optimal falling rule list $f_{\text{ref}}$ with

objective $L(f_{\text{ref}}, D, w, \lambda)$ over $T_1$ iterations. This sets the baseline objective to compare with in the second phase. In the second phase, we run the algorithm for $T_2$ iterations to construct an approximate Rashomon set of FRLs. In each iteration, we start with an empty rule list prefix and sequentially sample antecedents. When we sample an antecedent, we add it to the prefix if its improvement in accuracy compensates the increase in complexity. After adding the antecedent, we check if it is possible to improve on the new prefix by adding any rule. The precise condition for this check can be found in Theorem 4.6 of Chen & Rudin (2018). Once the prefix satisfies this stopping condition, the final FRL $f$ is constructed by appending the default rule to the prefix. We add it to the approximate Rashomon set if it satisfies the reference Rashomon bound $L(f, D, w, \lambda) \leq L(f_{\text{ref}}, w, D, \lambda) + \varepsilon$. See Algorithm 5 for a formal description of this algorithm (which we call FRAME).

Note that FRAME can work with antecedents (features) of length 1 (e.g., age $> 25$) or length 2 (e.g., age $> 25 \wedge$ sex$=$ female as binary features). We experiment with both below.

---

**Algorithm 5** FRAME - Falling Rule List Approximate Model Enumerator (A simple extension of Chen & Rudin, 2018)

---

**Require:** Dataset $D$, mined antecedents $A$, positive weight $w$, regularization term $\lambda$, Rashomon set threshold $\varepsilon$, first pass iterations $T_1$, second pass iterations $T_2$, probability of early stopping $p$

1: Run FRL algorithm of Chen & Rudin (2018) for $T_1$ iterations to obtain reference model $f^*$ and objective $L(f^*, D, w, \lambda)$
2: Initialize an empty FRLTrie object {Data structure for storing FRLs}
3: **for** $t = 1$ to $T_2$ **do**
4:     Initialize prefix $e \leftarrow []$, $\alpha \leftarrow 1$
    {Check if the prefix satisfies Theorem 4.6 in Chen & Rudin (2018)}
5:     **while** it is possible to improve $e$'s regularized objective by adding a rule **do**
6:         $p' \leftarrow \text{random}(0, 1)$; if $p' > p$, **Break** {Allow early stopping}
7:         Initialize candidate set $S_t \leftarrow \emptyset$
        {Find all feasible antecedents to extend the current prefix}
8:         **for** each antecedent $a \in A$ **do**
9:             $D_{\text{uncov}} \leftarrow \{(\mathbf{x}, y) \in D : a'(\mathbf{x}) = 0 \; \forall a' \in e\}$ {Points remaining after current prefix}
10:            $e_a \leftarrow \text{append}(e, a)$ {Prefix extended by adding $a$}
11:            $D_a \leftarrow \{(\mathbf{x}, y) \in D_{\text{uncov}} : a(\mathbf{x}) = 1\}$ {Uncaptured points now captured by $a$}
12:            $\hat{p}^{e_a}_{|e|+1} \leftarrow \frac{|\{(\mathbf{x},y) \in D_a : y=1\}|}{|D_{\text{uncov}}|}$ {Positive proportion among newly captured points} {Satisfies falling constraint and improves error}
13:            **if** $\frac{1}{1+w} < \hat{p}^{e_a}_{|e|+1} \leq \hat{p}^{e_a}_{|e|}$ {Compute lower bound from Theorem 4.6 of Chen & Rudin (2018)} **then**
14:                $L_{\text{lb}}(e_a, D, w, \lambda) \leftarrow$ lower bound on the objective of prefix $e_a$
15:                **if** $L_{\text{lb}}(e_a, D, w, \lambda) < L(f^*, D, w, \lambda) + \varepsilon$ {Rashomon bound} **then**
16:                    Add $a$ to candidate set $S_t$
17:                **end if**
18:            **end if**
19:         **end for**
20:         **if** $S_t \neq \emptyset$ **then**
21:            $a \sim \text{softmax}(\{C(e, a, D, t) \; \forall \, a \in S_t\})$ {Sample next antecedent}
22:            $e \leftarrow \text{append}(e, a)$ {Append the antecedent to the prefix}
23:         **else**
24:            **Break**
25:         **end if**
26:     **end while**
27:     $f \leftarrow \text{append}(e, a_\emptyset)$ {Add the default rule to the prefix}
28:     Compute $L(f, D, w, \lambda)$ {Rashomon bound}
29:     **if** $L(f, D, w, \lambda) < L(f^*, D, w, \lambda) + \varepsilon$ **then**
30:         Insert $f$ into FRLTrie {Add to Rashomon set $\mathcal{R}_t$}
31:     **end if**
32: **end for**
33: **return** FRLTrie

---

For the experiments in Tables 3 and 4, we set the Rashomon tolerance $\varepsilon = 0.02$. However, the Rashomon sets are defined

with respect to different objectives (Chen & Rudin, 2018, also penalizes the number of leaves in the rule list) making direct comparisons difficult. To make the methods somewhat comparable, we set the same minimum support $\alpha = 0.05$ and have the same depth budget of 5 for both methods. We set the leaf penalty $\lambda = 0$ for FRAME, making it equivalent to $\mu = 1$ for **GRAVITree** since the latter does not have an explicit leaf penalty.

| Dataset | GraviTree | | | | FRAME (max len 1) | | | |
|---|---|---|---|---|---|---|---|---|
| | RSet | Test Loss | Pos Decision Sparsity $\beta(T, D^+)$ | FNR | RSet | Test Loss | Pos Decision Sparsity $\beta(T, D^+)$ | FNR |
| Adult | $2.0 \pm 0.0$ | $\mathbf{0.322 \pm 0.008}$ | $\mathbf{1.65 \pm 0.10}$ | $0.104 \pm 0.009$ | $18.6 \pm 3.6$ | $0.332 \pm 0.009$ | $2.21 \pm 0.07$ | $\mathbf{0.074 \pm 0.005}$ |
| Aging | $13.8 \pm 5.3$ | $0.383 \pm 0.035$ | $\mathbf{2.29 \pm 0.30}$ | $0.343 \pm 0.058$ | $35.0 \pm 9.9$ | $\mathbf{0.369 \pm 0.030}$ | $2.35 \pm 0.22$ | $0.315 \pm 0.049$ |
| Bar | $2.0 \pm 0.0$ | $\mathbf{0.311 \pm 0.006}$ | $1.91 \pm 0.05$ | $0.556 \pm 0.017$ | $1.0 \pm 0.0$ | $0.311 \pm 0.006$ | $\mathbf{1.86 \pm 0.02}$ | $0.556 \pm 0.017$ |
| Bar7 | $2.0 \pm 0.0$ | $\mathbf{0.311 \pm 0.006}$ | $2.02 \pm 0.05$ | $0.556 \pm 0.017$ | $1.4 \pm 0.2$ | $0.314 \pm 0.006$ | $\mathbf{2.01 \pm 0.09}$ | $0.549 \pm 0.017$ |
| BCW | $1.0 \pm 0.0$ | $0.081 \pm 0.009$ | $2.95 \pm 0.06$ | $0.221 \pm 0.025$ | $1.0 \pm 0.0$ | $0.081 \pm 0.009$ | $\mathbf{2.88 \pm 0.07}$ | $0.221 \pm 0.025$ |
| Bike | $14.8 \pm 2.4$ | $0.319 \pm 0.005$ | $\mathbf{1.11 \pm 0.05}$ | $0.188 \pm 0.018$ | $19.8 \pm 3.7$ | $\mathbf{0.316 \pm 0.006}$ | $1.23 \pm 0.02$ | $0.146 \pm 0.015$ |
| Car Evaluation | $20.6 \pm 3.4$ | $\mathbf{0.411 \pm 0.010}$ | $\mathbf{1.40 \pm 0.11}$ | $0.302 \pm 0.019$ | $18.6 \pm 3.3$ | $0.416 \pm 0.009$ | $1.47 \pm 0.10$ | $0.259 \pm 0.014$ |
| Carryout Takeaway | $4.4 \pm 2.4$ | $0.397 \pm 0.034$ | $\mathbf{1.62 \pm 0.08}$ | $0.369 \pm 0.057$ | $6.2 \pm 1.9$ | $\mathbf{0.382 \pm 0.013}$ | $1.92 \pm 0.11$ | $0.334 \pm 0.017$ |
| Coffee House | $1.4 \pm 0.2$ | $\mathbf{0.335 \pm 0.005}$ | $2.59 \pm 0.09$ | $0.363 \pm 0.004$ | $1.4 \pm 0.2$ | $0.337 \pm 0.005$ | $\mathbf{2.57 \pm 0.09}$ | $0.341 \pm 0.015$ |
| COMPAS | $147 \pm 44$ | $0.350 \pm 0.006$ | $2.81 \pm 0.05$ | $0.361 \pm 0.010$ | $83.0 \pm 10.6$ | $0.351 \pm 0.005$ | $\mathbf{2.64 \pm 0.02}$ | $0.349 \pm 0.020$ |
| FICO | $11.0 \pm 3.0$ | $\mathbf{0.306 \pm 0.004}$ | $\mathbf{1.41 \pm 0.07}$ | $0.370 \pm 0.007$ | $30.8 \pm 2.3$ | $0.307 \pm 0.004$ | $1.83 \pm 0.04$ | $0.356 \pm 0.006$ |
| Heart | $2.4 \pm 0.6$ | $0.269 \pm 0.005$ | $2.01 \pm 0.19$ | $0.223 \pm 0.024$ | $10.8 \pm 1.9$ | $\mathbf{0.263 \pm 0.009}$ | $\mathbf{1.80 \pm 0.04}$ | $0.197 \pm 0.022$ |
| MGH | $7.2 \pm 0.2$ | $\mathbf{0.514 \pm 0.037}$ | $\mathbf{0.89 \pm 0.07}$ | $0.145 \pm 0.062$ | $6.2 \pm 0.2$ | $0.517 \pm 0.019$ | $0.94 \pm 0.04$ | $0.105 \pm 0.034$ |
| Monks1 | $2.4 \pm 0.2$ | $\mathbf{0.272 \pm 0.048}$ | $1.24 \pm 0.18$ | $0.434 \pm 0.050$ | $1.2 \pm 0.2$ | $0.276 \pm 0.046$ | $\mathbf{1.15 \pm 0.11}$ | $0.431 \pm 0.052$ |
| Monks2 | $2.6 \pm 0.2$ | $\mathbf{0.394 \pm 0.008}$ | $\mathbf{0.20 \pm 0.08}$ | $0.979 \pm 0.021$ | $1.4 \pm 0.2$ | $0.400 \pm 0.012$ | $0.20 \pm 0.12$ | $0.969 \pm 0.031$ |
| Monks3 | $31.0 \pm 3.9$ | $\mathbf{0.359 \pm 0.011}$ | $1.52 \pm 0.04$ | $0.395 \pm 0.040$ | $22.0 \pm 2.1$ | $0.359 \pm 0.013$ | $\mathbf{1.49 \pm 0.03}$ | $0.374 \pm 0.040$ |
| Mushroom | $1.0 \pm 0.0$ | $\mathbf{0.053 \pm 0.002}$ | $1.43 \pm 0.01$ | $0.030 \pm 0.002$ | $1.8 \pm 0.2$ | $0.053 \pm 0.002$ | $\mathbf{1.43 \pm 0.01}$ | $0.030 \pm 0.002$ |
| NIJ Recidivism | $3.0 \pm 0.4$ | $\mathbf{0.310 \pm 0.004}$ | $\mathbf{2.02 \pm 0.06}$ | $\mathbf{0.190 \pm 0.005}$ | $16.8 \pm 1.3$ | $0.311 \pm 0.004$ | $2.19 \pm 0.03$ | $0.193 \pm 0.005$ |
| Restaurant 20 | $2.0 \pm 0.0$ | $\mathbf{0.298 \pm 0.005}$ | $\mathbf{1.48 \pm 0.06}$ | $0.249 \pm 0.005$ | $1.8 \pm 0.2$ | $0.298 \pm 0.005$ | $1.54 \pm 0.03$ | $0.249 \pm 0.005$ |
| Spambase | $4.4 \pm 0.7$ | $\mathbf{0.125 \pm 0.004}$ | $2.30 \pm 0.07$ | $0.148 \pm 0.008$ | $5.2 \pm 0.8$ | $0.126 \pm 0.004$ | $\mathbf{2.29 \pm 0.08}$ | $0.145 \pm 0.006$ |
| Student | $18.6 \pm 9.5$ | $0.257 \pm 0.036$ | $\mathbf{1.48 \pm 0.12}$ | $0.158 \pm 0.054$ | $25.4 \pm 11.0$ | $\mathbf{0.252 \pm 0.022}$ | $1.65 \pm 0.06$ | $0.136 \pm 0.032$ |
| Wine | $43.2 \pm 11.1$ | $\mathbf{0.322 \pm 0.007}$ | $2.70 \pm 0.04$ | $0.160 \pm 0.013$ | $37.6 \pm 8.5$ | $0.327 \pm 0.007$ | $\mathbf{2.08 \pm 0.06}$ | $0.149 \pm 0.011$ |

*Table 3.* Comparison between Rashomon sets output by **GRAVITree** with rule list mode ($\mu = 1$) and FRAME (max len 1). For **GRAVITree**, we set the min support $\alpha = 0.05$. For FRAME, we set the min support of a mined rule (i.e. the min proportion of points that need to satisfy the rule) to be 0.05. Bold indicates one method outperforms the other (lower is better).

| Dataset | GraviTree | | | | FRAME (max len 2) | | | |
|---|---|---|---|---|---|---|---|---|
| | RSet | Test Loss | Pos Decision Sparsity $\beta(T, D^+)$ | FNR | RSet | Test Loss | Pos Decision Sparsity $\beta(T, D^+)$ | FNR |
| Adult | $2.0 \pm 0.0$ | $0.322 \pm 0.008$ | $1.65 \pm 0.10$ | $0.104 \pm 0.009$ | $794 \pm 83.0$ | $0.322 \pm 0.005$ | $4.25 \pm 0.04$ | $\mathbf{0.093 \pm 0.002}$ |
| Aging | $13.8 \pm 5.3$ | $0.383 \pm 0.035$ | $\mathbf{2.29 \pm 0.30}$ | $0.343 \pm 0.058$ | $1351 \pm 206.2$ | $\mathbf{0.318 \pm 0.012}$ | $4.87 \pm 0.42$ | $0.236 \pm 0.021$ |
| Bar | $2.0 \pm 0.0$ | $0.311 \pm 0.006$ | $\mathbf{1.91 \pm 0.05}$ | $0.556 \pm 0.017$ | $416 \pm 29.0$ | $\mathbf{0.310 \pm 0.006}$ | $6.07 \pm 0.18$ | $0.401 \pm 0.012$ |
| Bar7 | $2.0 \pm 0.0$ | $\mathbf{0.311 \pm 0.006}$ | $2.02 \pm 0.05$ | $0.556 \pm 0.017$ | $71.6 \pm 19.8$ | $0.313 \pm 0.006$ | $3.95 \pm 0.24$ | $0.546 \pm 0.018$ |
| BCW | $1.0 \pm 0.0$ | $0.081 \pm 0.009$ | $2.95 \pm 0.06$ | $0.221 \pm 0.025$ | $743 \pm 31.9$ | $0.082 \pm 0.006$ | $4.48 \pm 0.10$ | $\mathbf{0.068 \pm 0.005}$ |
| Bike | $14.8 \pm 2.4$ | $0.319 \pm 0.005$ | $\mathbf{1.11 \pm 0.05}$ | $0.188 \pm 0.018$ | $1228 \pm 71.7$ | $\mathbf{0.286 \pm 0.003}$ | $2.95 \pm 0.05$ | $0.080 \pm 0.002$ |
| Car Evaluation | $20.6 \pm 3.4$ | $\mathbf{0.411 \pm 0.010}$ | $\mathbf{1.40 \pm 0.11}$ | $0.302 \pm 0.019$ | $457 \pm 82.7$ | $0.425 \pm 0.004$ | $3.60 \pm 0.12$ | $0.119 \pm 0.005$ |
| Carryout Takeaway | $4.4 \pm 2.4$ | $0.397 \pm 0.034$ | $\mathbf{1.62 \pm 0.08}$ | $0.369 \pm 0.057$ | $324 \pm 40.9$ | $\mathbf{0.350 \pm 0.010}$ | $4.37 \pm 0.18$ | $0.270 \pm 0.015$ |
| Coffee House | $1.4 \pm 0.2$ | $\mathbf{0.335 \pm 0.005}$ | $2.59 \pm 0.09$ | $0.363 \pm 0.004$ | $142 \pm 33.0$ | $0.344 \pm 0.006$ | $5.96 \pm 0.13$ | $0.320 \pm 0.017$ |
| COMPAS | $147 \pm 44.4$ | $0.350 \pm 0.006$ | $2.81 \pm 0.05$ | $0.361 \pm 0.010$ | $2159 \pm 127.7$ | $\mathbf{0.345 \pm 0.004}$ | $5.31 \pm 0.03$ | $0.317 \pm 0.004$ |
| Heart | $2.4 \pm 0.6$ | $0.269 \pm 0.005$ | $2.01 \pm 0.19$ | $0.223 \pm 0.024$ | $1821 \pm 160.7$ | $\mathbf{0.241 \pm 0.006}$ | $4.22 \pm 0.09$ | $0.153 \pm 0.009$ |
| FICO | $29.4 \pm 27.2$ | $0.316 \pm 0.005$ | $1.97 \pm 0.13$ | $0.109 \pm 0.011$ | $1516 \pm 74.9$ | $\mathbf{0.310 \pm 0.005}$ | $3.84 \pm 0.04$ | $0.094 \pm 0.004$ |
| MGH | $7.2 \pm 0.2$ | $0.514 \pm 0.037$ | $\mathbf{0.89 \pm 0.07}$ | $0.145 \pm 0.062$ | $60.6 \pm 8.2$ | $\mathbf{0.402 \pm 0.023}$ | $2.52 \pm 0.10$ | $0.061 \pm 0.011$ |
| Monks1 | $2.4 \pm 0.2$ | $\mathbf{0.272 \pm 0.048}$ | $\mathbf{1.24 \pm 0.18}$ | $0.434 \pm 0.050$ | $230 \pm 46.6$ | $0.294 \pm 0.036$ | $4.70 \pm 0.22$ | $0.287 \pm 0.036$ |
| Monks2 | $2.6 \pm 0.2$ | $\mathbf{0.394 \pm 0.008}$ | $\mathbf{0.20 \pm 0.08}$ | $0.979 \pm 0.021$ | $15.0 \pm 3.0$ | $0.434 \pm 0.013$ | $2.59 \pm 0.18$ | $0.796 \pm 0.061$ |
| Monks3 | $31.0 \pm 3.9$ | $\mathbf{0.359 \pm 0.011}$ | $1.52 \pm 0.04$ | $0.395 \pm 0.040$ | $409 \pm 133.4$ | $0.363 \pm 0.023$ | $3.78 \pm 0.20$ | $0.306 \pm 0.080$ |
| Mushroom | $1.0 \pm 0.0$ | $0.053 \pm 0.002$ | $1.43 \pm 0.01$ | $0.030 \pm 0.002$ | $587 \pm 89.1$ | $\mathbf{0.047 \pm 0.002}$ | $3.91 \pm 0.04$ | $0.028 \pm 0.002$ |
| NIJ Recidivism | $3.0 \pm 0.4$ | $\mathbf{0.310 \pm 0.004}$ | $\mathbf{2.02 \pm 0.06}$ | $0.190 \pm 0.005$ | $1015 \pm 57.9$ | $0.311 \pm 0.003$ | $4.33 \pm 0.02$ | $0.121 \pm 0.004$ |
| Restaurant 20 | $2.0 \pm 0.0$ | $\mathbf{0.298 \pm 0.005}$ | $\mathbf{1.48 \pm 0.06}$ | $\mathbf{0.249 \pm 0.005}$ | $246 \pm 45.1$ | $0.308 \pm 0.010$ | $3.58 \pm 0.04$ | $0.269 \pm 0.016$ |
| Spambase | $4.4 \pm 0.7$ | $0.125 \pm 0.004$ | $2.30 \pm 0.07$ | $\mathbf{0.148 \pm 0.008}$ | $531 \pm 180.0$ | $\mathbf{0.123 \pm 0.008}$ | $4.36 \pm 0.12$ | $0.152 \pm 0.010$ |
| Student | $18.6 \pm 9.5$ | $0.257 \pm 0.036$ | $\mathbf{1.48 \pm 0.12}$ | $0.158 \pm 0.054$ | $1106 \pm 450.9$ | $\mathbf{0.237 \pm 0.022}$ | $3.85 \pm 0.20$ | $0.132 \pm 0.014$ |
| Wine | $43.2 \pm 11.1$ | $\mathbf{0.322 \pm 0.007}$ | $2.70 \pm 0.04$ | $0.160 \pm 0.027$ | $1210 \pm 163.4$ | $0.336 \pm 0.013$ | $4.03 \pm 0.08$ | $0.137 \pm 0.018$ |

*Table 4.* Comparison between Rashomon sets output by **GRAVITree** with rule list mode ($\mu = 1$) and FRAME (max len 2). For **GRAVITree**, we set the min support $\alpha = 0.05$. For FRAME, we set the min support of a mined rule (i.e. the min proportion of points that need to satisfy the rule) to be 0.05. Bold indicates one method outperforms the other (lower is better).

From Table 3, we can see that **GRAVITree** generally has better overall test loss compared to FRAME. This is encouraging since FRAME is intended to be an approximate method while **GRAVITree** is exact. However, **GRAVITree** sometimes has worse decision sparsity and / or test loss.

From Table 4, we can see that **GRAVITree** surprisingly has test loss comparable to FRAME with max len 2, but no method uniformly dominates the other. **GRAVITree** also shines in terms of decision sparsity on positive examples, but this is compensated by a larger test loss on positive examples (FNR).

## D. Ablation Study on Various Components of GRAVITree

### D.1. GRAVITree with and without the falling constraint

Our next experiment exposes how a Rashomon set of falling trees differs from a regular Rashomon set. We compared **GRAVITree** with and without the falling constraint enabled. More precisely, this amounts to disabling the pruning conditions in lines 12 and lines $29-33$ in Algorithm 3, lines $18-20, 34-36$ in Algorithm 1.

We set the branching penalty $\mu = 0.01$ to enable modest branching capabilities and set $\alpha = 0.05, d = 5, \varepsilon = 0.01$. To enable a fair comparison, we also gave the unconstrained version the same overall Rashomon bound of $(1 + \varepsilon)L^*_{\text{falling}}$. Table 5 shows the results in this setting. We see that **GRAVITree** with the falling constraint generally has a much smaller Rashomon set size. While it obtains test losses competitive with the unconstrained set, both the decision sparsity on the positive examples $\beta(T, D^+)$ and the false negative rate are noticeably lower for the constrained version. This is crucial in cost-sensitive classification settings, where we might be willing to trade off the FPR for the FNR. With **GRAVITree**, we can integrate this cost-sensitivity directly into the tree structure whilst also enjoying the benefits of a faster runtime compared to unconstrained Rashomon set search, as in Table 6.

| | GRAVITree | | | | GRAVITree (No Falling Constraint) | | | |
|---|---|---|---|---|---|---|---|---|
| Dataset | RSet Size | Test Loss | Sparsity $\beta(T, D^+)$ | FNR | RSet Size | Test Loss | Sparsity $\beta(T, D^+)$ | FNR |
| Adult | $1.0 \pm 0.0$ | $\mathbf{0.204 \pm 0.003}$ | $3.85 \pm 0.21$ | $0.206 \pm 0.007$ | $167 \pm 49$ | $0.212 \pm 0.004$ | $\mathbf{3.40 \pm 0.03}$ | $\mathbf{0.204 \pm 0.006}$ |
| Aging | $57.0 \pm 80.8$ | $\mathbf{0.385 \pm 0.018}$ | $3.83 \pm 0.09$ | $\mathbf{0.379 \pm 0.038}$ | $2622 \pm 2300$ | $0.405 \pm 0.024$ | $\mathbf{3.83 \pm 0.07}$ | $0.408 \pm 0.037$ |
| Bar | $8.8 \pm 2.6$ | $0.299 \pm 0.014$ | $\mathbf{2.97 \pm 0.05}$ | $\mathbf{0.414 \pm 0.033}$ | $270 \pm 64$ | $\mathbf{0.298 \pm 0.015}$ | $3.64 \pm 0.07$ | $0.432 \pm 0.026$ |
| Bar7 | $4.6 \pm 2.2$ | $0.309 \pm 0.015$ | $\mathbf{2.46 \pm 0.44}$ | $\mathbf{0.523 \pm 0.060}$ | $92.4 \pm 61.4$ | $0.306 \pm 0.014$ | $3.48 \pm 0.21$ | $0.539 \pm 0.064$ |
| BCW | $1.0 \pm 0.0$ | $\mathbf{0.070 \pm 0.012}$ | $4.07 \pm 0.37$ | $0.075 \pm 0.017$ | $184 \pm 83$ | $0.072 \pm 0.014$ | $\mathbf{2.74 \pm 0.15}$ | $\mathbf{0.067 \pm 0.008}$ |
| Car Evaluation | $1.0 \pm 0.0$ | $0.141 \pm 0.015$ | $\mathbf{2.09 \pm 0.19}$ | $\mathbf{0.000 \pm 0.000}$ | $17.6 \pm 12.4$ | $\mathbf{0.136 \pm 0.010}$ | $2.33 \pm 0.18$ | $0.022 \pm 0.041$ |
| Carryout Takeaway | $12.4 \pm 8.4$ | $\mathbf{0.371 \pm 0.007}$ | $3.14 \pm 0.23$ | $\mathbf{0.333 \pm 0.004}$ | $1469 \pm 679$ | $0.401 \pm 0.012$ | $3.59 \pm 0.06$ | $0.402 \pm 0.016$ |
| Coffee House | $77.2 \pm 89.2$ | $0.300 \pm 0.009$ | $\mathbf{3.57 \pm 0.07}$ | $0.253 \pm 0.010$ | $1216 \pm 1875$ | $\mathbf{0.299 \pm 0.010}$ | $3.67 \pm 0.06$ | $0.262 \pm 0.013$ |
| COMPAS | $1248 \pm 861$ | $0.331 \pm 0.012$ | $\mathbf{2.99 \pm 0.05}$ | $\mathbf{0.379 \pm 0.015}$ | $7.4e5 \pm 6.3e5$ | $0.331 \pm 0.012$ | $3.55 \pm 0.03$ | $0.399 \pm 0.014$ |
| Heart | $1.0 \pm 0.0$ | $\mathbf{0.190 \pm 0.025}$ | $3.70 \pm 0.25$ | $\mathbf{0.186 \pm 0.027}$ | $1286 \pm 897$ | $0.212 \pm 0.021$ | $\mathbf{3.35 \pm 0.15}$ | $0.233 \pm 0.027$ |
| FICO | $154 \pm 145$ | $\mathbf{0.296 \pm 0.010}$ | $3.12 \pm 0.20$ | $\mathbf{0.351 \pm 0.029}$ | $8.1e5 \pm 3.4e5$ | $0.299 \pm 0.009$ | $3.19 \pm 0.09$ | $0.361 \pm 0.017$ |
| MGH | $2.6 \pm 0.5$ | $0.061 \pm 0.005$ | $\mathbf{2.04 \pm 0.03}$ | $0.025 \pm 0.009$ | $6.2 \pm 1.5$ | $\mathbf{0.058 \pm 0.006}$ | $2.04 \pm 0.23$ | $0.034 \pm 0.011$ |
| Monks1 | $12.0 \pm 6.2$ | $\mathbf{0.178 \pm 0.110}$ | $3.16 \pm 0.32$ | $\mathbf{0.151 \pm 0.134}$ | $2642 \pm 3295$ | $0.193 \pm 0.101$ | $3.36 \pm 0.17$ | $0.167 \pm 0.123$ |
| Monks2 | $1.0 \pm 0.0$ | $0.441 \pm 0.085$ | $\mathbf{3.51 \pm 0.22}$ | $0.615 \pm 0.049$ | $3327 \pm 2084$ | $\mathbf{0.411 \pm 0.083}$ | $3.64 \pm 0.16$ | $0.642 \pm 0.104$ |
| Monks3 | $1.0 \pm 0.0$ | $\mathbf{0.096 \pm 0.090}$ | $3.73 \pm 0.29$ | $\mathbf{0.150 \pm 0.122}$ | $124 \pm 48$ | $0.101 \pm 0.081$ | $\mathbf{3.56 \pm 0.16}$ | $0.153 \pm 0.116$ |
| Mushroom | $1.0 \pm 0.0$ | $\mathbf{0.030 \pm 0.002}$ | $2.44 \pm 0.01$ | $\mathbf{0.034 \pm 0.005}$ | $24.6 \pm 3.2$ | $0.031 \pm 0.002$ | $\mathbf{2.25 \pm 0.08}$ | $0.035 \pm 0.005$ |
| NIJ Recidivism | $227 \pm 27$ | $0.305 \pm 0.003$ | $\mathbf{2.91 \pm 0.04}$ | $\mathbf{0.126 \pm 0.009}$ | $1.2e4 \pm 2.4e3$ | $\mathbf{0.304 \pm 0.002}$ | $3.19 \pm 0.02$ | $0.177 \pm 0.005$ |
| Restaurant 20 | $3.4 \pm 1.4$ | $\mathbf{0.298 \pm 0.010}$ | $\mathbf{1.71 \pm 0.20}$ | $\mathbf{0.250 \pm 0.012}$ | $949 \pm 668$ | $0.323 \pm 0.006$ | $3.12 \pm 0.28$ | $0.303 \pm 0.021$ |
| Spambase | $1.0 \pm 0.0$ | $\mathbf{0.105 \pm 0.011}$ | $3.54 \pm 0.08$ | $0.189 \pm 0.028$ | $1.5e4 \pm 6.2e3$ | $0.112 \pm 0.005$ | $\mathbf{3.49 \pm 0.08}$ | $\mathbf{0.180 \pm 0.018}$ |
| Student | $1.0 \pm 0.0$ | $\mathbf{0.192 \pm 0.046}$ | $3.64 \pm 0.23$ | $\mathbf{0.182 \pm 0.072}$ | $1003 \pm 1027$ | $0.229 \pm 0.022$ | $3.70 \pm 0.08$ | $0.218 \pm 0.039$ |
| Wine | $530 \pm 520$ | $\mathbf{0.286 \pm 0.016}$ | $3.39 \pm 0.08$ | $\mathbf{0.188 \pm 0.021}$ | $1.1e6 \pm 1.2e6$ | $0.296 \pm 0.015$ | $3.68 \pm 0.06$ | $0.196 \pm 0.015$ |

*Table 5.* Properties of the Rashomon set with vs without falling constraint under the objective function in Equation 1. Parameters: $\mu = 0.01, \alpha = 0.05, d = 5, \varepsilon = 0.01$

### D.2. Quantized Caching Ablation: Relative Rashomon Bound

While our algorithms are exact, there is one subtlety in the way we cache subproblems that merits exploration. Specifically, the falling constraint can reduce the effectiveness of caching, since for a given subproblem, we also need to know the positive proportion of the last leaf adjacent to the current path. Thus, we typically have to store the tuple corresponding to $(s, d, p^+_{recent})$ where $s$ is the subproblem (aka the bitvector corresponding to the indices of the points that are in the current subproblem), $d$ is the depth budget, and $p^+_{recent}$ is the most recent probability observed on the current path. Because the latter is a floating point between $[0, 1]$, it is unlikely that we can effectively cache this tuple for retrieval when similar subproblems are encountered. To mitigate this, we experiment with a caching strategy based on quantizing $p^+_{recent}$. Specifically, we define a quantization factor $q \in [0, 1]$ (where a value closer to 1 indicates greater quantization). We then cache $\left(s, d, \left\lceil \frac{p^+_{recent}}{q} \right\rceil\right)$ and retrieve this when we reach a subproblem with the same $s$ and $d$, and the same quantized $p^+_{recent}$.

In this experiment, for each value of $q$, we train a separate optimal tree using Algorithm 1. The Rashomon bound for each $q$ is therefore $(1 + \varepsilon)\times$ loss of best tree with that quantization.

| Dataset | GRAVITree (s) | GRAVITree (No Falling Constraint) (s) |
|---|---|---|
| Adult | **15.034 ± 3.211** | 17.019 ± 3.683 |
| Aging | **1.727 ± 1.366** | 2.138 ± 1.590 |
| Bar | **0.238 ± 0.060** | 0.265 ± 0.066 |
| Bar7 | **0.040 ± 0.015** | 0.047 ± 0.019 |
| BCW | **0.028 ± 0.005** | 0.036 ± 0.007 |
| Bike | **62.989 ± 4.911** | 111.129 ± 9.216 |
| Car Evaluation | **0.064 ± 0.016** | 0.078 ± 0.020 |
| Carryout Takeaway | **0.277 ± 0.132** | 0.394 ± 0.206 |
| Coffee House | **0.695 ± 0.226** | 0.804 ± 0.291 |
| COMPAS | **51.030 ± 16.142** | 77.292 ± 34.417 |
| FICO | **2302.470 ± 950.036** | 3784.807 ± 1407.666 |
| Heart | **1.065 ± 0.633** | 1.353 ± 0.905 |
| MGH | **0.766 ± 0.517** | 1.050 ± 0.715 |
| Monks1 | **0.015 ± 0.006** | 0.093 ± 0.092 |
| Monks2 | **0.017 ± 0.007** | 0.129 ± 0.076 |
| Monks3 | **0.005 ± 0.003** | 0.008 ± 0.004 |
| Mushroom | **0.037 ± 0.006** | 0.044 ± 0.008 |
| NIJ Recidivism | **7.375 ± 0.706** | 8.759 ± 0.876 |
| Restaurant 20 | **0.515 ± 0.152** | 0.840 ± 0.290 |
| Spambase | **59.422 ± 3.401** | 75.041 ± 5.286 |
| Student | **1.538 ± 0.945** | 2.302 ± 1.492 |
| Wine | **251.600 ± 83.421** | 343.263 ± 130.793 |

*Table 6.* Comparison of runtime (seconds) for $\mu = 0.01$, $\alpha = 0.05$ with vs without the falling constraint enabled.

| Dataset | Config | Runtime (s) | R-set size | Avg. loss | Avg. sparsity |
|---|---|---|---|---|---|
| NIJ Recidivism | QC=0.001 | 6.91 ± 0.57 | 227.4 ± 26.9 | **0.305 ± 0.003** | 2.91 ± 0.04 |
|  | QC=0.01 | 5.87 ± 0.45 | 227.4 ± 26.9 | **0.305 ± 0.003** | 2.91 ± 0.04 |
|  | QC=0.1 | 4.46 ± 0.33 | 225.8 ± 24.9 | **0.305 ± 0.003** | 2.91 ± 0.04 |
|  | QC=1.0 | **4.14 ± 0.47** | 225.6 ± 26.8 | **0.305 ± 0.003** | **2.92 ± 0.05** |
| Adult | QC=0.001 | 15.31 ± 3.36 | 1.0 ± 0.0 | **0.204 ± 0.003** | **3.85 ± 0.21** |
|  | QC=0.01 | 13.24 ± 3.01 | 1.0 ± 0.0 | **0.204 ± 0.003** | **3.85 ± 0.21** |
|  | QC=0.1 | 10.86 ± 2.52 | 1.0 ± 0.0 | **0.204 ± 0.003** | **3.85 ± 0.21** |
|  | QC=1.0 | **6.51 ± 1.43** | 1.0 ± 0.0 | **0.204 ± 0.003** | **3.85 ± 0.21** |
| Aging | QC=0.001 | 1.83 ± 1.50 | 57.0 ± 80.8 | **0.385 ± 0.018** | **3.83 ± 0.09** |
|  | QC=0.01 | 1.73 ± 1.42 | 57.0 ± 80.8 | **0.385 ± 0.018** | **3.83 ± 0.09** |
|  | QC=0.1 | 1.28 ± 0.87 | 65.8 ± 81.3 | 0.386 ± 0.018 | 3.81 ± 0.10 |
|  | QC=1.0 | **0.72 ± 0.43** | 65.8 ± 84.0 | 0.386 ± 0.016 | 3.81 ± 0.08 |
| Bar7 | QC=0.001 | 0.043 ± 0.017 | 4.6 ± 2.2 | **0.309 ± 0.015** | **2.46 ± 0.44** |
|  | QC=0.01 | 0.039 ± 0.015 | 4.6 ± 2.2 | **0.309 ± 0.015** | **2.46 ± 0.44** |
|  | QC=0.1 | 0.031 ± 0.012 | 4.6 ± 2.2 | **0.309 ± 0.015** | **2.46 ± 0.44** |
|  | QC=1.0 | **0.021 ± 0.009** | 4.2 ± 1.9 | **0.309 ± 0.015** | 2.42 ± 0.41 |
| Bar | QC=0.001 | 0.23 ± 0.06 | 8.8 ± 2.6 | **0.299 ± 0.014** | **2.97 ± 0.05** |
|  | QC=0.01 | 0.23 ± 0.06 | 8.8 ± 2.6 | **0.299 ± 0.014** | **2.97 ± 0.05** |
|  | QC=0.1 | 0.18 ± 0.05 | 8.8 ± 2.6 | **0.299 ± 0.014** | **2.97 ± 0.05** |
|  | QC=1.0 | **0.11 ± 0.03** | 8.8 ± 2.6 | **0.299 ± 0.014** | 2.96 ± 0.04 |
| BCW | QC=0.001 | 0.027 ± 0.005 | 1.0 ± 0.0 | **0.070 ± 0.012** | **4.07 ± 0.37** |
|  | QC=0.01 | 0.029 ± 0.007 | 1.0 ± 0.0 | **0.070 ± 0.012** | **4.07 ± 0.37** |
|  | QC=0.1 | 0.023 ± 0.004 | 1.0 ± 0.0 | **0.070 ± 0.012** | **4.07 ± 0.37** |
|  | QC=1.0 | **0.021 ± 0.004** | 1.0 ± 0.0 | **0.070 ± 0.012** | **4.07 ± 0.37** |
| Bike | QC=0.001 | 64.28 ± 5.56 | 1.0 ± 0.0 | **0.152 ± 0.004** | **3.21 ± 0.12** |
|  | QC=0.01 | 63.08 ± 5.43 | 1.0 ± 0.0 | **0.152 ± 0.004** | **3.21 ± 0.12** |
|  | QC=0.1 | 60.25 ± 5.47 | 1.0 ± 0.0 | **0.152 ± 0.004** | **3.21 ± 0.12** |
|  | QC=1.0 | **47.92 ± 4.62** | 1.0 ± 0.0 | **0.152 ± 0.004** | **3.21 ± 0.12** |
| Car Evaluation | QC=0.001 | 0.063 ± 0.017 | 1.0 ± 0.0 | **0.141 ± 0.015** | **2.09 ± 0.19** |
|  | QC=0.01 | 0.062 ± 0.017 | 1.0 ± 0.0 | **0.141 ± 0.015** | **2.09 ± 0.19** |
|  | QC=0.1 | 0.055 ± 0.015 | 1.0 ± 0.0 | **0.141 ± 0.015** | **2.09 ± 0.19** |
|  | QC=1.0 | **0.038 ± 0.009** | 1.0 ± 0.0 | **0.141 ± 0.015** | **2.09 ± 0.19** |

*Continued on next page*

| Dataset | Config | Runtime (s) | Rset size | Test Loss | Test decision sparsity |
|---|---|---|---|---|---|
| Carryout Takeaway | QC=0.001 | $0.28 \pm 0.14$ | $12.4 \pm 8.4$ | $\mathbf{0.371 \pm 0.007}$ | $\mathbf{3.14 \pm 0.23}$ |
| | QC=0.01 | $0.26 \pm 0.12$ | $12.4 \pm 8.4$ | $\mathbf{0.371 \pm 0.007}$ | $\mathbf{3.14 \pm 0.23}$ |
| | QC=0.1 | $0.21 \pm 0.09$ | $12.4 \pm 8.4$ | $\mathbf{0.371 \pm 0.007}$ | $\mathbf{3.14 \pm 0.23}$ |
| | QC=1.0 | $\mathbf{0.16 \pm 0.07}$ | $12.4 \pm 8.4$ | $\mathbf{0.371 \pm 0.007}$ | $\mathbf{3.14 \pm 0.23}$ |
| Coffee House | QC=0.001 | $0.70 \pm 0.22$ | $77.2 \pm 89.2$ | $\mathbf{0.300 \pm 0.009}$ | $\mathbf{3.57 \pm 0.07}$ |
| | QC=0.01 | $0.66 \pm 0.21$ | $77.2 \pm 89.2$ | $\mathbf{0.300 \pm 0.009}$ | $\mathbf{3.57 \pm 0.07}$ |
| | QC=0.1 | $0.49 \pm 0.15$ | $77.2 \pm 89.2$ | $\mathbf{0.300 \pm 0.009}$ | $\mathbf{3.57 \pm 0.07}$ |
| | QC=1.0 | $\mathbf{0.27 \pm 0.09}$ | $77.2 \pm 89.2$ | $\mathbf{0.300 \pm 0.009}$ | $\mathbf{3.57 \pm 0.07}$ |
| COMPAS | QC=0.001 | $54.91 \pm 19.18$ | $1248.4 \pm 860.9$ | $\mathbf{0.331 \pm 0.012}$ | $\mathbf{2.99 \pm 0.05}$ |
| | QC=0.01 | $50.85 \pm 18.07$ | $1248.4 \pm 860.9$ | $\mathbf{0.331 \pm 0.012}$ | $\mathbf{2.99 \pm 0.05}$ |
| | QC=0.1 | $41.99 \pm 15.03$ | $1230.2 \pm 866.3$ | $\mathbf{0.331 \pm 0.012}$ | $2.98 \pm 0.05$ |
| | QC=1.0 | $\mathbf{33.54 \pm 12.97}$ | $1227.2 \pm 861.4$ | $\mathbf{0.331 \pm 0.012}$ | $2.98 \pm 0.05$ |
| Heart | QC=0.001 | $0.98 \pm 0.60$ | $1.0 \pm 0.0$ | $\mathbf{0.190 \pm 0.025}$ | $\mathbf{3.70 \pm 0.25}$ |
| | QC=0.01 | $0.95 \pm 0.59$ | $1.0 \pm 0.0$ | $\mathbf{0.190 \pm 0.025}$ | $\mathbf{3.70 \pm 0.25}$ |
| | QC=0.1 | $0.77 \pm 0.46$ | $1.0 \pm 0.0$ | $\mathbf{0.190 \pm 0.025}$ | $\mathbf{3.70 \pm 0.25}$ |
| | QC=1.0 | $\mathbf{0.53 \pm 0.32}$ | $1.0 \pm 0.0$ | $\mathbf{0.190 \pm 0.025}$ | $\mathbf{3.70 \pm 0.25}$ |
| MGH | QC=0.001 | $0.60 \pm 0.42$ | $2.6 \pm 0.5$ | $\mathbf{0.061 \pm 0.005}$ | $\mathbf{2.04 \pm 0.03}$ |
| | QC=0.01 | $0.57 \pm 0.40$ | $2.6 \pm 0.5$ | $\mathbf{0.061 \pm 0.005}$ | $\mathbf{2.04 \pm 0.03}$ |
| | QC=0.1 | $0.45 \pm 0.30$ | $2.6 \pm 0.5$ | $\mathbf{0.061 \pm 0.005}$ | $\mathbf{2.04 \pm 0.03}$ |
| | QC=1.0 | $\mathbf{0.26 \pm 0.17}$ | $2.6 \pm 0.5$ | $\mathbf{0.061 \pm 0.005}$ | $\mathbf{2.04 \pm 0.03}$ |
| Monks1 | QC=0.001 | $0.012 \pm 0.005$ | $12.0 \pm 6.2$ | $\mathbf{0.178 \pm 0.110}$ | $\mathbf{3.16 \pm 0.32}$ |
| | QC=0.01 | $0.012 \pm 0.005$ | $12.0 \pm 6.2$ | $\mathbf{0.178 \pm 0.110}$ | $\mathbf{3.16 \pm 0.32}$ |
| | QC=0.1 | $0.012 \pm 0.005$ | $12.0 \pm 6.2$ | $\mathbf{0.178 \pm 0.110}$ | $\mathbf{3.16 \pm 0.32}$ |
| | QC=1.0 | $\mathbf{0.010 \pm 0.004}$ | $10.2 \pm 5.0$ | $\mathbf{0.178 \pm 0.110}$ | $3.13 \pm 0.32$ |
| Monks2 | QC=0.001 | $0.014 \pm 0.006$ | $1.0 \pm 0.0$ | $0.441 \pm 0.085$ | $3.51 \pm 0.22$ |
| | QC=0.01 | $0.014 \pm 0.006$ | $1.0 \pm 0.0$ | $0.441 \pm 0.085$ | $3.51 \pm 0.22$ |
| | QC=0.1 | $0.013 \pm 0.006$ | $1.0 \pm 0.0$ | $0.441 \pm 0.085$ | $3.51 \pm 0.22$ |
| | QC=1.0 | $\mathbf{0.012 \pm 0.005}$ | $1.0 \pm 0.0$ | $\mathbf{0.429 \pm 0.066}$ | $\mathbf{3.57 \pm 0.35}$ |
| Monks3 | QC=0.001 | $0.004 \pm 0.003$ | $1.0 \pm 0.0$ | $\mathbf{0.096 \pm 0.090}$ | $\mathbf{3.73 \pm 0.29}$ |
| | QC=0.01 | $0.005 \pm 0.003$ | $1.0 \pm 0.0$ | $\mathbf{0.096 \pm 0.090}$ | $\mathbf{3.73 \pm 0.29}$ |
| | QC=0.1 | $0.004 \pm 0.003$ | $1.0 \pm 0.0$ | $\mathbf{0.096 \pm 0.090}$ | $\mathbf{3.73 \pm 0.29}$ |
| | QC=1.0 | $\mathbf{0.003 \pm 0.002}$ | $1.4 \pm 0.8$ | $0.128 \pm 0.093$ | $3.50 \pm 0.52$ |
| Mushroom | QC=0.001 | $0.037 \pm 0.006$ | $1.0 \pm 0.0$ | $\mathbf{0.030 \pm 0.002}$ | $\mathbf{2.44 \pm 0.01}$ |
| | QC=0.01 | $0.037 \pm 0.006$ | $1.0 \pm 0.0$ | $\mathbf{0.030 \pm 0.002}$ | $\mathbf{2.44 \pm 0.01}$ |
| | QC=0.1 | $0.035 \pm 0.006$ | $1.0 \pm 0.0$ | $\mathbf{0.030 \pm 0.002}$ | $\mathbf{2.44 \pm 0.01}$ |
| | QC=1.0 | $\mathbf{0.024 \pm 0.004}$ | $1.0 \pm 0.0$ | $\mathbf{0.030 \pm 0.002}$ | $\mathbf{2.44 \pm 0.01}$ |
| Restaurant 20 | QC=0.001 | $0.49 \pm 0.17$ | $3.4 \pm 1.4$ | $\mathbf{0.298 \pm 0.010}$ | $\mathbf{1.71 \pm 0.20}$ |
| | QC=0.01 | $0.46 \pm 0.16$ | $3.4 \pm 1.4$ | $\mathbf{0.298 \pm 0.010}$ | $\mathbf{1.71 \pm 0.20}$ |
| | QC=0.1 | $0.39 \pm 0.15$ | $3.4 \pm 1.4$ | $\mathbf{0.298 \pm 0.010}$ | $\mathbf{1.71 \pm 0.20}$ |
| | QC=1.0 | $\mathbf{0.31 \pm 0.12}$ | $3.4 \pm 1.4$ | $\mathbf{0.298 \pm 0.010}$ | $\mathbf{1.71 \pm 0.20}$ |
| Spambase | QC=0.001 | $56.41 \pm 3.26$ | $1.0 \pm 0.0$ | $\mathbf{0.105 \pm 0.011}$ | $3.54 \pm 0.08$ |
| | QC=0.01 | $52.39 \pm 3.03$ | $1.0 \pm 0.0$ | $\mathbf{0.105 \pm 0.011}$ | $3.54 \pm 0.08$ |
| | QC=0.1 | $36.46 \pm 2.13$ | $1.0 \pm 0.0$ | $\mathbf{0.105 \pm 0.011}$ | $3.54 \pm 0.08$ |
| | QC=1.0 | $\mathbf{20.41 \pm 2.09}$ | $1.0 \pm 0.0$ | $0.112 \pm 0.006$ | $\mathbf{3.62 \pm 0.06}$ |
| Student | QC=0.001 | $1.36 \pm 0.85$ | $1.0 \pm 0.0$ | $\mathbf{0.192 \pm 0.046}$ | $\mathbf{3.64 \pm 0.23}$ |
| | QC=0.01 | $1.32 \pm 0.83$ | $1.0 \pm 0.0$ | $\mathbf{0.192 \pm 0.046}$ | $\mathbf{3.64 \pm 0.23}$ |
| | QC=0.1 | $1.08 \pm 0.66$ | $1.0 \pm 0.0$ | $\mathbf{0.192 \pm 0.046}$ | $\mathbf{3.64 \pm 0.23}$ |
| | QC=1.0 | $\mathbf{0.65 \pm 0.40}$ | $1.0 \pm 0.0$ | $\mathbf{0.192 \pm 0.046}$ | $\mathbf{3.64 \pm 0.23}$ |
| Wine | QC=0.001 | $263.23 \pm 108.02$ | $530.4 \pm 520.1$ | $\mathbf{0.286 \pm 0.016}$ | $\mathbf{3.39 \pm 0.08}$ |
| | QC=0.01 | $242.62 \pm 98.18$ | $530.4 \pm 520.1$ | $\mathbf{0.286 \pm 0.016}$ | $\mathbf{3.39 \pm 0.08}$ |
| | QC=0.1 | $199.48 \pm 81.09$ | $530.4 \pm 520.1$ | $\mathbf{0.286 \pm 0.016}$ | $\mathbf{3.39 \pm 0.08}$ |
| | QC=1.0 | $\mathbf{130.14 \pm 48.85}$ | $536.2 \pm 515.3$ | $\mathbf{0.286 \pm 0.016}$ | $\mathbf{3.39 \pm 0.08}$ |

*Table 7.* Ablation on different quantizations for the cache given $\mu = 0.01$. Values are mean $\pm$ std over 5 train-test trials.

In this regime, quantized caching affects the Rashomon set size in two opposing ways.

- It can result in solving the global optimal tree problem (Alg 2) to suboptimality. Since the Rashomon bound depends on the training objective of the optimal tree, quantized caching results in a larger Rashomon bound.

- It can result in child subproblems in the search graph induced by Alg 1 being solved to suboptimality. This can result in trees being left out of the Rashomon set, because the remaining budget decreases (e.g. lines 19, 26 in Alg 5). The practical effect on the Rashomon set size depends on the relative strength of these forces.

This is the reason why the Rashomon set size is sometimes inconsistent as $q$ increases (e.g. in it goes down in COMPAS but goes up in Wine).

### D.3. Quantized Caching Ablation: Absolute Rashomon Bound

In this experiment, we first train an optimal falling tree using Algorithm 1 with no quantization. All runs of **GRAVITree** with different values of $q$ are provided the same Rashomon corresponding to $(1 + \varepsilon)\times$ loss of best tree with no quantized caching.

| Dataset | Config | Runtime (s) | R-set size | Avg. loss | Avg. sparsity |
|---|---|---|---|---|---|
| NIJ Recidivism | QC=0.001 | $7.04 \pm 0.60$ | $227.4 \pm 26.9$ | $\mathbf{0.305 \pm 0.003}$ | $2.91 \pm 0.04$ |
| | QC=0.01 | $5.99 \pm 0.46$ | $227.4 \pm 26.9$ | $\mathbf{0.305 \pm 0.003}$ | $2.91 \pm 0.04$ |
| | QC=0.1 | $4.56 \pm 0.34$ | $225.8 \pm 24.9$ | $\mathbf{0.305 \pm 0.003}$ | $2.91 \pm 0.04$ |
| | QC=1.0 | $\mathbf{4.24 \pm 0.49}$ | $225.6 \pm 26.8$ | $\mathbf{0.305 \pm 0.003}$ | $\mathbf{2.92 \pm 0.05}$ |
| Adult | QC=0.001 | $15.03 \pm 3.45$ | $1.0 \pm 0.0$ | $\mathbf{0.204 \pm 0.003}$ | $\mathbf{3.85 \pm 0.21}$ |
| | QC=0.01 | $12.98 \pm 3.04$ | $1.0 \pm 0.0$ | $\mathbf{0.204 \pm 0.003}$ | $\mathbf{3.85 \pm 0.21}$ |
| | QC=0.1 | $10.67 \pm 2.57$ | $1.0 \pm 0.0$ | $\mathbf{0.204 \pm 0.003}$ | $\mathbf{3.85 \pm 0.21}$ |
| | QC=1.0 | $\mathbf{6.41 \pm 1.45}$ | $1.0 \pm 0.0$ | $\mathbf{0.204 \pm 0.003}$ | $\mathbf{3.85 \pm 0.21}$ |
| Aging | QC=0.001 | $1.69 \pm 1.37$ | $57.0 \pm 80.8$ | $0.385 \pm 0.018$ | $\mathbf{3.83 \pm 0.09}$ |
| | QC=0.01 | $1.60 \pm 1.32$ | $57.0 \pm 80.8$ | $0.385 \pm 0.018$ | $\mathbf{3.83 \pm 0.09}$ |
| | QC=0.1 | $1.14 \pm 0.84$ | $49.0 \pm 84.2$ | $0.381 \pm 0.014$ | $3.77 \pm 0.13$ |
| | QC=1.0 | $\mathbf{0.66 \pm 0.41}$ | $41.0 \pm 68.3$ | $\mathbf{0.375 \pm 0.017}$ | $3.81 \pm 0.09$ |
| Bar7 | QC=0.001 | $0.039 \pm 0.015$ | $4.6 \pm 2.2$ | $\mathbf{0.309 \pm 0.015}$ | $\mathbf{2.46 \pm 0.44}$ |
| | QC=0.01 | $0.036 \pm 0.014$ | $4.6 \pm 2.2$ | $\mathbf{0.309 \pm 0.015}$ | $\mathbf{2.46 \pm 0.44}$ |
| | QC=0.1 | $0.028 \pm 0.011$ | $4.6 \pm 2.2$ | $\mathbf{0.309 \pm 0.015}$ | $\mathbf{2.46 \pm 0.44}$ |
| | QC=1.0 | $\mathbf{0.019 \pm 0.008}$ | $4.2 \pm 1.9$ | $\mathbf{0.309 \pm 0.015}$ | $2.42 \pm 0.41$ |
| Bar | QC=0.001 | $0.24 \pm 0.06$ | $8.8 \pm 2.6$ | $\mathbf{0.299 \pm 0.014}$ | $\mathbf{2.97 \pm 0.05}$ |
| | QC=0.01 | $0.23 \pm 0.06$ | $8.8 \pm 2.6$ | $\mathbf{0.299 \pm 0.014}$ | $\mathbf{2.97 \pm 0.05}$ |
| | QC=0.1 | $0.18 \pm 0.05$ | $8.8 \pm 2.6$ | $\mathbf{0.299 \pm 0.014}$ | $\mathbf{2.97 \pm 0.05}$ |
| | QC=1.0 | $\mathbf{0.11 \pm 0.03}$ | $8.8 \pm 2.6$ | $\mathbf{0.299 \pm 0.014}$ | $2.96 \pm 0.04$ |
| BCW | QC=0.001 | $0.028 \pm 0.005$ | $1.0 \pm 0.0$ | $\mathbf{0.070 \pm 0.012}$ | $\mathbf{4.07 \pm 0.37}$ |
| | QC=0.01 | $0.027 \pm 0.005$ | $1.0 \pm 0.0$ | $\mathbf{0.070 \pm 0.012}$ | $\mathbf{4.07 \pm 0.37}$ |
| | QC=0.1 | $0.024 \pm 0.005$ | $1.0 \pm 0.0$ | $\mathbf{0.070 \pm 0.012}$ | $\mathbf{4.07 \pm 0.37}$ |
| | QC=1.0 | $\mathbf{0.021 \pm 0.004}$ | $1.0 \pm 0.0$ | $\mathbf{0.070 \pm 0.012}$ | $\mathbf{4.07 \pm 0.37}$ |
| Bike | QC=0.001 | $64.07 \pm 6.85$ | $1.0 \pm 0.0$ | $\mathbf{0.152 \pm 0.004}$ | $\mathbf{3.21 \pm 0.12}$ |
| | QC=0.01 | $63.36 \pm 6.80$ | $1.0 \pm 0.0$ | $\mathbf{0.152 \pm 0.004}$ | $\mathbf{3.21 \pm 0.12}$ |
| | QC=0.1 | $60.58 \pm 6.74$ | $1.0 \pm 0.0$ | $\mathbf{0.152 \pm 0.004}$ | $\mathbf{3.21 \pm 0.12}$ |
| | QC=1.0 | $\mathbf{47.78 \pm 4.87}$ | $1.0 \pm 0.0$ | $\mathbf{0.152 \pm 0.004}$ | $\mathbf{3.21 \pm 0.12}$ |
| Car Evaluation | QC=0.001 | $0.063 \pm 0.017$ | $1.0 \pm 0.0$ | $\mathbf{0.141 \pm 0.015}$ | $\mathbf{2.09 \pm 0.19}$ |
| | QC=0.01 | $0.061 \pm 0.017$ | $1.0 \pm 0.0$ | $\mathbf{0.141 \pm 0.015}$ | $\mathbf{2.09 \pm 0.19}$ |
| | QC=0.1 | $0.054 \pm 0.014$ | $1.0 \pm 0.0$ | $\mathbf{0.141 \pm 0.015}$ | $\mathbf{2.09 \pm 0.19}$ |
| | QC=1.0 | $\mathbf{0.037 \pm 0.009}$ | $1.0 \pm 0.0$ | $\mathbf{0.141 \pm 0.015}$ | $\mathbf{2.09 \pm 0.19}$ |
| Carryout Takeaway | QC=0.001 | $0.27 \pm 0.14$ | $12.4 \pm 8.4$ | $\mathbf{0.371 \pm 0.007}$ | $\mathbf{3.14 \pm 0.23}$ |
| | QC=0.01 | $0.25 \pm 0.13$ | $12.4 \pm 8.4$ | $\mathbf{0.371 \pm 0.007}$ | $\mathbf{3.14 \pm 0.23}$ |
| | QC=0.1 | $0.21 \pm 0.09$ | $12.4 \pm 8.4$ | $\mathbf{0.371 \pm 0.007}$ | $\mathbf{3.14 \pm 0.23}$ |

*Continued on next page*

| Dataset | Config | Runtime (s) | R-set size | Avg. loss | Avg. sparsity |
|---|---|---|---|---|---|
| | QC=1.0 | **0.16 ± 0.07** | 12.4 ± 8.4 | **0.371 ± 0.007** | **3.14 ± 0.23** |
| Coffee House | QC=0.001 | 0.70 ± 0.23 | 77.2 ± 89.2 | **0.300 ± 0.009** | **3.57 ± 0.07** |
| | QC=0.01 | 0.66 ± 0.20 | 77.2 ± 89.2 | **0.300 ± 0.009** | **3.57 ± 0.07** |
| | QC=0.1 | 0.49 ± 0.15 | 77.2 ± 89.2 | **0.300 ± 0.009** | **3.57 ± 0.07** |
| | QC=1.0 | **0.27 ± 0.09** | 77.2 ± 89.2 | **0.300 ± 0.009** | **3.57 ± 0.07** |
| COMPAS | QC=0.001 | 54.19 ± 19.10 | 1248.4 ± 860.9 | **0.331 ± 0.012** | **2.99 ± 0.05** |
| | QC=0.01 | 50.34 ± 17.99 | 1248.4 ± 860.9 | **0.331 ± 0.012** | **2.99 ± 0.05** |
| | QC=0.1 | 41.61 ± 14.92 | 1230.2 ± 866.3 | **0.331 ± 0.012** | 2.98 ± 0.05 |
| | QC=1.0 | **33.20 ± 12.95** | 1227.2 ± 861.4 | **0.331 ± 0.012** | 2.98 ± 0.05 |
| FICO | QC=0.001 | 2416.31 ± 905.35 | 154.4 ± 144.9 | **0.296 ± 0.010** | **3.12 ± 0.20** |
| | QC=0.01 | 2194.87 ± 851.91 | 154.4 ± 144.9 | **0.296 ± 0.010** | **3.12 ± 0.20** |
| | QC=0.1 | 1598.19 ± 520.68 | 154.4 ± 144.9 | **0.296 ± 0.010** | 3.11 ± 0.17 |
| | QC=1.0 | **1501.52 ± 377.98** | 154.4 ± 144.9 | **0.296 ± 0.010** | 3.11 ± 0.17 |
| Heart | QC=0.001 | 1.03 ± 0.63 | 1.0 ± 0.0 | **0.190 ± 0.025** | **3.70 ± 0.25** |
| | QC=0.01 | 1.00 ± 0.63 | 1.0 ± 0.0 | **0.190 ± 0.025** | **3.70 ± 0.25** |
| | QC=0.1 | 0.80 ± 0.49 | 1.0 ± 0.0 | **0.190 ± 0.025** | **3.70 ± 0.25** |
| | QC=1.0 | **0.55 ± 0.34** | 1.0 ± 0.0 | **0.190 ± 0.025** | **3.70 ± 0.25** |
| MGH | QC=0.001 | 0.79 ± 0.56 | 2.6 ± 0.5 | **0.061 ± 0.005** | **2.04 ± 0.03** |
| | QC=0.01 | 0.75 ± 0.52 | 2.6 ± 0.5 | **0.061 ± 0.005** | **2.04 ± 0.03** |
| | QC=0.1 | 0.59 ± 0.40 | 2.6 ± 0.5 | **0.061 ± 0.005** | **2.04 ± 0.03** |
| | QC=1.0 | **0.34 ± 0.22** | 2.6 ± 0.5 | **0.061 ± 0.005** | **2.04 ± 0.03** |
| Monks1 | QC=0.001 | 0.015 ± 0.006 | 12.0 ± 6.2 | **0.178 ± 0.110** | **3.16 ± 0.32** |
| | QC=0.01 | 0.014 ± 0.005 | 12.0 ± 6.2 | **0.178 ± 0.110** | **3.16 ± 0.32** |
| | QC=0.1 | 0.014 ± 0.005 | 12.0 ± 6.2 | **0.178 ± 0.110** | **3.16 ± 0.32** |
| | QC=1.0 | **0.012 ± 0.004** | 10.2 ± 5.0 | **0.178 ± 0.110** | 3.13 ± 0.32 |
| Monks2 | QC=0.001 | 0.016 ± 0.007 | 1.0 ± 0.0 | 0.441 ± 0.085 | 3.51 ± 0.22 |
| | QC=0.01 | 0.016 ± 0.007 | 1.0 ± 0.0 | 0.441 ± 0.085 | 3.51 ± 0.22 |
| | QC=0.1 | 0.015 ± 0.006 | 1.0 ± 0.0 | 0.441 ± 0.085 | 3.51 ± 0.22 |
| | QC=1.0 | **0.014 ± 0.007** | 1.0 ± 0.0 | **0.429 ± 0.066** | **3.57 ± 0.35** |
| Monks3 | QC=0.001 | 0.005 ± 0.003 | 1.0 ± 0.0 | **0.096 ± 0.090** | **3.73 ± 0.29** |
| | QC=0.01 | 0.005 ± 0.003 | 1.0 ± 0.0 | **0.096 ± 0.090** | **3.73 ± 0.29** |
| | QC=0.1 | 0.005 ± 0.003 | 1.0 ± 0.0 | **0.096 ± 0.090** | **3.73 ± 0.29** |
| | QC=1.0 | **0.003 ± 0.001** | 1.0 ± 0.0 | 0.128 ± 0.093 | 3.62 ± 0.35 |
| Mushroom | QC=0.001 | 0.037 ± 0.006 | 1.0 ± 0.0 | **0.030 ± 0.002** | **2.44 ± 0.01** |
| | QC=0.01 | 0.037 ± 0.006 | 1.0 ± 0.0 | **0.030 ± 0.002** | **2.44 ± 0.01** |
| | QC=0.1 | 0.034 ± 0.006 | 1.0 ± 0.0 | **0.030 ± 0.002** | **2.44 ± 0.01** |
| | QC=1.0 | **0.024 ± 0.004** | 1.0 ± 0.0 | **0.030 ± 0.002** | **2.44 ± 0.01** |
| Restaurant 20 | QC=0.001 | 0.50 ± 0.17 | 3.4 ± 1.4 | **0.298 ± 0.010** | **1.71 ± 0.20** |
| | QC=0.01 | 0.47 ± 0.16 | 3.4 ± 1.4 | **0.298 ± 0.010** | **1.71 ± 0.20** |
| | QC=0.1 | 0.40 ± 0.15 | 3.4 ± 1.4 | **0.298 ± 0.010** | **1.71 ± 0.20** |
| | QC=1.0 | **0.31 ± 0.13** | 3.4 ± 1.4 | **0.298 ± 0.010** | **1.71 ± 0.20** |
| Spambase | QC=0.001 | 59.44 ± 3.56 | 1.0 ± 0.0 | **0.105 ± 0.011** | 3.54 ± 0.08 |
| | QC=0.01 | 55.30 ± 3.51 | 1.0 ± 0.0 | **0.105 ± 0.011** | 3.54 ± 0.08 |
| | QC=0.1 | 38.48 ± 2.32 | 1.0 ± 0.0 | **0.105 ± 0.011** | 3.54 ± 0.08 |
| | QC=1.0 | **20.44 ± 1.69** | 1.0 ± 0.0 | 0.112 ± 0.006 | **3.62 ± 0.06** |
| Student | QC=0.001 | 1.46 ± 0.91 | 1.0 ± 0.0 | **0.192 ± 0.046** | **3.64 ± 0.23** |
| | QC=0.01 | 1.40 ± 0.87 | 1.0 ± 0.0 | **0.192 ± 0.046** | **3.64 ± 0.23** |
| | QC=0.1 | 1.13 ± 0.69 | 1.0 ± 0.0 | **0.192 ± 0.046** | **3.64 ± 0.23** |
| | QC=1.0 | **0.68 ± 0.41** | 1.0 ± 0.0 | **0.192 ± 0.046** | **3.64 ± 0.23** |
| Wine | QC=0.001 | 282.50 ± 114.86 | 530.4 ± 520.1 | **0.286 ± 0.016** | **3.39 ± 0.08** |
| | QC=0.01 | 259.02 ± 103.83 | 530.4 ± 520.1 | **0.286 ± 0.016** | **3.39 ± 0.08** |
| | QC=0.1 | 213.41 ± 85.69 | 530.4 ± 520.1 | **0.286 ± 0.016** | **3.39 ± 0.08** |
| | QC=1.0 | **139.75 ± 51.93** | 529.0 ± 519.0 | **0.286 ± 0.016** | **3.39 ± 0.09** |

*Table 8.* Ablation on different quantizations for the cache given $\mu = 0.01$. Values are mean ± std over 5 train-test trials.

# E. Comparison with Contemporary Rashomon Set Approaches

## E.1. Rationale for hyperparameter choices in the main paper

Table 2 shows that the Rashomon set output by **GRAVITree** can achieve a favourable tradeoff between the overall test loss, test sparsity, and the test loss on positive examples compared to TreeFARMS, an unconstrained, exact method for finding the Rashomon set.

Since both Rashomon sets are with respect to different objectives, making direct comparisons difficult, we discuss here the rationale for our specific hyperparameter choices, in the hope that they seem as fair as possible.

- For **GRAVITree**, we set $\mu = 0.01$, $\alpha = 0.05$, $d = 5$, and $\varepsilon = 0.01$. We empirically observed that $\mu = 0.01$ leads to modest branching, ensuring that trees remain expressive without being siloed into rule lists (e.g. see the middle tree in Figure 5).

- For TreeFARMS, we set depth $d = 5$. Technically, TreeFARMS has an off by one depth budget relative to ours. We correct for this by inputting $d = 6$ as an argument in these methods, but the trees output are actually of depth 5 (where depth 0 is the root).

- We needed to determine a value for $\lambda$ that is somewhat comparable to $\mu$ and $\alpha$. Intuitively, $\mu$ represents the minimum reduction in loss a subtree must incur relative to a leaf in order to be accepted into the tree. Likewise, the $\lambda$ penalty in TreeFARMS represents the minimum loss reduction that must occur in order to make a new split (i.e., add a new leaf). Since branching out can result in **several** additional leaves, this requires $\mu$ to be somewhat larger than $\lambda$ to ensure that both model classes have similar expressivity.

- We refer to the same rationale for SORTeD (Arslan et al., 2025). For both TreeFARMS and SORTeD, we perform experiments with $\mu = 0.01$ and 2 $\lambda$ values (0.005 and 0.002), as we may not know which values of $\mu$ and $\lambda$ result in trees with the most similar sparsity levels.

## E.2. TreeFARMS (Xin et al., 2022)

As we saw in Table 2, **GRAVITree** is comparable in overall test loss compared to TreeFARMS, but compensates for that with comparable or improved sparsity and false negative rate (FNR). We observe similar trends for $\lambda = 0.002$ as well (we included the results for $\lambda = 0.005$ from the main paper).

| Dataset | Rashomon Set Size | | Test Loss (↓) | |
|---|---|---|---|---|
| | **GRAVITree** | **TreeFARMS** | **GRAVITree** | **TreeFARMS** |
| NIJ Recidivism | $16.2 \pm 7.0$ | $4141.8 \pm 1266.5$ | $0.308 \pm 0.007$ | $\mathbf{0.305 \pm 0.002}$ |
| Adult | $1.0 \pm 0.0$ | $120.4 \pm 81.3$ | $0.293 \pm 0.005$ | $\mathbf{0.240 \pm 0.010}$ |
| Aging | $7.8 \pm 7.5$ | $1720.8 \pm 991.7$ | $0.387 \pm 0.044$ | $\mathbf{0.378 \pm 0.019}$ |
| Bar7 | $2.0 \pm 1.7$ | $49.2 \pm 42.7$ | $0.311 \pm 0.013$ | $\mathbf{0.292 \pm 0.007}$ |
| Bar | $7.6 \pm 2.3$ | $136.2 \pm 56.9$ | $0.298 \pm 0.015$ | $\mathbf{0.285 \pm 0.007}$ |
| BCW | $1.0 \pm 0.0$ | $32.4 \pm 28.6$ | $0.081 \pm 0.017$ | $\mathbf{0.071 \pm 0.009}$ |
| Bike | $1.0 \pm 0.0$ | $357.8 \pm 194.7$ | $0.175 \pm 0.006$ | $\mathbf{0.148 \pm 0.006}$ |
| Car Evaluation | $1.0 \pm 0.0$ | $2.0 \pm 0.0$ | $0.242 \pm 0.016$ | $\mathbf{0.141 \pm 0.015}$ |
| Carryout Takeaway | $1.8 \pm 1.6$ | $42.6 \pm 33.2$ | $\mathbf{0.360 \pm 0.026}$ | $0.394 \pm 0.022$ |
| Coffee House | $14.4 \pm 18.5$ | $292.8 \pm 91.6$ | $0.299 \pm 0.007$ | $\mathbf{0.296 \pm 0.008}$ |
| Compas | $249.0 \pm 220.4$ | $426.0 \pm 199.9$ | $\mathbf{0.331 \pm 0.012}$ | $\mathbf{0.331 \pm 0.012}$ |
| FICO | $2.8 \pm 2.2$ | $154.4 \pm 56.3$ | $0.301 \pm 0.008$ | $\mathbf{0.296 \pm 0.009}$ |
| Heart | $12.4 \pm 10.8$ | $10304.2 \pm 9487.4$ | $0.223 \pm 0.033$ | $\mathbf{0.203 \pm 0.032}$ |
| HELOC | $1.0 \pm 0.0$ | $130.8 \pm 67.1$ | $\mathbf{0.287 \pm 0.017}$ | $0.288 \pm 0.013$ |
| MGH | $1.4 \pm 0.5$ | $5.4 \pm 1.4$ | $0.061 \pm 0.005$ | $\mathbf{0.031 \pm 0.006}$ |
| Monks1 | $8.4 \pm 7.8$ | $6.0 \pm 5.1$ | $\mathbf{0.173 \pm 0.088}$ | $0.173 \pm 0.087$ |
| Monks2 | $444.8 \pm 469.6$ | $105.6 \pm 121.5$ | $0.501 \pm 0.070$ | $\mathbf{0.438 \pm 0.091}$ |
| Monks3 | $3.6 \pm 3.1$ | $45.2 \pm 39.1$ | $\mathbf{0.133 \pm 0.071}$ | $0.158 \pm 0.070$ |
| Mushroom | $1.0 \pm 0.0$ | $18.0 \pm 0.0$ | $0.037 \pm 0.003$ | $\mathbf{0.007 \pm 0.002}$ |
| Restaurant 20 | $2.4 \pm 1.1$ | $23.2 \pm 17.3$ | $\mathbf{0.298 \pm 0.010}$ | $0.313 \pm 0.011$ |
| Spambase | $5.8 \pm 4.2$ | $292.8 \pm 91.6$ | $0.126 \pm 0.006$ | $\mathbf{0.102 \pm 0.004}$ |
| Student | $1.4 \pm 0.5$ | $23.2 \pm 17.3$ | $0.267 \pm 0.062$ | $\mathbf{0.202 \pm 0.037}$ |
| Wine | $34.0 \pm 25.6$ | $380.6 \pm 211.7$ | $0.291 \pm 0.017$ | $\mathbf{0.277 \pm 0.008}$ |

*Table 9.* Rashomon Set Size and Global Objective Loss. $\lambda = 0.005$, $\varepsilon = 0.01$, $\mu = 0.01$, $\alpha = 0.05$

| Dataset | Sparsity (Pos) (↓) | | FNR (↓) | |
|---|---|---|---|---|
| | **GRAVITree** | **TreeFARMS** | **GRAVITree** | **TreeFARMS** |
| NIJ Recidivism | $2.66 \pm 0.16$ | $\mathbf{2.28 \pm 0.16}$ | $\mathbf{0.139 \pm 0.034}$ | $0.151 \pm 0.005$ |
| Adult | $\mathbf{2.49 \pm 0.42}$ | $2.64 \pm 0.05$ | $\mathbf{0.082 \pm 0.017}$ | $0.159 \pm 0.017$ |
| Aging | $\mathbf{3.31 \pm 0.08}$ | $3.62 \pm 0.11$ | $\mathbf{0.380 \pm 0.074}$ | $0.383 \pm 0.038$ |
| Bar7 | $\mathbf{2.14 \pm 0.25}$ | $3.03 \pm 0.10$ | $0.544 \pm 0.044$ | $\mathbf{0.497 \pm 0.037}$ |
| Bar | $\mathbf{2.78 \pm 0.07}$ | $3.19 \pm 0.04$ | $0.412 \pm 0.035$ | $\mathbf{0.397 \pm 0.018}$ |
| BCW | $2.88 \pm 0.13$ | $\mathbf{2.87 \pm 0.22}$ | $0.221 \pm 0.050$ | $\mathbf{0.076 \pm 0.010}$ |
| Bike | $\mathbf{3.01 \pm 0.14}$ | $3.38 \pm 0.09$ | $\mathbf{0.084 \pm 0.015}$ | $0.095 \pm 0.009$ |
| Car Evaluation | $\mathbf{2.00 \pm 0.00}$ | $2.00 \pm 0.00$ | $\mathbf{0.000 \pm 0.000}$ | $0.000 \pm 0.000$ |
| Carryout Takeaway | $\mathbf{2.61 \pm 0.38}$ | $3.07 \pm 0.08$ | $\mathbf{0.324 \pm 0.027}$ | $0.388 \pm 0.034$ |
| Coffee House | $3.31 \pm 0.06$ | $\mathbf{3.29 \pm 0.05}$ | $\mathbf{0.247 \pm 0.010}$ | $0.247 \pm 0.013$ |
| Compas | $2.74 \pm 0.05$ | $\mathbf{2.58 \pm 0.04}$ | $\mathbf{0.382 \pm 0.016}$ | $0.391 \pm 0.014$ |
| FICO | $\mathbf{1.89 \pm 0.71}$ | $1.98 \pm 0.02$ | $0.380 \pm 0.017$ | $\mathbf{0.356 \pm 0.019}$ |
| Heart | $\mathbf{2.70 \pm 0.35}$ | $3.26 \pm 0.31$ | $\mathbf{0.198 \pm 0.031}$ | $0.237 \pm 0.039$ |
| HELOC | $2.98 \pm 0.11$ | $\mathbf{2.25 \pm 0.08}$ | $\mathbf{0.139 \pm 0.016}$ | $0.146 \pm 0.014$ |
| MGH | $\mathbf{2.00 \pm 0.00}$ | $3.05 \pm 0.03$ | $\mathbf{0.023 \pm 0.009}$ | $0.024 \pm 0.009$ |
| Monks1 | $2.56 \pm 0.38$ | $\mathbf{2.19 \pm 0.27}$ | $\mathbf{0.135 \pm 0.143}$ | $0.144 \pm 0.156$ |
| Monks2 | $\mathbf{3.09 \pm 0.22}$ | $3.69 \pm 0.11$ | $0.641 \pm 0.190$ | $\mathbf{0.626 \pm 0.125}$ |
| Monks3 | $\mathbf{2.97 \pm 0.35}$ | $3.25 \pm 0.14$ | $\mathbf{0.067 \pm 0.097}$ | $0.221 \pm 0.096$ |
| Mushroom | $\mathbf{2.02 \pm 0.04}$ | $3.63 \pm 0.00$ | $0.023 \pm 0.002$ | $\mathbf{0.014 \pm 0.004}$ |
| Restaurant 20 | $\mathbf{1.51 \pm 0.10}$ | $2.18 \pm 0.29$ | $\mathbf{0.249 \pm 0.011}$ | $0.280 \pm 0.015$ |
| Spambase | $\mathbf{2.43 \pm 0.05}$ | $3.33 \pm 0.28$ | $0.181 \pm 0.022$ | $\mathbf{0.154 \pm 0.032}$ |
| Student | $\mathbf{2.88 \pm 0.47}$ | $3.86 \pm 0.04$ | $0.199 \pm 0.070$ | $\mathbf{0.191 \pm 0.040}$ |
| Wine | $\mathbf{2.58 \pm 0.03}$ | $2.59 \pm 0.16$ | $\mathbf{0.169 \pm 0.017}$ | $0.203 \pm 0.021$ |

*Table 10.* Positive Class Performance: Average Sparsity and Loss. $\lambda = 0.005$, $\varepsilon = 0.01$, $\mu = 0.01$, $\alpha = 0.05$

| Dataset | GRAVITree Runtime (s) ($\downarrow$) | TreeFARMS Runtime (s) ($\downarrow$) |
|---|---|---|
| NIJ Recidivism | **1.01 $\pm$ 0.04** | 3.03 $\pm$ 0.10 |
| Adult | **3.45 $\pm$ 0.54** | 6.95 $\pm$ 1.23 |
| Aging | **0.28 $\pm$ 0.18** | 2.47 $\pm$ 1.42 |
| Bar7 | **0.01 $\pm$ 0.00** | 0.06 $\pm$ 0.02 |
| Bar | **0.07 $\pm$ 0.02** | 0.39 $\pm$ 0.12 |
| BCW | **0.01 $\pm$ 0.00** | 0.10 $\pm$ 0.02 |
| Bike | **6.11 $\pm$ 0.76** | 31.07 $\pm$ 4.48 |
| Car Evaluation | **0.03 $\pm$ 0.01** | 0.12 $\pm$ 0.03 |
| Carryout Takeaway | **0.05 $\pm$ 0.03** | 0.51 $\pm$ 0.23 |
| Coffee House | **0.21 $\pm$ 0.05** | 0.65 $\pm$ 0.22 |
| Compas | **5.55 $\pm$ 1.34** | 11.62 $\pm$ 2.76 |
| FICO | **45.61 $\pm$ 9.76** | 223.45 $\pm$ 34.35 |
| Heart | **0.21 $\pm$ 0.09** | 4.64 $\pm$ 2.31 |
| MGH | **0.18 $\pm$ 0.11** | 0.49 $\pm$ 0.29 |
| Monks1 | **0.01 $\pm$ 0.01** | 0.10 $\pm$ 0.07 |
| Monks2 | **0.03 $\pm$ 0.03** | 0.10 $\pm$ 0.03 |
| Monks3 | **0.00 $\pm$ 0.00** | 0.03 $\pm$ 0.02 |
| Mushroom | **0.02 $\pm$ 0.00** | 0.10 $\pm$ 0.01 |
| Restaurant 20 | **0.08 $\pm$ 0.02** | 0.64 $\pm$ 0.14 |
| Spambase | **3.57 $\pm$ 0.31** | 30.96 $\pm$ 1.98 |
| Student | **0.22 $\pm$ 0.11** | 2.46 $\pm$ 1.60 |
| Wine | **15.79 $\pm$ 4.04** | 100.71 $\pm$ 35.98 |

*Table 11.* Computational Efficiency: **GRAVITree** vs. TreeFARMS Runtime. $\lambda = 0.005$, $\varepsilon = 0.01$, $\mu = 0.01$, $\alpha = 0.05$

| Dataset | Rashomon Set Size | | Test Loss ($\downarrow$) | |
|---|---|---|---|---|
| | GRAVITree | TreeFARMS | GRAVITree | TF |
| NIJ Recidivism | $16.2 \pm 8.3$ | $1517.4 \pm 424.6$ | $0.308 \pm 0.007$ | $\mathbf{0.299 \pm 0.003}$ |
| Adult | $1.0 \pm 0.0$ | $5310.4 \pm 2987.5$ | $0.293 \pm 0.005$ | $\mathbf{0.226 \pm 0.017}$ |
| Aging | $7.8 \pm 9.7$ | $7113.6 \pm 2255.7$ | $\mathbf{0.387 \pm 0.044}$ | $0.396 \pm 0.024$ |
| Bar7 | $2.0 \pm 1.4$ | $368.4 \pm 110.5$ | $0.311 \pm 0.013$ | $\mathbf{0.285 \pm 0.009}$ |
| Bar | $7.6 \pm 2.4$ | $1027.2 \pm 322.9$ | $0.298 \pm 0.015$ | $\mathbf{0.280 \pm 0.008}$ |
| BCW | $1.0 \pm 0.0$ | $122.4 \pm 160.6$ | $0.081 \pm 0.017$ | $\mathbf{0.079 \pm 0.006}$ |
| Bike | $1.0 \pm 0.0$ | $441.0 \pm 148.2$ | $0.175 \pm 0.006$ | $\mathbf{0.142 \pm 0.003}$ |
| Car Evaluation | $1.0 \pm 0.0$ | $2.8 \pm 1.0$ | $0.242 \pm 0.016$ | $\mathbf{0.136 \pm 0.012}$ |
| Carryout Takeaway | $1.8 \pm 1.7$ | $2318.0 \pm 549.6$ | $\mathbf{0.360 \pm 0.026}$ | $0.382 \pm 0.019$ |
| Coffee House | $14.4 \pm 15.9$ | $263.6 \pm 148.4$ | $0.299 \pm 0.007$ | $\mathbf{0.291 \pm 0.010}$ |
| Compas | $249.0 \pm 165.6$ | $29662.2 \pm 3444.7$ | $0.331 \pm 0.012$ | $\mathbf{0.327 \pm 0.015}$ |
| FICO | $2.8 \pm 1.3$ | $5145.2 \pm 285.8$ | $0.301 \pm 0.008$ | $\mathbf{0.294 \pm 0.011}$ |
| Heart | $12.4 \pm 11.4$ | $209.6 \pm 322.3$ | $0.223 \pm 0.033$ | $\mathbf{0.209 \pm 0.042}$ |
| HELOC | $1.0 \pm 0.0$ | $12522.4 \pm 2892.3$ | $0.287 \pm 0.017$ | $\mathbf{0.287 \pm 0.015}$ |
| MGH | $1.4 \pm 0.5$ | $4.0 \pm 0.0$ | $0.061 \pm 0.005$ | $\mathbf{0.032 \pm 0.010}$ |
| Monks1 | $8.4 \pm 6.2$ | $16.0 \pm 14.4$ | $\mathbf{0.173 \pm 0.088}$ | $0.184 \pm 0.088$ |
| Monks2 | $444.8 \pm 546.0$ | $106.6 \pm 102.4$ | $0.501 \pm 0.070$ | $\mathbf{0.443 \pm 0.099}$ |
| Monks3 | $3.6 \pm 3.2$ | $31.2 \pm 17.6$ | $\mathbf{0.133 \pm 0.071}$ | $0.157 \pm 0.070$ |
| Mushroom | $1.0 \pm 0.0$ | $20.0 \pm 0.0$ | $0.037 \pm 0.003$ | $\mathbf{0.004 \pm 0.001}$ |
| Restaurant 20 | $2.4 \pm 1.0$ | $2828.6 \pm 430.9$ | $\mathbf{0.298 \pm 0.010}$ | $0.325 \pm 0.014$ |
| Spambase | $5.8 \pm 3.2$ | $1003.6 \pm 489.0$ | $0.126 \pm 0.006$ | $\mathbf{0.097 \pm 0.005}$ |
| Student | $1.4 \pm 0.5$ | $1627.2 \pm 1411.3$ | $0.267 \pm 0.062$ | $\mathbf{0.208 \pm 0.037}$ |
| Wine | $34.0 \pm 22.5$ | $2367.4 \pm 223.5$ | $0.291 \pm 0.017$ | $\mathbf{0.276 \pm 0.009}$ |

*Table 12.* Rashomon Set Size and Test Loss. $\lambda = 0.002, \varepsilon = 0.01, \mu = 0.01, \alpha = 0.05$

| Dataset | Sparsity (Pos) ($\downarrow$) | | FNR ($\downarrow$) | |
|---|---|---|---|---|
| | GRAVITree | TreeFARMS | GRAVITree | TreeFARMS |
| NIJ Recidivism | $\mathbf{2.66 \pm 0.16}$ | $2.76 \pm 0.01$ | $\mathbf{0.139 \pm 0.034}$ | $0.161 \pm 0.006$ |
| Adult | $\mathbf{2.49 \pm 0.42}$ | $2.91 \pm 0.10$ | $\mathbf{0.082 \pm 0.017}$ | $0.168 \pm 0.031$ |
| Aging | $\mathbf{3.31 \pm 0.08}$ | $3.90 \pm 0.07$ | $\mathbf{0.380 \pm 0.074}$ | $0.403 \pm 0.048$ |
| Bar7 | $\mathbf{2.14 \pm 0.25}$ | $3.34 \pm 0.08$ | $0.544 \pm 0.044$ | $\mathbf{0.477 \pm 0.046}$ |
| Bar | $\mathbf{2.78 \pm 0.07}$ | $3.57 \pm 0.05$ | $0.412 \pm 0.035$ | $\mathbf{0.391 \pm 0.014}$ |
| BCW | $2.88 \pm 0.13$ | $\mathbf{3.07 \pm 0.10}$ | $0.221 \pm 0.050$ | $\mathbf{0.101 \pm 0.030}$ |
| Bike | $\mathbf{3.01 \pm 0.14}$ | $3.60 \pm 0.19$ | $\mathbf{0.084 \pm 0.015}$ | $0.091 \pm 0.009$ |
| Car Evaluation | $\mathbf{2.00 \pm 0.00}$ | $2.28 \pm 0.34$ | $\mathbf{0.000 \pm 0.000}$ | $0.005 \pm 0.006$ |
| Carryout Takeaway | $\mathbf{2.61 \pm 0.38}$ | $3.74 \pm 0.14$ | $\mathbf{0.324 \pm 0.027}$ | $0.372 \pm 0.028$ |
| Coffee House | $\mathbf{3.31 \pm 0.06}$ | $3.49 \pm 0.04$ | $0.247 \pm 0.010$ | $\mathbf{0.247 \pm 0.013}$ |
| Compas | $\mathbf{2.74 \pm 0.05}$ | $3.05 \pm 0.10$ | $\mathbf{0.382 \pm 0.016}$ | $0.392 \pm 0.017$ |
| FICO | $\mathbf{1.89 \pm 0.71}$ | $2.87 \pm 0.18$ | $0.380 \pm 0.017$ | $\mathbf{0.355 \pm 0.015}$ |
| Heart | $\mathbf{2.70 \pm 0.35}$ | $3.64 \pm 0.17$ | $\mathbf{0.198 \pm 0.031}$ | $0.252 \pm 0.042$ |
| MGH | $\mathbf{2.00 \pm 0.00}$ | $3.06 \pm 0.03$ | $\mathbf{0.023 \pm 0.009}$ | $0.023 \pm 0.010$ |
| Monks1 | $\mathbf{2.56 \pm 0.38}$ | $3.32 \pm 0.15$ | $\mathbf{0.135 \pm 0.143}$ | $0.198 \pm 0.104$ |
| Monks2 | $\mathbf{3.09 \pm 0.22}$ | $3.78 \pm 0.09$ | $0.641 \pm 0.190$ | $\mathbf{0.576 \pm 0.205}$ |
| Monks3 | $\mathbf{2.97 \pm 0.35}$ | $3.35 \pm 0.12$ | $\mathbf{0.067 \pm 0.097}$ | $0.250 \pm 0.115$ |
| Mushroom | $\mathbf{2.02 \pm 0.04}$ | $3.58 \pm 0.00$ | $0.023 \pm 0.002$ | $\mathbf{0.008 \pm 0.003}$ |
| Restaurant 20 | $\mathbf{1.51 \pm 0.10}$ | $3.23 \pm 0.20$ | $\mathbf{0.249 \pm 0.011}$ | $0.312 \pm 0.027$ |
| Spambase | $\mathbf{2.43 \pm 0.05}$ | $3.39 \pm 0.19$ | $0.181 \pm 0.022$ | $\mathbf{0.178 \pm 0.010}$ |
| Student | $\mathbf{2.88 \pm 0.47}$ | $3.93 \pm 0.04$ | $0.199 \pm 0.070$ | $\mathbf{0.184 \pm 0.045}$ |
| Wine | $\mathbf{2.58 \pm 0.03}$ | $3.57 \pm 0.22$ | $\mathbf{0.169 \pm 0.017}$ | $0.205 \pm 0.026$ |

*Table 13.* Positive Class Performance: Average Sparsity and Loss. $\lambda = 0.002, \varepsilon = 0.01, \mu = 0.01, \alpha = 0.05$

| Dataset | GRAVITree Runtime (s) (↓) | TreeFARMS Runtime (s) (↓) |
|---|---|---|
| NIJ Recidivism | **1.03 ± 0.04** | 3.28 ± 0.16 |
| Adult | **3.44 ± 0.56** | 8.12 ± 1.30 |
| Aging | **0.28 ± 0.18** | 2.54 ± 1.37 |
| Bar7 | **0.01 ± 0.00** | 0.07 ± 0.02 |
| Bar | **0.07 ± 0.02** | 0.42 ± 0.13 |
| BCW | **0.01 ± 0.00** | 0.11 ± 0.02 |
| Bike | **6.15 ± 0.74** | 32.08 ± 4.73 |
| Car Evaluation | **0.03 ± 0.01** | 0.14 ± 0.04 |
| Carryout Takeaway | **0.05 ± 0.03** | 0.53 ± 0.23 |
| Coffee House | **0.22 ± 0.04** | 0.67 ± 0.23 |
| Compas | **5.54 ± 1.36** | 12.52 ± 2.85 |
| FICO | **45.63 ± 9.81** | 243.81 ± 27.33 |
| Heart | **0.23 ± 0.10** | 5.12 ± 2.35 |
| MGH | **0.18 ± 0.10** | 0.95 ± 0.64 |
| Monks1 | **0.01 ± 0.01** | 0.10 ± 0.07 |
| Monks2 | **0.03 ± 0.03** | 0.11 ± 0.04 |
| Monks3 | **0.00 ± 0.00** | 0.03 ± 0.02 |
| Mushroom | **0.02 ± 0.00** | 0.12 ± 0.01 |
| Restaurant 20 | **0.07 ± 0.02** | 0.64 ± 0.13 |
| Spambase | **3.57 ± 0.31** | 32.24 ± 1.64 |
| Student | **0.20 ± 0.10** | 2.38 ± 1.49 |
| Wine | **15.44 ± 4.04** | 102.80 ± 37.97 |

*Table 14.* Computational Efficiency: **GRAVITree** vs. TreeFARMS Runtime. $\lambda = 0.002, \varepsilon = 0.01, \mu = 0.01, \alpha = 0.05$

### E.3. SORTeD (Arslan et al., 2025)

We report similar results for the SORTeD algorithm as TreeFARMS. Note that SORTeD optimizes the same objective function as TreeFARMS. In that spirit, we perform experiments with the same hyperparameters as above, i.e:

- For **GRAVITree**, we set $\mu = 0.01, \alpha = 0.05, \varepsilon = 0.01$, depth $= 5$.

- For SORTeD, we set $\lambda \in \{0.002, 0.005\}, \varepsilon = 0.01$ depth $= 5$

| Dataset | GRAVITree Runtime (s) (↓) | SORTeD Runtime (s) (↓) |
|---|---|---|
| NIJ Recidivism | $1.49 \pm 0.07$ | $\mathbf{0.23 \pm 0.03}$ |
| Adult | $5.01 \pm 0.89$ | $\mathbf{1.45 \pm 0.08}$ |
| Aging | $0.38 \pm 0.22$ | $\mathbf{0.07 \pm 0.01}$ |
| Bar7 | $\mathbf{0.02 \pm 0.01}$ | $0.03 \pm 0.00$ |
| Bar | $0.07 \pm 0.02$ | $\mathbf{0.04 \pm 0.01}$ |
| BCW | $\mathbf{0.01 \pm 0.00}$ | $0.02 \pm 0.00$ |
| Bike | $8.94 \pm 0.99$ | $\mathbf{1.05 \pm 0.05}$ |
| Car Evaluation | $0.03 \pm 0.01$ | $\mathbf{0.02 \pm 0.00}$ |
| Carryout Takeaway | $0.07 \pm 0.03$ | $\mathbf{0.04 \pm 0.01}$ |
| Coffee House | $0.18 \pm 0.04$ | $\mathbf{0.06 \pm 0.01}$ |
| Compas | $5.63 \pm 1.50$ | $\mathbf{0.70 \pm 0.11}$ |
| FICO | $103.51 \pm 34.09$ | $\mathbf{4.94 \pm 0.90}$ |
| Heart | $0.20 \pm 0.09$ | $\mathbf{0.05 \pm 0.01}$ |
| HELOC | $5.94 \pm 0.98$ | $\mathbf{0.39 \pm 0.01}$ |
| MGH | $0.26 \pm 0.14$ | $\mathbf{0.08 \pm 0.02}$ |
| Monks1 | $\mathbf{0.01 \pm 0.00}$ | $0.01 \pm 0.00$ |
| Monks2 | $\mathbf{0.01 \pm 0.00}$ | $0.01 \pm 0.00$ |
| Monks3 | $\mathbf{0.00 \pm 0.00}$ | $0.01 \pm 0.00$ |
| Mushroom | $\mathbf{0.03 \pm 0.00}$ | $0.04 \pm 0.00$ |
| Restaurant 20 | $0.13 \pm 0.03$ | $\mathbf{0.04 \pm 0.01}$ |
| Spambase | $6.93 \pm 0.70$ | $\mathbf{0.39 \pm 0.03}$ |
| Student | $0.31 \pm 0.15$ | $\mathbf{0.05 \pm 0.01}$ |
| Wine | $21.03 \pm 5.70$ | $\mathbf{1.52 \pm 0.26}$ |

*Table 15.* Computational Efficiency: **GRAVITree** vs. SORTeD Runtimes. $\lambda = 0.002, \varepsilon = 0.01, \mu = 0.01, \alpha = 0.05$

| Dataset | Rashomon Set Size | | Test Loss (↓) | |
|---|---|---|---|---|
| | GRAVITree | SORTeD | GRAVITree | SORTeD |
| NIJ Recidivism | $94.8 \pm 22.0$ | $833.0 \pm 270.0$ | $0.305 \pm 0.003$ | $\mathbf{0.298 \pm 0.003}$ |
| Adult | $2.0 \pm 1.3$ | $282.8 \pm 141.6$ | $0.227 \pm 0.031$ | $\mathbf{0.222 \pm 0.016}$ |
| Aging | $16.6 \pm 20.3$ | $222.2 \pm 70.5$ | $\mathbf{0.388 \pm 0.043}$ | $0.389 \pm 0.023$ |
| Bar7 | $4.6 \pm 2.2$ | $797.6 \pm 239.5$ | $0.309 \pm 0.015$ | $\mathbf{0.284 \pm 0.010}$ |
| Bar | $8.4 \pm 2.3$ | $928.8 \pm 292.1$ | $0.298 \pm 0.015$ | $\mathbf{0.281 \pm 0.008}$ |
| BCW | $1.0 \pm 0.0$ | $21.8 \pm 28.6$ | $\mathbf{0.067 \pm 0.007}$ | $0.081 \pm 0.008$ |
| Bike | $1.0 \pm 0.0$ | $122.6 \pm 41.2$ | $0.156 \pm 0.007$ | $\mathbf{0.141 \pm 0.003}$ |
| Car Evaluation | $1.0 \pm 0.0$ | $2.8 \pm 1.0$ | $0.141 \pm 0.015$ | $\mathbf{0.136 \pm 0.012}$ |
| Carryout Takeaway | $8.8 \pm 5.7$ | $870.8 \pm 206.1$ | $\mathbf{0.370 \pm 0.008}$ | $0.380 \pm 0.016$ |
| Coffee House | $43.6 \pm 54.9$ | $617.0 \pm 347.7$ | $0.300 \pm 0.009$ | $\mathbf{0.289 \pm 0.010}$ |
| Compas | $594.0 \pm 388.6$ | $1143.4 \pm 132.8$ | $0.331 \pm 0.012$ | $\mathbf{0.328 \pm 0.015}$ |
| FICO | $251.8 \pm 201.4$ | $1077.4 \pm 59.8$ | $0.296 \pm 0.010$ | $\mathbf{0.294 \pm 0.010}$ |
| Heart | $1.8 \pm 1.6$ | $200.8 \pm 308.8$ | $\mathbf{0.200 \pm 0.024}$ | $0.204 \pm 0.046$ |
| HELOC | $85.8 \pm 60.5$ | $1316.0 \pm 303.6$ | $0.293 \pm 0.014$ | $\mathbf{0.286 \pm 0.015}$ |
| MGH | $2.6 \pm 0.5$ | $4.0 \pm 0.0$ | $0.061 \pm 0.005$ | $\mathbf{0.032 \pm 0.010}$ |
| Monks1 | $12.6 \pm 10.6$ | $31.8 \pm 28.7$ | $\mathbf{0.177 \pm 0.089}$ | $0.185 \pm 0.089$ |
| Monks2 | $1.0 \pm 0.0$ | $569.8 \pm 547.3$ | $\mathbf{0.418 \pm 0.051}$ | $0.433 \pm 0.099$ |
| Monks3 | $11.2 \pm 17.9$ | $31.2 \pm 17.6$ | $\mathbf{0.141 \pm 0.085}$ | $0.157 \pm 0.070$ |
| Mushroom | $1.0 \pm 0.0$ | $20.0 \pm 0.0$ | $0.030 \pm 0.002$ | $\mathbf{0.004 \pm 0.001}$ |
| Restaurant 20 | $3.4 \pm 1.4$ | $1133.4 \pm 172.4$ | $\mathbf{0.298 \pm 0.010}$ | $0.326 \pm 0.015$ |
| Spambase | $2.6 \pm 3.2$ | $105.8 \pm 51.6$ | $0.114 \pm 0.005$ | $\mathbf{0.097 \pm 0.004}$ |
| Student | $1.0 \pm 0.0$ | $48.8 \pm 42.3$ | $\mathbf{0.206 \pm 0.023}$ | $0.208 \pm 0.037$ |
| Wine | $86.0 \pm 56.6$ | $1060.2 \pm 100.0$ | $0.287 \pm 0.016$ | $\mathbf{0.276 \pm 0.009}$ |

*Table 16.* Rashomon Set Size and Global Objective Loss. $\lambda = 0.002, \varepsilon = 0.01, \mu = 0.01, \alpha = 0.05$

| Dataset | Sparsity (Pos) (↓) | | FNR (↓) | |
|---|---|---|---|---|
| | GRAVITree | SORTeD | GRAVITree | SORTeD |
| NIJ Recidivism | **2.67 ± 0.07** | 2.77 ± 0.03 | **0.130 ± 0.012** | 0.161 ± 0.008 |
| Adult | **2.73 ± 0.18** | 3.03 ± 0.10 | 0.185 ± 0.046 | **0.174 ± 0.030** |
| Aging | **3.35 ± 0.15** | 3.92 ± 0.06 | **0.384 ± 0.070** | 0.394 ± 0.044 |
| Bar7 | **2.31 ± 0.36** | 3.32 ± 0.08 | 0.523 ± 0.060 | **0.477 ± 0.046** |
| Bar | **2.80 ± 0.07** | 3.54 ± 0.07 | 0.414 ± 0.033 | **0.394 ± 0.017** |
| BCW | **2.92 ± 0.26** | 3.01 ± 0.22 | **0.067 ± 0.016** | 0.106 ± 0.032 |
| Bike | **3.27 ± 0.13** | 3.50 ± 0.14 | **0.076 ± 0.008** | 0.087 ± 0.006 |
| Car Evaluation | **2.09 ± 0.18** | 2.27 ± 0.34 | **0.000 ± 0.000** | 0.005 ± 0.006 |
| Carryout Takeaway | **2.98 ± 0.20** | 3.74 ± 0.13 | **0.334 ± 0.004** | 0.369 ± 0.026 |
| Coffee House | **3.31 ± 0.08** | 3.55 ± 0.03 | 0.253 ± 0.012 | **0.237 ± 0.010** |
| Compas | **2.74 ± 0.03** | 2.92 ± 0.11 | **0.383 ± 0.015** | 0.393 ± 0.021 |
| FICO | **2.70 ± 0.11** | 2.74 ± 0.18 | 0.362 ± 0.021 | **0.349 ± 0.018** |
| Heart | **3.15 ± 0.20** | 3.65 ± 0.17 | **0.201 ± 0.028** | 0.250 ± 0.043 |
| HELOC | **2.90 ± 0.06** | 3.25 ± 0.15 | **0.141 ± 0.015** | 0.147 ± 0.019 |
| MGH | **2.01 ± 0.01** | 3.06 ± 0.03 | 0.025 ± 0.009 | **0.023 ± 0.010** |
| Monks1 | **2.59 ± 0.36** | 3.30 ± 0.14 | **0.149 ± 0.137** | 0.196 ± 0.103 |
| Monks2 | **3.51 ± 0.32** | 3.82 ± 0.09 | **0.508 ± 0.125** | 0.574 ± 0.206 |
| Monks3 | **3.14 ± 0.28** | 3.35 ± 0.12 | **0.164 ± 0.114** | 0.250 ± 0.115 |
| Mushroom | **2.51 ± 0.01** | 3.58 ± 0.00 | 0.034 ± 0.005 | **0.008 ± 0.003** |
| Restaurant 20 | **1.65 ± 0.20** | 3.18 ± 0.24 | **0.250 ± 0.012** | 0.313 ± 0.029 |
| Spambase | **2.60 ± 0.24** | 3.34 ± 0.24 | **0.177 ± 0.018** | 0.178 ± 0.011 |
| Student | **3.48 ± 0.17** | 3.92 ± 0.04 | 0.200 ± 0.041 | **0.185 ± 0.043** |
| Wine | **2.79 ± 0.12** | 3.59 ± 0.26 | **0.182 ± 0.018** | 0.208 ± 0.028 |

*Table 17.* Positive Class Performance: Sparsity and Loss. $\lambda = 0.002$, $\varepsilon = 0.01$, $\mu = 0.01$, $\alpha = 0.05$

| Dataset | GRAVITree Runtime (s) (↓) | SORTeD Runtime (s) (↓) |
|---|---|---|
| NIJ Recidivism | 1.52 ± 0.09 | **0.21 ± 0.01** |
| Adult | 4.99 ± 0.92 | **1.32 ± 0.08** |
| Aging | 0.37 ± 0.21 | **0.07 ± 0.01** |
| Bar7 | **0.02 ± 0.01** | **0.02 ± 0.00** |
| Bar | 0.07 ± 0.02 | **0.03 ± 0.00** |
| BCW | **0.01 ± 0.00** | 0.02 ± 0.00 |
| Bike | 9.54 ± 1.04 | **1.23 ± 0.07** |
| Car Evaluation | 0.03 ± 0.01 | **0.02 ± 0.00** |
| Carryout Takeaway | 0.07 ± 0.03 | **0.04 ± 0.01** |
| Coffee House | 0.18 ± 0.04 | **0.06 ± 0.01** |
| Compas | 5.88 ± 1.50 | **0.70 ± 0.13** |
| FICO | 96.60 ± 31.00 | **4.15 ± 0.75** |
| Heart | 0.19 ± 0.09 | **0.05 ± 0.01** |
| HELOC | 6.00 ± 0.94 | **0.50 ± 0.02** |
| MGH | 0.31 ± 0.17 | **0.12 ± 0.04** |
| Monks1 | **0.01 ± 0.00** | 0.01 ± 0.00 |
| Monks2 | **0.01 ± 0.00** | 0.01 ± 0.00 |
| Monks3 | **0.00 ± 0.00** | 0.01 ± 0.00 |
| Mushroom | **0.03 ± 0.00** | 0.04 ± 0.00 |
| Restaurant 20 | 0.15 ± 0.03 | **0.04 ± 0.01** |
| Spambase | 6.84 ± 0.64 | **0.38 ± 0.02** |
| Student | 0.30 ± 0.14 | **0.05 ± 0.01** |
| Wine | 20.92 ± 5.61 | **1.58 ± 0.26** |

*Table 18.* Computational Efficiency: **GRAVITree** vs. SORTeD Runtimes. $\lambda = 0.005$, $\varepsilon = 0.01$, $\mu = 0.01$, $\alpha = 0.05$

| | Rashomon Set Size | | Test Loss (↓) | |
|---|---|---|---|---|
| **Dataset** | **GRAVITree** | **SORTeD** | **GRAVITree** | **SORTeD** |
| NIJ Recidivism | 94.8 ± 22.0 | 87.0 ± 17.9 | **0.305 ± 0.003** | 0.306 ± 0.002 |
| Adult | 2.0 ± 1.3 | 32.0 ± 22.9 | **0.227 ± 0.031** | 0.233 ± 0.017 |
| Aging | 16.6 ± 20.3 | 223.2 ± 101.8 | 0.388 ± 0.043 | **0.380 ± 0.029** |
| Bar7 | 4.6 ± 2.2 | 47.6 ± 21.4 | 0.309 ± 0.015 | **0.292 ± 0.007** |
| Bar | 8.4 ± 2.3 | 136.2 ± 56.9 | 0.298 ± 0.015 | **0.285 ± 0.007** |
| BCW | 1.0 ± 0.0 | 25.8 ± 18.4 | **0.067 ± 0.007** | 0.070 ± 0.008 |
| Bike | 1.0 ± 0.0 | 65.2 ± 23.7 | 0.156 ± 0.007 | **0.148 ± 0.005** |
| Car Evaluation | 1.0 ± 0.0 | 2.0 ± 0.0 | **0.141 ± 0.015** | **0.141 ± 0.015** |
| Carryout Takeaway | 8.8 ± 5.7 | 42.6 ± 33.2 | **0.370 ± 0.008** | 0.394 ± 0.022 |
| Coffee House | 43.6 ± 54.9 | 292.8 ± 91.6 | 0.300 ± 0.009 | **0.296 ± 0.008** |
| Compas | 594.0 ± 388.6 | 426.0 ± 199.9 | **0.331 ± 0.012** | **0.331 ± 0.012** |
| FICO | 251.8 ± 201.4 | 154.4 ± 56.3 | 0.296 ± 0.010 | **0.296 ± 0.009** |
| Heart | 1.8 ± 1.6 | 31.8 ± 24.1 | 0.200 ± 0.024 | **0.193 ± 0.046** |
| HELOC | 85.8 ± 60.5 | 130.8 ± 67.1 | 0.293 ± 0.014 | **0.288 ± 0.013** |
| MGH | 2.6 ± 0.5 | 5.4 ± 1.4 | 0.061 ± 0.005 | **0.031 ± 0.006** |
| Monks1 | 12.6 ± 10.6 | 6.0 ± 5.1 | 0.177 ± 0.089 | **0.173 ± 0.087** |
| Monks2 | 1.0 ± 0.0 | 105.6 ± 121.5 | **0.418 ± 0.051** | 0.438 ± 0.091 |
| Monks3 | 11.2 ± 17.9 | 45.2 ± 39.1 | **0.141 ± 0.085** | 0.158 ± 0.070 |
| Mushroom | 1.0 ± 0.0 | 18.0 ± 0.0 | 0.030 ± 0.002 | **0.007 ± 0.002** |
| Restaurant 20 | 3.4 ± 1.4 | 23.2 ± 17.3 | **0.298 ± 0.010** | 0.313 ± 0.011 |
| Spambase | 2.6 ± 3.2 | 7.0 ± 6.6 | 0.114 ± 0.005 | **0.103 ± 0.003** |
| Student | 1.0 ± 0.0 | 3.2 ± 1.9 | 0.206 ± 0.023 | **0.203 ± 0.037** |
| Wine | 86.0 ± 56.6 | 243.0 ± 94.3 | 0.287 ± 0.016 | **0.277 ± 0.008** |

*Table 19.* Rashomon Set Size and Global Objective Loss. $\lambda = 0.005$, $\varepsilon = 0.01$, $\mu = 0.01$, $\alpha = 0.05$

| | Sparsity (Pos) (↓) | | FNR (↓) | |
|---|---|---|---|---|
| **Dataset** | **GRAVITree** | **SORTeD** | **GRAVITree** | **SORTeD** |
| NIJ Recidivism | 2.67 ± 0.07 | **1.87 ± 0.02** | **0.130 ± 0.012** | 0.144 ± 0.003 |
| Adult | 2.73 ± 0.18 | **2.71 ± 0.08** | 0.185 ± 0.046 | **0.169 ± 0.033** |
| Aging | **3.35 ± 0.15** | 3.62 ± 0.08 | **0.384 ± 0.070** | 0.386 ± 0.046 |
| Bar7 | **2.31 ± 0.36** | 3.02 ± 0.09 | 0.523 ± 0.060 | **0.496 ± 0.038** |
| Bar | **2.80 ± 0.07** | 3.19 ± 0.04 | 0.414 ± 0.033 | **0.397 ± 0.018** |
| BCW | **2.92 ± 0.26** | 2.94 ± 0.23 | **0.067 ± 0.016** | 0.075 ± 0.010 |
| Bike | **3.27 ± 0.13** | 3.36 ± 0.10 | **0.076 ± 0.008** | 0.098 ± 0.010 |
| Car Evaluation | 2.09 ± 0.18 | **2.00 ± 0.00** | **0.000 ± 0.000** | **0.000 ± 0.000** |
| Carryout Takeaway | **2.98 ± 0.20** | 3.07 ± 0.08 | **0.334 ± 0.004** | 0.388 ± 0.034 |
| Coffee House | 3.31 ± 0.08 | **3.29 ± 0.05** | 0.253 ± 0.012 | **0.247 ± 0.013** |
| Compas | 2.74 ± 0.03 | **2.58 ± 0.04** | **0.383 ± 0.015** | 0.391 ± 0.014 |
| FICO | 2.70 ± 0.11 | **1.98 ± 0.02** | 0.362 ± 0.021 | **0.356 ± 0.019** |
| Heart | 3.15 ± 0.20 | **2.99 ± 0.27** | **0.201 ± 0.028** | 0.211 ± 0.064 |
| HELOC | 2.90 ± 0.06 | **2.25 ± 0.08** | **0.141 ± 0.015** | 0.146 ± 0.014 |
| MGH | **2.01 ± 0.01** | 3.05 ± 0.03 | 0.025 ± 0.009 | **0.024 ± 0.009** |
| Monks1 | 2.59 ± 0.36 | **2.19 ± 0.27** | 0.149 ± 0.137 | **0.144 ± 0.156** |
| Monks2 | **3.51 ± 0.32** | 3.69 ± 0.11 | **0.508 ± 0.125** | 0.626 ± 0.125 |
| Monks3 | **3.14 ± 0.28** | 3.25 ± 0.14 | **0.164 ± 0.114** | 0.221 ± 0.096 |
| Mushroom | **2.51 ± 0.01** | 3.63 ± 0.01 | 0.034 ± 0.005 | **0.014 ± 0.004** |
| Restaurant 20 | **1.65 ± 0.20** | 2.18 ± 0.29 | **0.250 ± 0.012** | 0.280 ± 0.015 |
| Spambase | **2.60 ± 0.24** | 3.21 ± 0.40 | 0.177 ± 0.018 | **0.161 ± 0.039** |
| Student | **3.48 ± 0.17** | 3.85 ± 0.06 | 0.200 ± 0.041 | **0.190 ± 0.040** |
| Wine | 2.79 ± 0.12 | **2.58 ± 0.16** | **0.182 ± 0.018** | 0.203 ± 0.021 |

*Table 20.* Positive Class Performance: Sparsity and Loss. $\lambda = 0.005$, $\varepsilon = 0.01$, $\mu = 0.01$, $\alpha = 0.05$

# F. Description of Datasets and Methods

Table 21 provides a brief overview of the datasets used in this paper, including summary statistics and any dataset-specific processing steps in addition to the standard procedures described below.

All categorical features were one-hot encoded, and all continuous features were discretized using the GOSDT+Guesses strategy introduced by (McTavish et al., 2022). We also applied the GOSDT+Guesses strategy to perform feature selection for binary features (using num estimators = 100). Unless missing entries could be straightforwardly filled without imputation, any rows containing missing data were discarded. All reported statistics correspond to the datasets after every processing step, including feature selection.

In cases of substantial class imbalance, we upsampled the minority class so that the resulting models would not be trivial. Table 21 shows the dataset sizes in the original, numerical, processed form as well as an approximate number of binarized features obtained when applying (McTavish et al., 2022) on 5 random train test splits of the datasets.

*Table 21.* Summary count statistics of all datasets after preprocessing

| Data Name | # samples | # features | # binarized features | Processing notes |
|---|---|---|---|---|
| MGH | 5000 | 28 | $12.6 \pm 2.1$ | |
| NIJ Recidivism Challenge | 16264 | 14 | $13.8 \pm 0.4$ | Fill missing prison offenses with 'Unknown' |
| COMPAS (Recidivism) | 6907 | 7 | $27.4 \pm 1.7$ | Same as (Xin et al., 2022; Semenova et al., 2023) |
| FICO (Credit) | 10459 | 23 | $41.6 \pm 2.9$ | Same as (Xin et al., 2022; Semenova et al., 2023) |
| monks1 | 124 | 11 | $8.2 \pm 1.3$ | Same as (Xin et al., 2022; Semenova et al., 2023) |
| monks2 | 169 | 11 | $8.4 \pm 0.9$ | Same as (Xin et al., 2022; Semenova et al., 2023) |
| monks3 | 122 | 11 | $6.4 \pm 1.1$ | Same as (Xin et al., 2022; Semenova et al., 2023) |
| Breast Cancer Wisconsin | 699 | 10 | $9.8 \pm 0.4$ | Same as (Xin et al., 2022; Semenova et al., 2023) |
| Car Evaluation | 1728 | 15 | $8.6 \pm 0.5$ | Same as (Xin et al., 2022; Semenova et al., 2023) |
| bar | 1913 | 15 | $11.6 \pm 0.9$ | Same as (Xin et al., 2022; Semenova et al., 2023) |
| bar7 | 1913 | 14 | $8.6 \pm 0.5$ | Same as (Xin et al., 2022; Semenova et al., 2023) |
| Carryout Takeaway | 2280 | 15 | $10.8 \pm 1.1$ | Same as (Xin et al., 2022; Semenova et al., 2023) |
| Coffee House | 3816 | 15 | $12.2 \pm 0.8$ | Same as (Xin et al., 2022; Semenova et al., 2023) |
| Restaurant 20 | 2653 | 15 | $12.4 \pm 0.5$ | Same as (Xin et al., 2022; Semenova et al., 2023) |
| Heart | 297 | 24 | $19.6 \pm 2.4$ | |
| Mushroom | 8124 | 11 | $10.0 \pm 0.0$ | |
| Spambase | 4601 | 67 | $29.6 \pm 0.5$ | |
| Wine | 6497 | 61 | $34.8 \pm 2.9$ | |
| Student | 649 | 31 | $19.8 \pm 2.6$ | |
| Aging | 714 | 57 | $25.0 \pm 1.9$ | |
| Adult | 48842 | 14 | $13.4 \pm 0.5$ | |
| Bike | 17379 | 104 | $26.0 \pm 1.0$ | |

*Table 22.* Licensing and Data Source Information for all Datasets

| Data Name | License | Citation |
|---|---|---|
| MGH | N/A | N/A |
| NIJ Recidivism Challenge | Publicly Available | (NIJ, 2021) |
| COMPAS (Recidivism) | Publicly Available | (Angwin et al., 2016) |
| FICO (Credit) | FICO | (FICO, 2018) |
| monks1 | CC BY 4.0 | (Wnek, 1992) |
| monks2 | CC BY 4.0 | (Wnek, 1992) |
| monks3 | CC BY 4.0 | (Wnek, 1992) |
| Breast Cancer Wisconsin | CC BY 4.0 | (Wolberg, 1992) |
| Car Evaluation | CC BY 4.0 | (Bohanec, 1997) |
| bar | CC BY 4.0 | (Wang et al., 2020) |
| bar7 | CC BY 4.0 | (Wang et al., 2020) |
| Carryout Takeaway | CC BY 4.0 | (Wang et al., 2020) |
| Coffee House | CC BY 4.0 | (Wang et al., 2020) |
| Restaurant 20 | CC BY 4.0 | (Wang et al., 2020) |
| Heart | Publicly Available | (Janosi et al., 1989) |
| Mushroom | Publicly Available | (Guide, 1981) |
| Spambase | Publicly Available | (Hopkins et al., 1999) |
| Wine | Publicly Available | (OpenML, 2025) |
| Student | Publicly Available | (Cortez, 2008) |
| Aging | Publicly Available | (Malani et al., 2017) |
| Adult | Publicly Available | (Becker & Kohavi, 1996) |
| Bike | Publicly Available | (Fanaee-T, 2013) |

