# OpenReview forum: "Falling Trees: A Model Class for Interpretable Risk Prioritization"
_ICML.cc/2026/Conference — ICML 2026 spotlight_

### Official Review · Reviewer_1scc · 2026-03-05

**Soundness:** 3
**Presentation:** 3
**Significance:** 3
**Originality:** 2
**Overall Recommendation:** 4
**Confidence:** 4

**Summary:**

The article introduces GRAVITree algorithm to compute the Rashomon sets of falling trees, a type of models combining decision trees and falling rule lists. GRAVITree is based on dynamic programming to solve a constraint optimization problem and find the Rashomon set of falling trees. Notice that GRAVITree is parametrized by parameter $\mu$ to set the shape of falling trees, where high values lead to model forms close to falling rule lists, and low values to model forms of decision trees.
Falling trees achieve improved tradeoffs between accuracy and interpretability than the original decision trees and falling rule lists, as shown in several experiments.

**Compliance With Llm Reviewing Policy:**

Affirmed.

**Final Justification:**

The authors efficiently addressed the main limitation I identified about the stability with respect to data perturbation in the rebuttal. From their additional experiments, GRAVITree seems stable under data perturbation. Then, I recommend acceptation.

**Key Questions For Authors:**

See Strengths and Weaknesses section.

**Limitations:**

yes

**Strengths And Weaknesses:**

The article tackles the important problem of building interpretable models, while preserving a good accuracy. Such models are of primary importance in high-stake domains, such as healthcare, as illustrated in the paper. The introduced method is based on Rashomon sets, which  is a key and quite new principle of interpretable models.
The article is well written, and the problem is clearly explained, with a rigourous formulation of the introduced algorithm.
Experiments are convincing to show that GRAVITree provides interesting tradeoffs between interpretability and accuracy over decision trees and falling rule lists.

The stability of GRAVITree with respect to data perturbation is not analyzed in the article, and we believe this is a critical limitation. Indeed, instability is one the main limitation of interpretable models with a simple structure, such as decision trees and falling rule lists. These models are well-known to be highly unstable when the training data is slightly pertubated, because many different models of these types exist and achieve about the same accuracy. This is precisely the issue addressed by the Rashomon set: instead of focusing on one arbitrary simple model, the Rashomon set provides all relevant models of the same type.

However, GRAVITree provides the Rashomon set for a given training dataset. In practice, doctors often rerun a model adding a new batch data or removing a small subpopulation to update or deepen the analysis. To be meaningful, the output Rashomon sets should be stable with respect to these small data perturbations. Otherwise, GRAVITree only provides Rashomon sets conditional on the training data, which is not really useful.

We are completely open to increase the overall score, if the authors can show the stability of GRAVITree when training data is perturbated. We think that such stability analysis is quite easy to conduct: in the experimental setup each dataset is randomly split in 80-20 train-test, which can be repeated to compare the overlap between the Rashomon sets obtained across runs. A stability metric can also be derived: the data is discretized by definition, so splits in two trees are identical if they split on the same bin of the same variable (and small variabilities of cut values at the same bin can be neglected). For example, the similarity metric between two trees can be  the proportion of identical splits, which can finally be extended to compute the similarity of two sets of falling trees, by matching trees with their most similar counterpart in the other set, and deriving the mean similarity of tree pairs.

---

> ### Author Rebuttal · Authors · 2026-03-30
>
> Thank you for your review, and we are glad that you found our problem, algorithm, and experiments convincing. We note that prior work has addressed stability concerns of Rashomon sets of regular sparse trees (e.g. Section 6.4 in [1]), showing that stability is maintained even after removing examples.
>
> However, we agree that stability for the Rashomon set of falling trees under perturbations of data should also be tested. We address this below according to your suggested mechanism.
>
> We ran a stability experiment with the following setup, on the COMPAS and FICO (HELOC) datasets from our benchmarks. We binarized the data using the GOSDT+Guesses method proposed by [2]. We fit Rashomon sets with a maximum depth of $5$ and a branching cost of $0.01$. On COMPAS, we used $\varepsilon=0.01$ and a min support of $0.05$, and on FICO we used $\varepsilon=0.03$ and a min support of $0.02$. These parameters were chosen to achieve Rashomon sizes in the tens-to-hundreds on most bootstraps.
>
> We divided the dataset into a training set, consisting of 80% of the data, and a held out test set consisting of 20% of the data. The similarity metrics that we used to compare two trees are the metric proposed in the review, as well as the hamming distance between their prediction vectors on the held out test set (i.e. the proportion of test samples that the trees disagree on).
>
> We implemented the proposed metric as follows. `f1` and `f2` consist of (feature name, threshold) tuples. We use a threshold of 1.0 because both datasets' features take on integer values, and the split points are all halfway between feature values (i.e. 21.5).
>
> ```
> overlap = []
> for f1 in feature_names1:
>     for f2 in feature_names2:
>         # add to the overlap if the feature names are the same
>         # and the thresholds are within the specified threshold
>         if f1[0] == f2[0] and abs(float(f1[1]) - float(f2[1])) <= 1.0:
>             if f1 not in overlap:
>                 overlap.append(f1)
>             if f2 not in overlap:
>                 overlap.append(f2)
>
> union = [f for f in feature_names1] + [f for f in feature_names2 if f not in feature_names1]
> similarity = len(overlap) / len(union) if len(union) > 0 else 1.0
> ```
>
> In each trial of our experiment, we subsampled two different bootstraps consisting of 80% of the training set (resulting in two bootstrap sets consisting of 80% x 80% = **64%** of the original data, since we do a 80-20 train test split). We fit a Rashomon set with the above parameters to each of these training sets. For each model in the first Rashomon set, we computed the closest model by each distance/similarity metric in the second Rashomon set. We then reported the average nearest distance/similarity over all models in the first Rashomon set. We repeated this experiment for 10 trials with independently sampled sub-training sets (with different random seeds every time).
>
> | dataset | mean tree similarity (std) | mean hamming distance (std) |
> | - | - | - |
> | COMPAS | 0.91 (0.047) | 0.0025 (0.0019) |
> | FICO | 0.74 (0.16) | 0.016 (0.01) |
>
> For the proposed tree similarity metric, on average each model in the first Rashomon set had a model in the second Rashomon set whose overlapping features (within the threshold tolerance) had a 91% overlap on COMPAS and a 74% overlap on FICO compared to the combined feature set.
>
> On the hamming distance metric, each model in the first Rashomon set had a model in the second Rashomon set that agreed on all but 0.25% of the input samples on COMPAS and 1.6% on FICO.
>
> We hope that this experiment helps with your concern regarding the stability of Rashomon sets of falling trees. We will incorporate a more thorough version of this experiment, applied to all of our datasets, into the final paper.
>
> [1] Xin et al., (2022) Exploring the Whole Rashomon Set of Sparse Decision Trees
>
> [2] McTavish et al., (2022) Fast Sparse Decision Tree Optimization via Reference Ensembles.

---

> > ### Author Rebuttal · Reviewer_1scc · 2026-04-01
> >
> > Thank you for these additional experiments to analyze stability. From the two tested datasets, GRAVITree seems quite stable. I will increase my score.

---

### Official Review · Reviewer_ZBbd · 2026-03-09

**Soundness:** 3
**Presentation:** 3
**Significance:** 3
**Originality:** 3
**Overall Recommendation:** 5
**Confidence:** 4

**Summary:**

The paper proposes a new method GAVITree, based on the state-of-the art interpretable falling rule lists. The novel approach finds the Rashomon set of falling trees. The extensive numerical experiments show that GAVITree - a dynamic-programming-with-bounds algorithm -- increases the decision sparsity and results in an optimal performance.

**Compliance With Llm Reviewing Policy:**

Affirmed.

**Key Questions For Authors:**

I would appreciate some clarifications. First, Figure 1 and Figure 2 seem to illustrate the same idea, or did I miss anything? Second, the Rashomon sets of falling trees make think about a number of different trees produced by the method and gathered further in an ensemble manner. However, the sentence "the Rashomon sets for a node's left and right children.." in Section 4 changes this impression. Is the Rashomon set estimated for nodes or for trees?
I was also looking for the merging and filtering procedure applied to the Rashomon sets, could you point out what you do exactly?

**Limitations:**

Yes (the authors adequately discussed the limitations).

**Strengths And Weaknesses:**

Strengths. The paper is well-written and the method is sound. The numerical experiments are convincing. The approach constructs interpretable models and can be practical for real-life applications.

Weaknesses. The paper is rather dense (it might be difficult to read it for someone not working on similar topics).
While reading there is an impression that the proposed approach orders the rules in a monotonicically decreasing risks order, however, it is a property of the Falling Rule Lists which are used in the proposed method, and is not a novelty of the proposed approach. This moment was misleading, since a reader (at least I) expected the monotonicity constraint in the paper.

---

> ### Author Rebuttal · Authors · 2026-03-30
>
> Thank you for your review! We are glad that you found our approach and experiments convincing and practically applicable.
>
> > **Weakness 1:** The paper is dense.
>
> Thank you for this feedback! We will include additional exposition and text explanation of our theory and algorithms in a revised version of the paper. We’ll also reduce some notational overhead in the Algorithm pseudocode and add some figures in the Appendix illustrating the flow of information and bounds in the algorithm. If there are particular sections that you found challenging to read, please do let us know, so we can pay particularly close attention to them.
>
> > **Weakness 2:** Our paper presents the monotonicity constraints in a misleading manner.
>
> We discussed the work of Wang & Rudin (2015) and Chen & Rudin (2018) in the introduction, and we introduced our idea as a generalization of falling rule lists to falling trees. To the best of our knowledge, this is a novel generalization of the falling constraint to a more expressive model class. This is not trivial – the falling constraint, as proposed for rule lists, is not immediately applicable for trees, nor is the Rashomon set computation. We are happy to further clarify this in the introduction and description of the algorithm.
>
> > **Question 1:** Do Figures 1 and 2 illustrate the same idea?
>
> Our intent was for Figure 1 to present an example of a falling tree, and for Figure 2 to demonstrate the concepts of a) the falling constraint, b) a path from root to leaf, and c) what it means for leaf nodes to be adjacent to the path. Although the figures are similar, we agree that Figure 1 could indeed be used for both purposes. Thank you for your suggestion.
>
> > **Question 2:** Is the Rashomon set estimate for nodes or for trees? And do we use the Rashomon set for ensembling?
>
> A node can be viewed as a state in the search graph corresponding to a subproblem. Our method computes Rashomon sets recursively: for a given node (subproblem), we first compute the Rashomon sets for its left and right child subproblems induced by each possible split, and then combine and filter these to obtain the Rashomon set for the current node.
> Importantly, although Rashomon sets are computed at the level of subproblems (nodes), each set consists of trees that solve that subproblem. Thus, both intermediate subproblems and the root problem ultimately yield sets of trees.
> Regarding your question about ensembling – this is one potential application of the Rashomon set, but our goal in this paper is simply to compute the entire set of near-optimal falling trees.
>
> > **Question 3:** Can we explain the merging and filtering process?
>
> Thank you for your question. The main goal of Algorithm 1 is to find the Rashomon set recursively. That is, given a subproblem (i.e. a node in the search graph we encounter), we find the Rashomon set of its left and right subproblems. Once these sets are found, we can merge them by taking their cross product. The resulting set, however, can contain trees which a) lie outside the specified Rashomon bound, and b) are not falling. The filtering process removes these trees through constraint checks. Further details, as well as the exact merging and filtering algorithm, can be found in Section C.1 of the Appendix.

---

> > ### Author Rebuttal · Reviewer_ZBbd · 2026-04-01
> >
> > My score was "accept" and I keep it.

---

### Official Review · Reviewer_eP1N · 2026-03-12

**Soundness:** 3
**Presentation:** 3
**Significance:** 3
**Originality:** 3
**Overall Recommendation:** 5
**Confidence:** 4

**Summary:**

This paper introduces a novel hypothesis class, coined “falling trees”, which are falling rule lists where some tree-like branching is allowed. Therefore, “falling trees” have a structure in-between rule lists and decision trees, in which falling constraints (forcing higher-risk classifications to happen first, which is often required in high-risk settings such as intensive care units) are enforced.

The authors propose GRAVITree, a Dynamic-programming-with-bounds algorithm designed to efficiently build the Rashomon Set of this new hypothesis class, leveraging both its structure and the enforced falling constraints for efficient exploration, decomposition and pruning. By incorporating a branching penalty, the algorithm can control where the models lie on the spectrum between rule lists and full decision trees.

Experiments demonstrate the effectiveness of the proposed algorithm and the practical interest of falling trees, which can combine the simplicity of rule lists and the expressivity of decision trees to build falling models with better decision sparsity than decision trees, and better predictive performance than rule lists.

**Compliance With Llm Reviewing Policy:**

Affirmed.

**Final Justification:**

The rebuttal addressed my main concerns; therefore, I have opted to increase my score (see rebuttal acknowledgment).

**Key Questions For Authors:**

-	Theorem 4.2 states a sufficient condition verified within the proposed algorithms to ensure that the model being built satisfies the falling constraint. Isn’t this condition also necessary? I think it would be worth explicitly stating, even if it does not affect the correctness of the method. Indeed, ensuring a sufficient but non-necessary condition might prune arbitrarily more solutions than needed to enforce the constraint, therefore the method would be unable to certify the Rashomon Set.

-	On a related topic, is the condition enforced in Algorithm 2 strictly equivalent to the one stated in Theorem 4.2 ? Theorem 4.2 focuses on the case where an internal node has one leaf-child and one subtree-child. However, it seems like the positive rates $p_1^+$ and $p_2^+$ are compared even if the two children are leaves and even if they are both subtrees. Could you clarify this point ?

-	One important component speeding up exploration is the designed quantized caching strategy. While the paper states that “this approach makes the algorithm faster without sacrificing optimality” (lines 311-312), experiments conducted in Appendix D.2 show that varying the quantization factor $q$ results in different Rashomon Set sizes. This suggests that the quantized caching results in a form of heuristic approximation of the algorithm, as otherwise it should not impact the algorithm’s output. Could you please elaborate on that ? I also think the paper should be more clear with this aspect. On the same line, Table 7 shows that increasing the quantization factor $q$ typically increases the size of the output Rashomon set (e.g., NIJ Recidivism dataset), but sometimes also decreases it (e.g., Car evaluation dataset). Why is that the case ?

-	How does the runtime efficiency of GRAVITree change when falling constraints are not enforced?

**Limitations:**

The proposed algorithm has two main limitations:

-	It operates on binary features, which means that if some features are numerical, they must be binarized beforehand, as done in the experiments. In general, this means that optimality (and therefore the Rashomon set) can only be achieved with respect to the binarized data, not the original one.

-	Instead of incorporating a sparsity regularizer (with a coefficient $\lambda$ which would indicate “the user is willing to sacrifice $\lambda$ accuracy for sparsity”), the approach relies on predefined maximum size and minimum support constraints. However, it is unclear how these constraints should be set in practice without expensive tuning, as too large values would hinder sparsity with no accuracy gain while too low ones would result in poor predictive performance.

**Strengths And Weaknesses:**

Strengths:

-	The paper is clearly written and structured

-	Generalization of falling constraints to tree-structured models is sound

-	The proposed GRAVITree algorithm uses effective bounds, problem decomposition strategies (mostly leveraging the falling constraints and the tree structure) and caching to efficiently construct the Rashomon Set of falling trees

-	The ability of the proposed approach to interpolate between rule lists and decisions trees by controlling a tree imbalance-based regularization term is very interesting and empirically very effective. I believe that falling trees are a great idea and can have real-world applications in high-risk contexts

-	The experimental evaluation is thorough, including comparison with meaningful baselines (including new ones adapted from the literature) and ablation study

Weaknesses:

-	I have a few technical questions (detailed in my Key questions for authors textbox) that need to be clarified

-	Notation is not always clear:
$n$ is used to denote the number of examples in dataset $D$, and also to denote an internal node (in definition of $H(T)$). Using notation in a more consistent manner (e.g., rather using $v$ to denote nodes as in Theorem 4.2) would facilitate understanding.
Similarly, the set of internal nodes within tree $T$ is introduced as $N(T)$ but simply used as $N$ later.
In Algorithm 2, what does $1[T_1,T_2]$ stand for ?
In Algorithm 2, why does the condition for replacing subtree 2 by a leaf compare the potential new cost () only with $\mathcal{L}_2$ ? (shouldn’t the right-hand side of the inequality be $\mathcal{L}_j$ ?)

-	The approach is only applicable for binary features (see my Limitations textbox)

-	Since the objective does not explicitly penalize model size (e.g., number of leaves/rules), the algorithm may return arbitrarily large models whenever additional splits do not increase the branching cost and do not worsen training misclassification. While this is partly controlled by hard constraints on maximum depth and minimum support, these hyperparameters generally require tuning in practice, and the paper does not discuss how they should be selected or how sensitive results are to these choices (see my Limitations textbox).

-	While falling constraints are clearly relevant in certain high-stakes applications, they may be less central in more general settings. Since the efficiency of the proposed method partly relies on these constraints through specialized bounds and pruning strategies, I wonder to what extent GRAVITree’s runtime is affected when falling constraints are removed. This would be particularly useful to report, as Appendix D.1 does not include runtime results.

Minor:

-	Line 186: $\mathcal{L}(T;D)$ -> should be $\mathcal{L}(T,D)$

-	Inputs of Algorithm 2: “global dataset Size” -> should be “global dataset size”

-	Theorem 4.1: “Assume that we running Algorithm 2” -> “we are”

-	Figures are not displayed in the order in which they are discussed in the text

-	Figure 4 is never mentioned in the text

---

> ### Author Rebuttal · Authors · 2026-03-31
>
> Thank you for your thoughtful review. We are glad that you found our problem set-up, algorithms, and experiments convincing, and we appreciate your useful feedback. We will include the minor revisions in the final paper.
> > **W1:** Clarity of notation.
>
> Thank you for pointing out these examples. We will revise our paper to avoid overloading of the same variable names and ensure consistent notation.
>
> > **W2/Limitation 1:** The approach is only applicable for binary features.
>
> It is true that our approach is only optimal with respect to the binarized data. There are binarization strategies and algorithms for rule-based models that seek to minimize this potential optimality gap [1, 2]. There is also recent research that explores when simple models, e.g., decision trees, perform competitively with any model, in which case this optimality gap will also be small (see [3]).
>
> [1] McTavish et al., (2022) Fast Sparse Decision Tree Optimization via Reference Ensembles.
>
> [2] Briţa et al. (2025) Optimal Classification Trees for Continuous Feature Data.
>
> [3] Semenova et al., (2023) A Path to Simpler Models Starts with Noise.
>
> > **W3/Limitation 2:** The paper does not discuss how to tune hyperparameters (specifically max depth and min support) to avoid finding arbitrarily large models.
>
> Thank you for pointing this out! Please see our response to reviewer DY65, **W1/Q1**, for discussion on tuning best practices and recommended values of maximum depth, minimum support, and branching cost. We show that the combination of these parameters strongly regularizes model size in practice. In addition, the falling constraint prevents making splits that do not obey the monotonicity constraint. The deeper the tree, the stronger this constraint gets, because the upper bound on the maximum acceptable probability is reduced, which reduces the rate at which the Rashomon set grows with depth.
>
> > **W4/Q4:** How does GraviTree’s runtime change if the falling constraint is not enforced?
>
> Please refer to Table 6 in the Appendix, which shows that the falling constraint improves runtime on all of our benchmark datasets, in some cases significantly (2-4x).
>
> > **Q1:** Is Theorem 4.2 also a necessary condition?
>
> Yes it is. Assume there exists an internal node $v$ where one child is a leaf and the other is an internal node such that the leaf’s empirical probability is less than the maximum leaf probability below the internal node. Then the leaf probabilities on the path through $v$ to that maximum probability leaf are not monotonically decreasing, and $T$ cannot be falling. We will modify the Theorem to state and prove that this is a necessary and sufficient condition.
>
> > **Q2:** Is the condition in Theorem 4.2 the same as the one applied on Algorithm 2?
>
> Yes - the condition in Theorem 4.2 is recursively applied in Algorithm 2 (lines 33-40, where we tighten the falling budget based on the solutions to subproblems). If the condition is satisfied recursively at all nodes that have a leaf in one child and a node in the other, then the tree must be falling. We also apply a similar bound in Algorithm 1 and during the merge and filter operation once the child Rashomon sets have been found.
>
> > **Q3:** Can we elaborate on the heuristic aspect of the quantized caching method? Why is the behavior of the Rashomon set size inconsistent under this approximation?
>
> On lines 311-312, we meant to say that, although it is a heuristic, quantized caching resulted in minimal changes to optimality and Rashomon set size in our experiments. We will make this clearer in the paper.
>
> Quantized caching affects the Rashomon set size in two opposing ways. 1) it can result in solving the global optimal tree problem (Alg 2) to suboptimality. Since the Rashomon bound depends on the training objective of the optimal tree, quantized caching results in a larger Rashomon bound. 2) it can result in child subproblems in the search graph induced by Alg 1 being solved to suboptimality. This can result in trees being left out of the Rashomon set, because the remaining budget decreases (e.g. lines 27, 34 in Alg 1). The practical effect on the Rashomon set size depends on the relative strength of these forces.
>
> To isolate the effect of quantized caching on the solutions to subproblems in the Rashomon set, we ran experiments with an absolute Rashomon bound. This is different from Table 7 in Section D.2, since that table had a Rashomon bound that varied with the quantization level, i.e., B(q) = (1+eps)*loss of best tree(q). Now, the upper bound is fixed as B = (1+eps)*loss of best tree(q=0.01) for all quantized caching levels.
>
> Our results can be found at: https://docs.google.com/document/d/e/2PACX-1vSFwEln62iIGu7c0THxrINDqAas-5Z45cYc4PGbLdWQoFsbYfj9QKPADlmMuEAD8JyFqJyWA0TjopFx/pub
>
> In this setting, we see that quantization results only in a small reduction in Rashomon set size (e.g. 15813 vs 15870 trees for COMPAS). As in the paper, we get a runtime boost of 1.5-2x across the datasets tested.

---

> > ### Author Rebuttal · Reviewer_eP1N · 2026-04-03
> >
> > Thank you for your detailed answer. I think that the discussed modifications (in particular, modifying Theorem 4.2 to state and prove that the condition is both necessary and sufficient) will benefit the paper. The detailed discussion on hyper-parameter tuning (answering my question but also the concerns of Reviewer DY65) is also worth including in the revised paper, as well the discussion and complementary results on the heuristic effect of quantized caching. Based on these clarifications, I have raised my score.

---

### Official Review · Reviewer_DY65 · 2026-03-13

**Soundness:** 3
**Presentation:** 3
**Significance:** 3
**Originality:** 4
**Overall Recommendation:** 5
**Confidence:** 4

**Summary:**

The authors introduce Graivtree - a novel dynamic programming method for learning the Rashom set of of fallling trees with branching. Unlike competing methods, Graivtree does not rely on rules being preconstructed, and uses a novel objective function that penalizes both complexity and mislcassification in order to get simple representations of rules. The authors run Graivtree on 21 datasets in order to show improvement over competing methods.

**Compliance With Llm Reviewing Policy:**

Affirmed.

**Final Justification:**

Authors provided detailed responses to all my concerns and their rebuttal has strengthened the work. I believe the work is novel and significant, and should be accepted.

**Key Questions For Authors:**

- Can you describe a hyperparameter tuning design within the algorithim? How sensitive are the sets to changes in hyperparameters?
- Are there examples of when practioniers should explicity prefer falling trees over unconstrained sparse trees? Are there counterexamples? Specific examples and how interpretability would change would them would be very interesting.

**Limitations:**

The authors do not explicitly mention limitations. A few that can be mentioned are structural restrictions, namely that Graivtree assumes a monotone falling constraint along a tree. If a model does not have this structural restriction, Gravitree will misrepresent it's decision boundary. Other limitations may arise from the reliance on hyperparameter tuning.

**Strengths And Weaknesses:**

- The set up and motivation are clear - the authors do a convincing job of explaining why falling rule lists need to be extended and the need for branching.
- The branching penalty is simple but makes intuitive sense, and is well-supported by algorithmic design and empirical results showing improvements to backup the new method.

- From section E, there are several hyperparameters to be tuned, but there is no quantitative or empirical rational for tuning.
- Ultimate interpretability of models is not explored though it is emphasized as a benefit initially.

---

> ### Author Rebuttal · Authors · 2026-03-31
>
> Thank you for your review, and we are glad that you found our problem setup, algorithms, and experiments convincing.
>
> > **Weakness 1/Question 1:** There is no quantitative or empirical rationale for tuning hyperparameters. Can we describe the hyperparameter tuning procedure? How sensitive are our results to changes in hyperparameters?
>
> **Tuning procedure:**
>  The branching cost can typically be set to ~[0.01, 0.02], which yields near rule-list structures (1–2 branches) with moderate imbalance (as reflected by the normalized Colless index in Figure 3).
> The most influential hyperparameters are min support and eps, which primarily control runtime and Rashomon set size. A practical tuning strategy is:
> - Start with min support = 0.1, eps = 0.005. The algorithm should be very fast in this regime.
> - Gradually reduce min support / increase eps until the Rashomon set size, runtime, and test loss match acceptable limits.
>
> Across 20+ datasets, we found min support = 0.05 and eps = 0.01 to provide a good balance between accuracy, sparsity, and computational cost; we recommend these as default values. Consistent with prior work [1,2], we also observed that depth 4–5 is near-optimal for tabular data.
>
> **Sensitivity analysis:** We use the COMPAS dataset (~7k samples, ~30 features) as an example, and we will include guidelines in a revised version of the paper. The results of our analysis can be found at: https://docs.google.com/document/d/e/2PACX-1vRZwwH-LnNFDDIY4gTJ4ByJuXB3ro4LntNXZocuUD1nRqiYMu_6K1vrwPhwCop123xUr3pC-mLV5kno/pub
>
> - **Depth:** We find minimal test loss differences between depths 3, 4 and 5 on the COMPAS dataset, while runtime and Rashomon set size increase substantially with depth. See Table 1 in the link.
>
> - **Quantization Caching:** We recommend a default quantization level between 0.01-0.1 to maximize runtime benefits without sacrificing Rashomon set size or performance. See results in our response to Reviewer eP1N, Question 3.
>
> - **Min Support:** This parameter strongly affects runtime and Rashomon set size. Lower values (e.g., 0.01) increase both (5-10× runtime, 10-100× set size), while higher values reduce them with slight loss in accuracy. See Table 2 in the link.
>
> - **Epsilon:** Runtime increases with epsilon but modestly (e.g., doubling from 0.01 to 0.02 increases runtime by ~5–10%), likely due to effective pruning. See Table 3 in the link.
>
> [1] Babbar et. al. (2025) Near Optimal Decision Trees in a SPLIT Second
>
> [2] van der Linden et. al. (2025) Optimal or Greedy Decision Trees? Revisiting their Objectives, Tuning, and Performance.
>
> > **Weakness 2/Question 2:** The ultimate interpretability of falling trees is underexplored. Are there examples of when practitioners should explicitly prefer falling trees over unconstrained sparse trees? Counterexamples?
>
> Falling trees already inherit the interpretability benefits of regular sparse trees. The additional constraint makes them simpler to understand and easier to work with. The falling rule list with $\mu=0.01$ in Figure 5 in Appendix B provides a good example.
>
> In the medical triage setting, a falling tree allows rapid identification of high risk individuals when administering tests and checking clinical records.
>
> A reason not to use a falling tree would be if the falling constraint led to a large drop in test performance relative to regular sparse trees - in almost all of our benchmark datasets this is not the case. Also, if the decision sparsity for the positive class is not relevant to a problem then falling trees add an unnecessary constraint. However, even in this case, we found that the combination of the branching cost and falling constraint led to trees that are easy to understand.
>
> > **Limitation 1:** If a model does not satisfy the falling constraint, GraviTree will misrepresent its decision boundary.
>
> Prior work on the Rashomon Effect [1, 2] showed that on noisy real-world data there often exist many different equally good accurate models. Even if the original data is generated by a tree that does not obey the falling constraint, the multiplicity of good models suggests that there is likely to be a falling tree that has similar performance to the tree that generates the data. In our experimental results, we show that the gap in test performance between regular sparse trees and GraviTree is small (see Table 2). We also show that GraviTree offers advantages such as decision sparsity on the positive class and a better false negative rate. Our goal is to find an easy-to-use set of good interpretable models, not necessarily to exactly recover the true model generating the data.
>
> [1] Rudin et al. (2024) Amazing Things Come From Having Many Good Models.
>
> [2] Semenova et al., (2023) A Path to Simpler Models Starts with Noise.

---

> > ### Author Rebuttal · Reviewer_DY65 · 2026-04-03
> >
> > My questions and concerns have been adequately addressed and I thank the authors for their detailed responses and additional experiments. I have increased my score.

---

### Decision · Program_Chairs · 2026-04-30

**Decision:**

Accept (spotlight)

**Comment:**

All reviewers agree that the paper is well-written and tackles an important problem in an interesting way. I agree with them. The rebuttal discussion was pleasant and effective; it cleared up minor confusions and errors, showed the method performs well on additional settings, and overall led to a higher appreciation of the work. I'm happy to recommend acceptance of the paper, with the expectation the authors make the corrections and include the additional experimental results like they indicated they will.

Ps. I love the name of the method.